# PAC-Bayesian Meta-Learning
# for Few-Shot Identification of Linear Dynamical Systems

**Chenfeng Huang**  *chenfenghuang@ucla.edu*
*Department of Statistics & Data Science*
*University of California, Los Angeles*

**George Michailidis**  *gmichail@stat.ucla.edu*
*Department of Statistics & Data Science*
*University of California, Los Angeles*

**Reviewed on OpenReview:** *https://openreview.net/forum?id=CiGFpSLzFv*

## Abstract

Identifying linear time-invariant (LTI) dynamical systems from data is especially challenging when trajectories are short, noisy, or high-dimensional. Traditional system identification methods typically treat each system in isolation and therefore fail to exploit shared structure across related systems. We propose a PAC-Bayesian meta-learning framework for few-shot LTI system identification (PBML-LTI), which learns a transferable prior over task-specific dynamics while preserving task-level heterogeneity. Each task corresponds to an unknown LTI system, and a meta-learner uses a collection of training trajectories to learn a data-dependent prior over transition matrices. Given a new system with limited trajectory data, PBML-LTI performs Bayesian adaptation under the learned prior to produce a task-specific posterior, yielding both accurate point estimates and principled uncertainty quantification in the few-shot regime.

A key technical challenge is temporal dependence: trajectories generated by LTI systems violate the i.i.d. assumptions underlying most existing PAC-Bayes analyses for meta-learning. To address this, we develop a martingale PAC-Bayes analysis for dependent trajectory losses and use it to motivate a fit–KL surrogate objective for meta-training. The resulting support–query predictive-risk bound clarifies how empirical fit, posterior complexity, and prior quality interact in few-shot adaptation under sequential dependence. We further show how this predictive bound induces problem-specific corollaries for transition-matrix recovery and multi-step trajectory prediction. Together, these results connect uncertainty-aware meta-identification with finite-sample analysis for dependent dynamical data.

## 1 Introduction

Linear time-invariant (LTI) dynamical systems are a foundational modeling framework in control, signal processing, and time-series analysis. A central challenge in this setting is to recover the unknown transition operator from finite observations. Although this problem can be written in a regression form, it differs fundamentally from standard linear regression because the covariates are generated by the system itself. The resulting temporal dependence violates the independence assumptions underlying classical statistical learning theory and makes estimation sensitive to both trajectory length and system stability. Recent non-asymptotic analyses have clarified these challenges and established finite-sample error guarantees across stable, marginally stable, and unstable regimes (Shirani Faradonbeh et al., 2018; Simchowitz et al., 2018; Sarkar & Rakhlin, 2019).

In many modern applications, however, system identification is not performed once for a single system, but repeatedly across a collection of related systems, such as different devices, subjects, environments, operating

modes, or geographic regions. Such multi-system settings often exhibit shared latent structure that can be exploited to improve data efficiency when individual trajectories are short or high-dimensional. Existing joint and multi-task approaches leverage this structure by pooling information across systems, typically through shared-basis, low-dimensional, or representation-based assumptions, in order to improve estimation accuracy over the observed collection of tasks (Lin et al., 2024; Modi et al., 2024; Chen et al., 2023).

Yet many downstream objectives require a different capability: rapid adaptation to a previously unseen system from only a small amount of new data. This setting arises when deploying dynamical models across populations of related but heterogeneous entities, where each system has its own local dynamics and only a short calibration window is available. For example, one may need to tune a controller for a new aircraft operating condition or configuration (Bosworth, 1992), or calibrate a forecasting model for a newly observed region or economic unit in macroeconomic or financial monitoring settings (Pesaran, 2015; Stock & Watson, 2016). In such cases, the primary challenge is no longer only accurate estimation on the observed systems, but reliable few-shot adaptation to new ones under strict data constraints.

This challenge is especially important when long trajectories are expensive, unsafe, or infeasible to collect because of changing operating conditions, nonstationarity, or limited observation horizons (García & Fernández, 2015; Ji et al., 2023). Similar constraints arise in physiological system identification, where only short data segments may be practically available (Ludvig & Perreault, 2012). These considerations motivate a shift in perspective: rather than only pooling data to improve estimation on the training cohort, we seek to extract transferable structure that supports principled few-shot adaptation and uncertainty quantification on previously unseen systems.

To this end, we propose a Bayesian meta-learning framework for few-shot identification of linear dynamical systems. We treat each system as a task drawn from a shared environment and use trajectories from multiple training tasks to learn a data-dependent prior over system dynamics. When a new task is encountered, this learned prior enables rapid adaptation from limited observations through Bayesian inference, yielding both a point estimate and a calibrated measure of uncertainty. Unlike joint learning approaches that primarily use shared structure to improve estimation on the observed tasks, our objective is to learn an inductive bias that is useful for adaptation to future systems (Finn et al., 2017; Patacchiola et al., 2020; Muthirayan et al., 2025).

This formulation raises a central theoretical question: how can we characterize generalization to new systems when adaptation is based on limited, temporally dependent data? PAC-Bayes theory provides a natural framework for this purpose, as it relates generalization to a tradeoff between empirical fit and a complexity penalty measured by the Kullback–Leibler divergence to a prior (McAllester, 1999; Seeger, 2002; Catoni, 2007). This is especially appealing in meta-learning, where the quality of the learned prior directly affects the speed and reliability of adaptation to new tasks (Liu et al., 2021; Rezazadeh, 2022; Guan & Lu, 2022; Farid & Majumdar, 2021). However, most existing PAC-Bayes meta-learning analyses assume that observations within each task are independent, whereas system identification typically provides a single temporally dependent trajectory per task.

To bridge this gap, we develop a martingale PAC-Bayes analysis tailored to dependent dynamical data and use it to justify a practical fit–KL surrogate for meta-training. The resulting theory yields a support–query predictive-risk perspective for few-shot adaptation under temporal dependence and clarifies how empirical fit, posterior complexity, and prior quality interact in this setting. This connects uncertainty-aware meta-identification with finite-sample analysis for dependent dynamical systems.

Our contributions are summarized as follows:

- We propose PBML-LTI, a PAC-Bayesian meta-learning framework for few-shot identification of linear time-invariant dynamical systems that learns a transferable prior and enables closed-form task adaptation via conjugate Bayesian inference.

- We develop a martingale PAC-Bayes analysis for temporally dependent trajectory losses and use it to motivate a fit–KL surrogate objective for meta-training.

- We show through synthetic and real fMRI-based experiments that PBML-LTI achieves improved data efficiency and robust multi-step prediction in high-dimensional, few-shot regimes.

## 2 Related Work

**Finite-sample identification of LTI systems from dependent trajectories.** Classical system identification typically treats dynamical systems in isolation, focusing on estimation from a single trajectory where covariates are endogenously generated and thus temporally dependent. A rich body of non-asymptotic analysis has established sharp finite-time guarantees across stable, marginally stable, and unstable regimes, characterizing how sample complexity scales with the time horizon, system dimension, and spectral properties (Shirani Faradonbeh et al., 2018; Simchowitz et al., 2018; Sarkar & Rakhlin, 2019). These foundational results underscore two critical constraints that our framework must address: the necessity of rigorously handling intra-task sequential dependence, and the dominant role of stability in determining estimation error when data are scarce.

**Joint and multi-task identification across related systems.** Recent advances address the identification of multiple LTI systems by exploiting shared structural assumptions, such as common bases, low-dimensional subspaces, or coupled optimization objectives (Modi et al., 2024; Chen et al., 2023). These joint-learning formulations excel at improving estimation accuracy across a fixed collection of observed systems by "borrowing strength" between tasks, aligning with classical multi-task learning paradigms (Argyriou et al., 2008). In contrast, our framework targets few-shot generalization to previously *unseen* systems. Here, the goal is not merely to refine estimates for the training set, but to leverage the learned structure to rapidly and reliably calibrate a new system from a short trajectory at deployment time.

**Meta-learning, hierarchical Bayes, and Bayesian few-shot adaptation.** Meta-learning formalizes the problem of learning transferable structure that supports rapid adaptation to new tasks drawn from a shared environment. Gradient-based approaches such as MAML optimize performance after a small number of task-level updates and were studied in both supervised and reinforcement-learning settings (Finn et al., 2017). Thus, the mere presence of non-i.i.d. data within a task is not, by itself, unique to our setting. A complementary perspective interprets meta-learning through hierarchical Bayes, where training tasks are used to learn a prior and test-time adaptation corresponds to posterior inference (Grant et al., 2018). Several Bayesian meta-learning methods instantiate this perspective in general few-shot supervised settings, including PLATIPUS (Finn et al., 2018), ABML (Ravi & Beatson, 2019), and iBAML (Zhang et al., 2023); related uncertainty-aware meta-learning ideas have also been explored in settings such as deep kernel transfer (Patacchiola et al., 2020) and online control (Muthirayan et al., 2025). Our approach is complementary but tailored to few-shot LTI identification from temporally dependent trajectories. In particular, we exploit a conjugate matrix-normal/Gaussian model so that task adaptation is available in closed form, rather than through iterative gradient-based, amortized, or implicit-gradient updates.

**PAC-Bayes generalization and meta-learning.** PAC-Bayes theory bounds generalization error through a tradeoff between empirical fit and a complexity term measured by the KL divergence to a reference prior (McAllester, 1999; Seeger, 2002; Catoni, 2007). This makes it particularly natural for meta-learning, where the quality of the learned prior directly affects adaptation to new tasks (Liu et al., 2021; Rezazadeh, 2022; Guan & Lu, 2022; Farid & Majumdar, 2021). Most existing PAC-Bayes meta-learning analyses, however, assume that observations within each task are independent. By contrast, LTI system identification provides a single temporally dependent trajectory per task. Relative to single-task non-asymptotic analyses for dependent LTI data (Simchowitz et al., 2018; Sarkar & Rakhlin, 2019), the additional challenge in our setting is to move from guarantees for point estimators on one system to a support–query meta-learning problem involving task-specific posterior distributions, a learned prior shared across tasks, and a PAC-Bayes change-of-measure argument under temporal dependence. Our contribution is therefore not simply to handle non-i.i.d. within-task data, but to combine few-shot LTI identification, hierarchical Bayesian adaptation, closed-form conjugate task inference, and martingale PAC-Bayes analysis in a unified framework.

**Joint learning versus meta-learning in the experimental evaluation.** While both joint learning and meta-learning evaluate performance on unseen tasks, they differ fundamentally in their training objectives and adaptation mechanisms. Joint learning typically extracts a static shared structure (e.g., a common subspace) from training tasks, which is then applied "frozen" or with minimal adjustment to new data (Modi et al., 2024; Chen et al., 2023). In contrast, our experimental protocol *explicitly targets few-shot adaptation*. We adopt a support–query split: for every test task, the model receives only a short support prefix to calibrate a task-specific posterior distribution, while evaluation is conducted on a disjoint query window. This distinction is crucial; unlike joint learning, which optimizes aggregate performance over a cohort, our meta-learning objective is designed to maximize post-adaptation accuracy by leveraging the learned prior to recover dynamics from minimal data.

## 3 Problem Formulation

We consider the problem of identifying a family of related LTI dynamical systems. The available data comprise trajectories collected from multiple tasks, indexed by $m \in [M] := \{1, 2, \ldots, M\}$. For each task $m$ and time $t = 0, 1, \ldots, T_m - 1$, the system evolves according to

$$x_{m,t+1} = A_m x_{m,t} + w_{m,t+1}, \tag{1}$$

where $x_{m,t} \in \mathbb{R}^d$ denotes the system state, $A_m \in \mathbb{R}^{d \times d}$ is an unknown transition matrix, and $\{w_{m,t}\}_{t \geq 1}$ is a stochastic noise process. We assume the full state trajectory $\{x_{m,t}\}_{t=0}^{T_m}$ is observed for each task.

It is convenient to rewrite equation 1 in regression form. To this end, define the per-task data matrices

$$X_m := [x_{m,0}, x_{m,1}, \ldots, x_{m,T_m-1}] \in \mathbb{R}^{d \times T_m}, \qquad Y_m := [x_{m,1}, x_{m,2}, \ldots, x_{m,T_m}] \in \mathbb{R}^{d \times T_m}, \tag{2}$$

where $X_m$ collects the state vectors serving as predictors at each time step, and $Y_m$ collects the responses, corresponding to the next-step-ahead states. Let $\Xi_m := [w_{m,1}, \ldots, w_{m,T_m}] \in \mathbb{R}^{d \times T_m}$ denote the stacked process-noise matrix. Then, the trajectory can be written compactly as

$$Y_m = A_m X_m + \Xi_m. \tag{3}$$

A critical distinction from standard linear regression is that the columns of $X_m$ exhibit inherent temporal dependence and are adapted to the history of the noise process.

To complete the model formulation, we impose the following assumptions to ensure that the model is well posed and to clearly specify the relationships among the members of the family of LTIs in equation 1.

**Assumption 1** (**Controlled Growth**). *We assume the dynamics of each system exhibit controlled growth over the observed horizon. Specifically, there exists a constant $c_\rho > 0$ such that for all tasks $m$,*

$$\rho(A_m) \leq 1 + c_\rho/T_m, \tag{4}$$

*where $\rho(A_m)$ denotes the spectral radius of $A_m$. This type of controlled-growth condition allows near-marginal stability while preventing rapid explosion over the finite horizon, which is standard in non-asymptotic LTI identification and is also adopted in single and multi-system learning settings (Shirani Faradonbeh et al., 2018; Modi et al., 2024; Simchowitz et al., 2018; Sarkar & Rakhlin, 2019).*

**Assumption 2** (**Martingale Difference Noise Process**). *Let $\{\mathcal{F}_{m,t}\}_{t \geq 0}$ denote the natural filtration for task $m$, capturing the history of information available up to time $t$. Formally, $\mathcal{F}_{m,t}$ is the smallest $\sigma$-algebra generated by the initial condition and the noise history up to time $t$:*

$$\mathcal{F}_{m,t} := \sigma(x_{m,0}, w_{m,1}, \ldots, w_{m,t}).$$

*We assume the noise process forms a martingale difference sequence with respect to this filtration and is conditionally sub-Gaussian. That is,*

$$\mathbb{E}[w_{m,t+1} \mid \mathcal{F}_{m,t}] = 0, \tag{5}$$

*and for all $u \in \mathbb{R}^d$,*

$$\mathbb{E}\left[\exp(\langle u, w_{m,t+1}\rangle) \mid \mathcal{F}_{m,t}\right] \leq \exp\left(\frac{\sigma_w^2}{2}\|u\|_2^2\right). \tag{6}$$

*Additionally, we assume the conditional covariance is uniformly bounded by a fixed positive semidefinite matrix, satisfying $\mathbb{E}[w_{m,t+1}w_{m,t+1}^\top \mid \mathcal{F}_{m,t}] \preceq \Sigma_w$. These conditions place our analysis within the self-normalized martingale framework, a standard setting for analyzing sequential least squares and LTI identification from single trajectories (Abbasi-Yadkori et al., 2011; Simchowitz et al., 2018; Sarkar & Rakhlin, 2019).*

**Assumption 3** (**Shared Environment**)**.** *We posit a hierarchical generative process where tasks are related via a shared latent structure. Specifically, there exists an unknown hyper-parameter $\phi_\star$ governing the distribution of system dynamics, such that each task parameter $A_m$ is drawn independently:*

$$A_m \mid \phi_\star \overset{\text{i.i.d.}}{\sim} p(\cdot \mid \phi_\star), \qquad m = 1, \ldots, M. \tag{7}$$

*Conditional on $A_m$, the observed trajectories are generated according to the LTI dynamics in equation 1. This assumption of i.i.d. task sampling from a fixed meta-distribution is standard in learning-to-learn theory and serves as the foundation for establishing generalization guarantees to novel tasks (Baxter, 2000; Maurer, 2005; Finn et al., 2017; Liu et al., 2021).*

## 4 PAC-Bayesian Meta-Learning for LTI Identification (PBML-LTI)

In this section, we introduce PBML-LTI, progressing from the hierarchical Bayesian formulation to the practical meta-training objective and the corresponding few-shot adaptation procedure. We build directly on the setup in Section 3: Assumption 3 motivates learning shared structure across tasks through a common prior, while Assumptions 1–2 provide the regularity conditions needed for the martingale PAC-Bayes analysis developed later in this section.

### 4.1 Hierarchical Bayesian Model and Task Adaptation

We treat each LTI system as a task with task-specific transition matrix $A_m$, and we seek shared meta-parameters $\phi$ that define a transferable prior over the family $\{A_m\}$. When a new task is encountered with limited data, adaptation is performed via Bayesian inference under this learned prior, producing both a point estimate and a task-specific measure of uncertainty (Grant et al., 2018).

**Hierarchical Bayes interpretation of meta-learning.** Our formulation follows a three-level hierarchical Bayesian model:

$$\phi \sim p(\phi), \qquad A_m \mid \phi \sim p(A \mid \phi), \qquad D_m \mid A_m \sim p(D \mid A_m), \tag{8}$$

where $\phi$ represents the environment-level structure shared across tasks, $A_m$ specifies the task-specific dynamics, and $D_m$ denotes the observed trajectory for task $m$. In a fully Bayesian treatment one would place a hyper-prior on $\phi$ and infer both $\phi$ and $\{A_m\}$ jointly. In contrast, we adopt an empirical-Bayes perspective: $\phi$ is learned across training tasks, while task-level adaptation on a new system is carried out by exact posterior updating for $A_m$ conditional on the learned $\phi$. This retains the statistical advantages of a shared prior while avoiding the computational overhead of full posterior inference over $\phi$ (Grant et al., 2018).

**Prior Distribution And Working Likelihood.** Consistent with Assumption 3, we posit a shared prior family $P_\phi$ over task parameters and instantiate it as a matrix-normal distribution (Dawid, 1981; Gupta & Nagar, 2018; West & Harrison, 1997):

$$A_m \mid \phi \sim \mathcal{MN}(W, I_d, V), \qquad \phi := (W, V, \sigma^2), \tag{9}$$

where $W \in \mathbb{R}^{d \times d}$ is the shared prior mean, and $V \in \mathbb{R}^{d \times d}$ is symmetric positive definite. Equivalently,

$$\text{vec}(A_m) \sim \mathcal{N}(\text{vec}(W), V \otimes I_d),$$

so $V$ controls how the entries of $A_m$ co-vary across columns. This structure provides a convenient and expressive way to couple tasks through a shared mean and covariance, while still admitting efficient closed-form adaptation.

For tractable task-level inference, we adopt a conditional Gaussian working likelihood:

$$x_{m,t+1} \mid x_{m,t}, A_m \sim \mathcal{N}\big(A_m x_{m,t}, \sigma^2 I_d\big), \qquad t = 0, \ldots, T_m - 1. \tag{10}$$

This matrix-normal/Gaussian specification is adopted not merely because it is conjugate, but because it matches the structure of the problem. The transition parameter $A_m$ is intrinsically matrix-valued, and the matrix-normal prior preserves this structure directly rather than imposing it only after vectorization. Under the Gaussian working likelihood in equation 10, it yields closed-form task adaptation and exact expressions for the posterior mean and covariance, the posterior-expected squared loss, and the posterior-to-prior KL divergence used by PBML-LTI. At the same time, it provides an interpretable and computationally efficient structured prior: $W$ captures average dynamics shared across tasks, while $V$ encodes cross-column variability in a compact form. Richer conjugate or nonconjugate priors are possible, but they would substantially complicate the inner-loop posterior updates and the resulting fit–KL objective.

**Closed-Form Task Adaptation.** Given task data $X_m, Y_m$ from the matrix regression model equation 3, combining the Gaussian working likelihood equation 10 with the matrix-normal prior equation 9 yields a conjugate matrix-normal posterior (Dawid, 1981; Gupta & Nagar, 2018; West & Harrison, 1997):

$$Q_{m,\phi} = A_m \mid D_m, \phi \sim \mathcal{MN}(M_m, I_d, V_m). \tag{11}$$

The posterior covariance and mean are

$$V_m = \left( V^{-1} + \frac{1}{\sigma^2} X_m X_m^\top \right)^{-1}, \tag{12}$$

$$M_m = \left( \frac{1}{\sigma^2} Y_m X_m^\top + W V^{-1} \right) V_m. \tag{13}$$

We use the posterior mean $\widehat{A}_m := M_m$ as the task-level point estimate, and $V_m$ as the task-level uncertainty measure.

The same conjugate structure also yields closed-form expressions for the posterior expectation of the Gaussian loss and for the posterior-to-prior KL divergence. In particular,

$$\mathbb{E}_{A \sim Q_{m,\phi}}\big[\|Y_m - AX_m\|_F^2\big] = \|Y_m - M_m X_m\|_F^2 + d\operatorname{tr}(V_m X_m X_m^\top), \tag{14}$$

and

$$D_{\mathrm{KL}}(Q_{m,\phi}\|P_\phi) = \frac{1}{2}\left[ d\Big(\operatorname{tr}(V^{-1}V_m) - \log\det(V^{-1}V_m) - d\Big) + \operatorname{tr}\big((M_m - W)V^{-1}(M_m - W)^\top\big)\right]. \tag{15}$$

Thus, both the posterior expectation of the loss and the KL term can be computed exactly through matrix products, linear solves, traces, and log-determinants, without Monte Carlo approximation.

Closed-form adaptation therefore specifies how a learned prior $P_\phi$ is converted into a task-specific posterior $Q_{m,\phi}$. The remaining question is how to learn $\phi$ from many related training tasks in a way that supports generalization to unseen systems. To address this under temporal dependence, we next use the martingale PAC-Bayes structure to motivate a practical fit–KL surrogate for meta-training.

## 4.2 Martingale PAC-Bayes Motivation for Meta-Training

Within each task, observations form a single trajectory and are therefore not i.i.d. Classical PAC-Bayes results for independent samples do not apply directly because the one-step losses depend on the past through the state. A natural way to handle this dependence is to work with conditional risks and exploit the martingale structure induced by the filtration in Assumption 2.

For the remainder of this subsection, we fix an arbitrary task $m$ and suppress the task index for notational simplicity. Write $x_t := x_{m,t}$ and $T := T_m$. The trajectory $x_0, \ldots, x_T$ induces $T$ one-step prediction losses.

We first state two supporting technical ingredients and then the main martingale PAC-Bayes result. Theorem 1 identifies the instantaneous Gaussian log-loss and the associated martingale-difference structure. Theorem 2 uses this structure together with an MGF bound to construct an exponential supermartingale. Theorem 3 then applies a PAC-Bayes change-of-measure argument to obtain the predictive-risk bound that motivates the fit–KL surrogate used by PBML-LTI.

**Theorem 1 (Instantaneous Gaussian Log-Loss).** [1] *Fix $\sigma^2 > 0$ and consider the Gaussian conditional likelihood model $p_\sigma(x_t \mid x_{t-1}, A) = \mathcal{N}(Ax_{t-1}, \sigma^2 I_d)$. The associated instantaneous log-loss is*

$$\ell_t(A) := -\log p_\sigma(x_t \mid x_{t-1}, A) = \frac{1}{2\sigma^2}\|x_t - Ax_{t-1}\|_2^2 + \frac{d}{2}\log(2\pi\sigma^2), \qquad t = 1, \ldots, T. \tag{16}$$

*Let $\{\mathcal{F}_t\}_{t=0}^T$ be any filtration such that $\ell_t(A)$ is $\mathcal{F}_t$-measurable, and define*

$$d_t(A) := \mathbb{E}[\ell_t(A) \mid \mathcal{F}_{t-1}] - \ell_t(A). \tag{17}$$

*Then for every fixed $A$, $\{d_t(A)\}_{t=1}^T$ is a martingale difference sequence: $\mathbb{E}[d_t(A) \mid \mathcal{F}_{t-1}] = 0$ for all $t$.*

**Theorem 2 (Exponential Supermartingale).** [2] *Suppose that for all $\lambda_{\mathrm{PB}} \in (0, 1]$ and all $t = 1, \ldots, T$,*

$$\mathbb{E}\big[\exp\big(\lambda_{\mathrm{PB}} d_t(A)\big) \mid \mathcal{F}_{t-1}\big] \leq \exp\big(\psi_t(\lambda_{\mathrm{PB}})\big). \tag{18}$$

*Let $S_t(A) := \sum_{i=1}^t d_i(A)$ and define*

$$Z_t(A; \lambda_{\mathrm{PB}}) := \exp\left(\lambda_{\mathrm{PB}} S_t(A) - \sum_{i=1}^t \psi_i(\lambda_{\mathrm{PB}})\right), \qquad t = 0, 1, \ldots, T, \tag{19}$$

*with $Z_0(A; \lambda_{\mathrm{PB}}) = 1$. Then $\{Z_t(A; \lambda_{\mathrm{PB}})\}_{t=0}^T$ is a nonnegative supermartingale with respect to $\{\mathcal{F}_t\}_{t=0}^T$.*

We continue to write $\ell_t(\cdot) := \ell_{m,t}(\cdot)$ and $\mathcal{F}_t := \mathcal{F}_{m,t}$.

**Theorem 3 (Martingale PAC-Bayes Bound).** [3] *Let $P$ be any prior distribution over $A$ independent of the trajectory, and let $Q$ be any posterior distribution over $A$. Under the setting of Theorems 1 and 2, for any $\delta \in (0, 1)$ and any $\lambda_{\mathrm{PB}} \in (0, 1]$, with probability at least $1 - \delta$,*

$$\frac{1}{T}\sum_{t=1}^T \mathbb{E}_{A\sim Q}[\mathbb{E}[\ell_t(A) \mid \mathcal{F}_{t-1}]] \leq \frac{1}{T}\sum_{t=1}^T \mathbb{E}_{A\sim Q}[\ell_t(A)] + \frac{1}{\lambda_{\mathrm{PB}}T}\Big(D_{\mathrm{KL}}(Q\|P) + \log\tfrac{1}{\delta} + \sum_{t=1}^T \psi_t(\lambda_{\mathrm{PB}})\Big). \tag{20}$$

Theorems 1 and 2 are auxiliary technical ingredients for the main result, Theorem 3. Specifically, Theorem 1 shows that the gap between conditional predictive loss and realized loss forms a martingale difference sequence, while Theorem 2 turns an MGF control on this sequence into an exponential supermartingale. Theorem 3 then applies a PAC-Bayes change-of-measure argument to lift this concentration statement from a fixed $A$ to a posterior distribution $Q$ over $A$, which is exactly the object produced by Bayesian adaptation.

For later reference, define the conditional and empirical trajectory risks

$$L_T(Q) := \frac{1}{T}\sum_{t=1}^T \mathbb{E}_{A\sim Q}[\mathbb{E}[\ell_t(A) \mid \mathcal{F}_{t-1}]], \qquad \widehat{L}_T(Q) := \frac{1}{T}\sum_{t=1}^T \mathbb{E}_{A\sim Q}[\ell_t(A)]. \tag{21}$$

Thus, Theorem 3 gives a generic predictive-risk statement for dependent trajectories: with high probability, the conditional predictive risk is controlled by empirical in-trajectory fit, a KL-to-prior complexity term, and a martingale concentration penalty.

---

[1] The proof is included in Appendix A.1.
[2] The proof is included in Appendix A.2.
[3] The proof is included in Appendix A.3.

In our hierarchical model, we take $P = P_\phi$ to be the learned matrix-normal prior equation 9, and we restrict $Q$ to be the conjugate posterior $Q_{m,\phi}$ in equation 11. At this stage, however, the result is still a generic predictive-risk statement. In the next subsection, we use this structure to motivate a practical fit–KL surrogate for meta-training. Later, in Section 4.5, we return to this result and specialize it to the support–query setting used in few-shot evaluation.

### 4.3 Meta-Training Objective

The martingale PAC-Bayes bound in Theorem 3 contains the term $\sum_{t=1}^{T} \psi_t(\lambda_{\mathrm{PB}})$, which captures the conditional fluctuation scale of the centered loss increments $d_t(A)$. Under a uniform conditional MGF control, this term becomes explicit and independent of the meta-parameters that define the prior, so it influences the bound only through constants and through the global weighting of the KL term.

**Assumption 4** (**Uniform moment generating function (MGF) control**). *Recall $d_t(A) := \mathbb{E}[\ell_t(A) \mid \mathcal{F}_{t-1}] - \ell_t(A)$. We assume there exists a constant $v > 0$ such that, for all $t \in \{1, \ldots, T\}$, all $A$, and all $\lambda_{\mathrm{PB}} \in (0, 1]$,*

$$\log \mathbb{E}\big[\exp\big(\lambda_{\mathrm{PB}} d_t(A)\big) \,\big|\, \mathcal{F}_{t-1}\big] \le \frac{\lambda_{\mathrm{PB}}^2 v}{2}. \tag{22}$$

Equivalently, equation 18 holds with $\psi_t(\lambda_{\mathrm{PB}}) = \lambda_{\mathrm{PB}}^2 v/2$ for $\lambda_{\mathrm{PB}} \in (0, 1]$, so that

$$\sum_{t=1}^{T} \psi_t(\lambda_{\mathrm{PB}}) \le \frac{\lambda_{\mathrm{PB}}^2 v T}{2}.$$

This is the range of $\lambda_{\mathrm{PB}}$ used in the PAC-Bayes theorem and corollaries below. The constant $v$ is a valid conditional MGF variance proxy: choosing a larger valid $v$ preserves the inequality but loosens the bound, whereas choosing $v$ too small may invalidate the MGF condition.

**Self-normalization and bound tightness.** The quantity $vT$ can be interpreted as a uniform predictable fluctuation scale for the centered martingale loss process $d_t(A)$. A sharper fully self-normalized analysis could replace this uniform proxy by a predictable quadratic-variation or data-dependent variance term. We use the simpler uniform formulation to keep the PAC-Bayes statement transparent and directly compatible with the fit–KL surrogate. This also clarifies when the bound is expected to be tight: it is tighter when trajectories remain controlled, posterior mass concentrates on predictors with small one-step residuals, and conditional loss fluctuations are small. Conversely, if state norms or prediction errors grow, as in unstable systems, the predictable variance scale becomes large and any valid uniform choice of $v$ necessarily yields a looser bound.

Under Assumption 4, Theorem 3 yields a generic predictive-risk bound of the form

$$L_T(Q) \le \widehat{L}_T(Q) + \frac{1}{\lambda_{\mathrm{PB}} T} D_{\mathrm{KL}}(Q\|P) + \text{terms depending only on } (\delta, \lambda_{\mathrm{PB}}, v, T), \qquad \lambda_{\mathrm{PB}} \in (0, 1]. \tag{23}$$

This result is *not* used as the exact optimization objective in training. Instead, it provides the structural motivation for the surrogate adopted by PBML-LTI. For fixed $(\delta, \lambda_{\mathrm{PB}}, v)$ and $T$, the dependence on the meta-parameters $\phi$ enters through the empirical fit $\widehat{L}_T(Q)$ and the KL term $D_{\mathrm{KL}}(Q\|P)$, suggesting a fit–KL tradeoff as the core training criterion.

In the main implementation, we use the canonical choice corresponding to $\lambda_{\mathrm{PB}} = 1$, which yields the standard fit–KL tradeoff, preserves the usual conjugate Bayesian update, and avoids introducing an additional bound-specific tuning parameter in meta-training. In the experiments, we also study an implementation-level training temperature $\lambda_{\mathrm{tr}} > 0$ that rescales the fit–KL tradeoff. This empirical temperature should be distinguished from the PAC-Bayes parameter $\lambda_{\mathrm{PB}}$: the formal bound is stated only for $\lambda_{\mathrm{PB}} \in (0, 1]$, while values $\lambda_{\mathrm{tr}} > 1$ are included only as practical stress tests of more data-driven adaptation. We report this sensitivity study in Section 6.1. For each training task $m \in [M_{\mathrm{tr}}]$, we observe a trajectory $D_m = \{x_{m,t}\}_{t=0}^{T_m}$ and its regression matrices $(X_m, Y_m)$ defined in equation 2. Given the current meta-parameters $\phi = (W, V, \sigma^2)$, we form the conjugate posterior $Q_{m,\phi}$ in equation 11 with parameters $(M_m, V_m)$ from equation 12 and

equation 13. We then define the per-task empirical fit term

$$\mathcal{L}_{\text{fit},m}(\phi) := \frac{1}{T_m}\mathbb{E}_{A\sim Q_{m,\phi}}\left[\sum_{t=1}^{T_m}\ell_{m,t}(A)\right] = \frac{1}{2\sigma^2 T_m}\left[\|Y_m - M_m X_m\|_F^2 + d\,\text{tr}(V_m X_m X_m^\top)\right] + \frac{d}{2}\log(2\pi\sigma^2), \quad (24)$$

where the equality follows from equation 14. Likewise, the per-task complexity term is

$$\mathcal{L}_{\text{kl},m}(\phi) := \frac{1}{T_m}D_{\text{KL}}(Q_{m,\phi}\,\|\,P_\phi) = \frac{1}{2T_m}\Big[d\Big(\text{tr}(V^{-1}V_m) - \log\det(V^{-1}V_m) - d\Big) \\ + \text{tr}\Big((M_m - W)V^{-1}(M_m - W)^\top\Big)\Big]. \quad (25)$$

Motivated by the PAC-Bayes structure above, we optimize the following surrogate:

$$\min_\phi \ \frac{1}{M_{\text{tr}}}\sum_{m=1}^{M_{\text{tr}}}\left[\mathcal{L}_{\text{fit},m}(\phi) + \mathcal{L}_{\text{kl},m}(\phi)\right] \ + \ \mathcal{R}_{\text{hyper}}(W,V) \ + \ \mathcal{R}_{\text{stab}}(W;\rho_0), \quad (26)$$

where the additional regularizers are practical optimization and prior-shaping terms rather than components of the formal PAC-Bayes theorem:

$$\mathcal{R}_{\text{hyper}}(W,V) := \frac{1}{2\tau_W^2}\|W\|_F^2 + \lambda_V\left(\frac{1}{2}\|V - I_d\|_F^2 - \log\det(V)\right), \quad (27)$$

and

$$\mathcal{R}_{\text{stab}}(W;\rho_0) := \big(\max\{0, \rho(W) - \rho_0\}\big)^2. \quad (28)$$

Here $\tau_W > 0$ controls shrinkage on $W$, $\lambda_V \geq 0$ encourages a well-scaled and well-conditioned $V$, and $\rho_0 \in (0,1)$ is a target stability margin for the shared mean dynamics. The stability regularizer should not be interpreted as directly enforcing Assumption 1, which is a data-generating condition on the true task matrices $A_m$. Instead, it acts on the learned shared prior mean and encourages the prior to concentrate near dynamically well-behaved transition matrices, which can indirectly improve posterior adaptation and numerical stability.

To ensure $V$ remains symmetric positive definite during optimization, we parameterize

$$V = LL^\top + \epsilon I_d, \quad (29)$$

where $L$ is learnable lower-triangular and $\epsilon > 0$ is a small diagonal shift.

With the objective equation 26 in place, meta-training reduces to repeatedly (i) computing conjugate posteriors for minibatches of tasks in closed form and (ii) updating $\phi$ through stochastic gradients of the resulting fit–KL surrogate.

### 4.4  Task-Level Inference and Adaptation

After meta-training, we perform task-level inference on previously unseen tasks by adapting the learned prior to a short support trajectory. For each new task $m$ with trajectory length $T_m$, we first select a support prefix of length $S_m = \min\{T_{\text{sup}}, T_m\}$ and form the corresponding regression matrices $X_m^{\text{sup}}$ and $Y_m^{\text{sup}}$ from the observed state transitions.

Given the learned meta-parameters $\phi = (W, V, \sigma^2)$, adaptation is carried out via a closed-form Bayesian update. Specifically, the matrix-normal prior is combined with the support data to produce a task-specific posterior

$$Q_{m,\phi} = \mathcal{MN}(M_m, I_d, V_m),$$

where the posterior mean $M_m$ and covariance $V_m$ are given by equation 12 and equation 13. We use the posterior mean $\widehat{A}_m := M_m$ as a point estimate of the task dynamics, while the posterior covariance quantifies task-specific uncertainty and governs the strength of shrinkage toward the meta-prior.

---

**Algorithm 1** PBML-LTI Meta-Training and Task Adaptation

---

    **Meta-training**

**Require:** Training tasks $\mathcal{T}_{\text{train}}$; steps $S$; batch size $B$.

 1: **for** $s = 1, \ldots, S$ **do**
 2:     Sample minibatch $\mathcal{B} \subset \mathcal{T}_{\text{train}}$ with $|\mathcal{B}| = \min(B, |\mathcal{T}_{\text{train}}|)$.
 3:     **for** each task $m \in \mathcal{B}$ **do**
 4:         Form $(X_m, Y_m)$ from the trajectory as in equation 2.
 5:         Compute posterior parameters $(M_m, V_m)$ via equation 12 and equation 13.
 6:         Evaluate $\mathcal{L}_{\text{fit},m}$ and $\mathcal{L}_{\text{kl},m}$ via equation 24 and equation 25.
 7:     **end for**
 8:     $L_{\text{tr}} \leftarrow \frac{1}{|\mathcal{B}|} \sum_{m \in \mathcal{B}} \left( \mathcal{L}_{\text{fit},m} + \mathcal{L}_{\text{kl},m} \right) + \mathcal{R}_{\text{hyper}}(W, V) + \mathcal{R}_{\text{stab}}(W)$.
 9:     Adam update of $(W, V, \sigma^2)$ using $\nabla L_{\text{tr}}$.
10: **end for**
11: Save learned $\phi = (W, V, \sigma^2)$.
    **Task adaptation on new tasks**

**Require:** New tasks $\mathcal{T}_{\text{test}}$; support horizon $T_{\text{sup}}$; (optional) query horizon $T_{\text{qry}}$.

12: **for** each task $m \in \mathcal{T}_{\text{test}}$ **do**
13:     Choose $S_m \leftarrow \min\{T_{\text{sup}}, T_m\}$.
14:     Form $(X_m^{\text{sup}}, Y_m^{\text{sup}})$ from the first $S_m$ transitions.
15:     Compute $(M_m, V_m)$ using $(X_m, Y_m) = (X_m^{\text{sup}}, Y_m^{\text{sup}})$ via equation 12 and equation 13.
16:     Set point estimate $\widehat{A}_m \leftarrow M_m$.
17:     Roll out over $K_m = \min\{T_{\text{qry}}, T_m - S_m\}$ using $\widehat{x}_{t+1} = \widehat{A}_m \widehat{x}_t$.
18: **end for**
19: **return** $\{\widehat{A}_m\}_{m \in \mathcal{T}_{\text{test}}}$ and rollout predictions $\{\widehat{x}_{m,t}\}$.

---

The posterior mean is used for the reported point-estimate metrics because both $E_A$ and $E_{\text{traj}}$ are defined for a single transition matrix and a single deterministic open-loop rollout. Under squared-error loss, $M_m$ is the natural Bayes point estimate. The posterior distribution itself is not discarded: the covariance $V_m$ enters the posterior-expected fit term in the meta-training objective and provides task-specific uncertainty information. In applications where uncertainty-aware prediction is desired, one can sample $A^{(s)} \sim Q_{m,\phi}$ and propagate each sampled transition matrix to obtain posterior predictive rollout bands or credible regions for entries of the transition matrix.

We summarize the overall procedure in Algorithm 1. In particular, meta-training optimizes the fit–KL surrogate over training tasks, whereas test-time adaptation consists only of the closed-form posterior update on the support prefix followed by prediction or rollout using the posterior mean.

### 4.5 Theoretical Analysis

The previous subsection introduced the fit–KL surrogate used for meta-training. We now make its theoretical origin more explicit in three steps. First, we state the generic martingale PAC-Bayes predictive-risk result under a sub-Gaussian MGF condition. Second, we specialize this result to the support–query protocol used in few-shot adaptation. Third, we record problem-specific consequences of this support–query predictive bound for transition-matrix recovery and open-loop rollout error. Thus, the theory should be interpreted as providing a PAC-Bayes justification for the fit–KL surrogate and for the support–query predictive-risk perspective, rather than as a direct statement of the exact training objective optimized in practice.

#### 4.5.1 From Martingale PAC-Bayes to Support–Query Prediction

**Corollary 1 (Sub-Gaussian martingale PAC-Bayes excess risk).** [4] *Under Assumption 4, for any prior $P$ independent of the trajectory and any (possibly data-dependent) posterior $Q$, for any $\delta \in (0, 1)$ and*

---

[4]The proof is included in Appendix A.4.

*any* $\lambda_{\mathrm{PB}} \in (0, 1]$, *with probability at least* $1 - \delta$,

$$L_T(Q) \le \widehat{L}_T(Q) + \frac{D_{\mathrm{KL}}(Q\|P) + \log(1/\delta)}{\lambda_{\mathrm{PB}} T} + \frac{\lambda_{\mathrm{PB}} v}{2}. \tag{30}$$

*Optimizing the right-hand side over* $\lambda_{\mathrm{PB}} \in (0, 1]$ *gives*

$$\lambda_{\mathrm{PB}}^\star = \min\left\{1, \ \sqrt{\frac{2\big(D_{\mathrm{KL}}(Q\|P) + \log(1/\delta)\big)}{vT}}\right\}. \tag{31}$$

*In the regime where* $\lambda_{\mathrm{PB}}^\star < 1$, *this simplifies to*

$$L_T(Q) \le \widehat{L}_T(Q) + \sqrt{\frac{2v\big(D_{\mathrm{KL}}(Q\|P) + \log(1/\delta)\big)}{T}}. \tag{32}$$

Corollary 1 shows that the excess-risk term decays at the canonical rate $O(T^{-1/2})$ under sub-Gaussian martingale concentration. It also makes explicit the role of meta-learning: for a fixed task posterior $Q$, a better prior $P$ reduces the divergence $D_{\mathrm{KL}}(Q\|P)$ and therefore tightens the gap between empirical in-trajectory fit and conditional predictive risk. This provides the generic PAC-Bayes structure that motivates the fit–KL objective used by PBML-LTI.

**Specialization to the support–query protocol.** The result above is stated for a generic trajectory. To connect it directly to few-shot adaptation, we now specialize it to the support–query protocol used in our experiments. Fix a task $m$ with trajectory $\{x_{m,t}\}_{t=0}^{T_m}$. Let the support prefix be

$$D_m^{\mathrm{sup}} := \{x_{m,t}\}_{t=0}^{S_m},$$

and form the task-specific posterior using only this prefix:

$$Q_{m,\phi}^{\mathrm{sup}} := p(A_m \mid D_m^{\mathrm{sup}}, \phi).$$

Conditioning on the support filtration $\mathcal{F}_{m,S_m}$ fixes the posterior. The remaining query suffix can then be analyzed using the shifted filtration

$$\mathcal{G}_u := \mathcal{F}_{m,S_m+u}, \qquad u = 0, \ldots, K_m,$$

where $K_m \le T_m - S_m$ is the query horizon.

Define the query regression matrices

$$X_{m,q} := [x_{m,S_m}, \ldots, x_{m,S_m+K_m-1}] \in \mathbb{R}^{d \times K_m}, \qquad Y_{m,q} := [x_{m,S_m+1}, \ldots, x_{m,S_m+K_m}] \in \mathbb{R}^{d \times K_m}. \tag{33}$$

We define the conditional support–query predictive risk of a posterior $Q$ by

$$R_{m,q}^{\mathrm{pred}}(Q) := \frac{1}{K_m} \sum_{u=1}^{K_m} \mathbb{E}_{A \sim Q}\big[\mathbb{E}\big[\|x_{m,S_m+u} - Ax_{m,S_m+u-1}\|_2^2 \,\big|\, \mathcal{G}_{u-1}\big]\big], \tag{34}$$

with empirical counterpart

$$\widehat{R}_{m,q}^{\mathrm{pred}}(Q) := \frac{1}{K_m} \mathbb{E}_{A \sim Q}\big[\|Y_{m,q} - AX_{m,q}\|_F^2\big]. \tag{35}$$

These quantities are nonnegative and correspond directly to prediction on future query time steps after adaptation.

**Corollary 2 (Support–query martingale PAC-Bayes bound).** [5] *Under Assumption 4, for any* $\delta \in (0, 1)$ *and any* $\lambda_{\mathrm{PB}} \in (0, 1]$, *with probability at least* $1 - \delta$,

$$R_{m,q}^{\mathrm{pred}}(Q_{m,\phi}^{\mathrm{sup}}) \le \widehat{R}_{m,q}^{\mathrm{pred}}(Q_{m,\phi}^{\mathrm{sup}}) + \frac{2\sigma^2}{\lambda_{\mathrm{PB}} K_m}\Big(D_{\mathrm{KL}}(Q_{m,\phi}^{\mathrm{sup}}\|P_\phi) + \log(1/\delta)\Big) + \sigma^2 \lambda_{\mathrm{PB}} v. \tag{36}$$

---

[5]The proof is included in Appendix A.5.

This result is obtained by applying Corollary 1 to the shifted query sequence $\{\ell_{m,S_m+u}(A)\}_{u=1}^{K_m}$ under the filtration $\{\mathcal{G}_u\}_{u=0}^{K_m}$, and then removing the constant term in the Gaussian log-loss and rescaling both sides by $2\sigma^2$. Consequently, the left-hand side becomes the support-conditioned predictive risk on the held-out query suffix, while the right-hand side contains the empirical query fit, the posterior-to-prior complexity term, and the martingale concentration penalty.

Importantly, equation 36 is not a support-only certificate whose right-hand side is computable before observing the query suffix. The term $\widehat{R}_{m,q}^{\mathrm{pred}}(Q_{m,\phi}^{\mathrm{sup}})$ is evaluated on the held-out query trajectory. The result should therefore be interpreted as a support-conditioned support–query PAC-Bayes diagnostic: it is informative when the posterior adapted from the support prefix also fits the held-out query suffix. If the empirical query fit is large, this term can dominate the bound and the derived transition-matrix and rollout consequences can become loose.

### 4.5.2 Support Fit Transfer to Query Fit Analysis

We now make explicit sufficient conditions under which the empirical query term is controlled by quantities associated with the support-adapted posterior. Let the support regression pair used to form $Q_{m,\phi}^{\mathrm{sup}} = \mathcal{MN}(M_m, I_d, V_m)$ be denoted by $(X_m^{\mathrm{sup}}, Y_m^{\mathrm{sup}})$, with $S_m$ transitions, and define

$$\widehat{\Sigma}_{\mathrm{sup}} := \frac{1}{S_m} X_m^{\mathrm{sup}}(X_m^{\mathrm{sup}})^\top, \qquad \widehat{\Sigma}_q := \frac{1}{K_m} X_{m,q} X_{m,q}^\top,$$

together with the stacked support and query process-noise matrices

$$\Xi_m^{\mathrm{sup}} := [w_{m,1}, \ldots, w_{m,S_m}], \qquad \Xi_{m,q} := [w_{m,S_m+1}, \ldots, w_{m,S_m+K_m}].$$

Also define the support empirical posterior-predictive risk

$$\widehat{R}_{\mathrm{sup}}^{\mathrm{pred}} := \frac{1}{S_m} \mathbb{E}_{A \sim Q_{m,\phi}^{\mathrm{sup}}} \left[ \|Y_m^{\mathrm{sup}} - A X_m^{\mathrm{sup}}\|_F^2 \right]. \tag{37}$$

By equation 14, this is equal to

$$\widehat{R}_{\mathrm{sup}}^{\mathrm{pred}} = \frac{1}{S_m} \|Y_m^{\mathrm{sup}} - M_m X_m^{\mathrm{sup}}\|_F^2 + d\operatorname{tr}(V_m \widehat{\Sigma}_{\mathrm{sup}}).$$

Define the support-to-query excitation-alignment constant

$$\gamma_m := \inf\{\gamma > 0 : \widehat{\Sigma}_q \preceq \gamma \widehat{\Sigma}_{\mathrm{sup}}\}. \tag{38}$$

When finite, this constant measures how much the query suffix excites directions already excited by the support prefix. Equivalently, $\gamma_m < \infty$ only when $\operatorname{range}(\widehat{\Sigma}_q) \subseteq \operatorname{range}(\widehat{\Sigma}_{\mathrm{sup}})$, in which case

$$\gamma_m = \lambda_{\max}\left(\widehat{\Sigma}_{\mathrm{sup}}^{\dagger/2} \widehat{\Sigma}_q \widehat{\Sigma}_{\mathrm{sup}}^{\dagger/2}\right).$$

Thus, $\gamma_m = O(1)$ formalizes the condition that the query directions are sufficiently aligned with the support directions.

**Proposition 1** (Support-to-query transfer under excitation alignment). *Assume $\gamma_m < \infty$. Then, pathwise,*

$$\widehat{R}_{m,q}^{\mathrm{pred}}(Q_{m,\phi}^{\mathrm{sup}}) \leq 4\gamma_m \widehat{R}_{\mathrm{sup}}^{\mathrm{pred}} + \frac{4\gamma_m}{S_m}\|\Xi_m^{\mathrm{sup}}\|_F^2 + \frac{2}{K_m}\|\Xi_{m,q}\|_F^2, \tag{39}$$

*where $\Xi_m^{\mathrm{sup}}$ and $\Xi_{m,q}$ are the stacked support and query process-noise matrices. Moreover, the posterior-width contribution satisfies*

$$d\operatorname{tr}(V_m \widehat{\Sigma}_q) \leq \gamma_m d \min\left\{\frac{\sigma^2 d}{S_m}, \operatorname{tr}(V \widehat{\Sigma}_{\mathrm{sup}})\right\}. \tag{40}$$

Proposition 1 shows that, under support–query excitation alignment, the held-out query empirical term is bounded by the support predictive risk plus pure noise energies. The factor $\gamma_m$ is the price of transferring support fit to query fit under temporal dependence. This is the dependent-data analogue of the role played by exchangeability in i.i.d. train–test arguments. The refined cap equation 40 also makes the meta-learning mechanism visible: when $S_m$ is small, the posterior-width contribution is governed by $\mathrm{tr}(V\widehat{\Sigma}_{\mathrm{sup}})$, that is, by the tightness of the *learned* prior covariance along the realized support excitation, and once the support is well excited it decays at the parametric rate $\sigma^2 d^2/S_m$.

### 4.5.3 Regularized Transfer and Few-Shot Limitations

However, in the strict few-shot regime $S_m < d$, the support Gram matrix is rank deficient; if the process noise is nondegenerate, the very first query state already exits range($\widehat{\Sigma}_{\mathrm{sup}}$), so that $\gamma_m = \infty$ almost surely. This is not merely a proof artifact: the support trajectory cannot identify task-specific directions it does not excite. We therefore also use a regularized alignment constant

$$\gamma_{m,\alpha} := \lambda_{\max}\left[(\widehat{\Sigma}_{\mathrm{sup}} + \alpha I_d)^{-1/2}\widehat{\Sigma}_q(\widehat{\Sigma}_{\mathrm{sup}} + \alpha I_d)^{-1/2}\right], \qquad \alpha > 0, \tag{41}$$

which is always finite.

**Proposition 2** (Regularized support-to-query transfer)**.** *For any $\alpha > 0$,*

$$\widehat{R}_{m,q}^{\mathrm{pred}}(Q_{m,\phi}^{\mathrm{sup}}) \leq 4\gamma_{m,\alpha}\,\widehat{R}_{\mathrm{sup}}^{\mathrm{pred}} + \frac{4\gamma_{m,\alpha}}{S_m}\|\Xi_m^{\mathrm{sup}}\|_F^2 + \frac{2}{K_m}\|\Xi_{m,q}\|_F^2$$
$$+ 2\gamma_{m,\alpha}\alpha\,\mathbb{E}_{A \sim Q_{m,\phi}^{\mathrm{sup}}}\|A - A_m\|_F^2. \tag{42}$$

*Moreover,*

$$\mathbb{E}_{A \sim Q_{m,\phi}^{\mathrm{sup}}}\|A - A_m\|_F^2 = \|M_m - A_m\|_F^2 + d\,\mathrm{tr}(V_m).$$

*Moreover, $\gamma_{m,\alpha} \leq \min\{\gamma_m,\,\lambda_{\max}(\widehat{\Sigma}_q)/\alpha\}$, and the map $\alpha \mapsto \gamma_{m,\alpha}$ is nonincreasing.*

The final term in equation 42 isolates the weakly excited directions. In those directions, transfer cannot come from the support data alone; it must come from posterior concentration and learned-prior quality. Indeed, by equation 13, the mean-error part obeys the closed-form identity $M_m - A_m = \left[\frac{1}{\sigma^2}\Xi_m^{\mathrm{sup}}(X_m^{\mathrm{sup}})^\top + (W - A_m)V^{-1}\right]V_m$, so it is controlled by support noise concentration together with the learned-prior quality $\Delta_m(\phi)$ defined in equation 44; this control is quantified in Appendix A.6. Thus, the regularized transfer result is the relevant statement in the high-dimensional few-shot regime. A prior-matched regularization choice replaces $\alpha I_d$ by $(\sigma^2/S_m)V^{-1}$, in which case the corresponding posterior-scaled alignment diagnostic is

$$\widetilde{\gamma}_m = \frac{S_m}{\sigma^2}\lambda_{\max}\left(V_m^{1/2}\widehat{\Sigma}_q V_m^{1/2}\right),$$

and replacing $\widehat{\Sigma}_q$ by the validation Gram gives the support-window proxy used in the empirical diagnostics of Appendix B.1. With the prior-matched choice, the regularized Gram becomes $\widehat{\Sigma}_{\mathrm{sup}} + (\sigma^2/S_m)V^{-1} = (\sigma^2/S_m)V_m^{-1}$ by equation 12, so $\widetilde{\gamma}_m$ is computable in closed form from the fitted posterior and the query Gram, and the posterior-width contribution obeys the unconditional bound $d\,\mathrm{tr}(V_m\widehat{\Sigma}_q) \leq \widetilde{\gamma}_m\,\sigma^2 d^2/S_m$.

### 4.5.4 A Conditional Support-to-Query Certificate

**Lemma 1** (Noise-energy concentration)**.** *Under Assumption 2, there is an absolute constant $c > 0$ such that, for a segment of length $n$ and any $\delta \in (0, 1)$, with probability at least $1 - \delta$,*

$$\frac{1}{n}\sum_{t=1}^{n}\|w_{m,t}\|_2^2 \leq \mathrm{tr}(\Sigma_w) + c\sigma_w^2 d\left(\sqrt{\frac{\log(1/\delta)}{n}} + \frac{\log(1/\delta)}{n}\right).$$

The leading dimension factor $d$ in the deviation term of Lemma 1 improves to $\sqrt{d}$ when the noise coordinates are conditionally independent, as for the Gaussian noise generator equation 63 used in the synthetic experiments; we state the general conditionally sub-Gaussian version for consistency with Assumption 2.

Combining Proposition 1, Lemma 1, and Corollary 2 gives the following conditional support-to-query certificate.

**Corollary 3** (Conditional support-to-query certificate). *Fix $\bar{\gamma} > 0$, $\lambda_{\mathrm{PB}} \in (0, 1]$, and $\delta \in (0, 1)$. On the event $\widehat{\Sigma}_q \preceq \bar{\gamma} \widehat{\Sigma}_{\mathrm{sup}}$, with probability at least $1 - 3\delta$,*

$$R_{m,q}^{\mathrm{pred}}(Q_{m,\phi}^{\mathrm{sup}}) \leq 4\bar{\gamma}\,\widehat{R}_{\mathrm{sup}}^{\mathrm{pred}} + (4\bar{\gamma} + 2)\mathrm{tr}(\Sigma_w) + 4\bar{\gamma}\,\epsilon_{S_m}(\delta) + 2\,\epsilon_{K_m}(\delta)$$

$$+ \frac{2\sigma^2}{\lambda_{\mathrm{PB}}K_m}\left(D_{\mathrm{KL}}(Q_{m,\phi}^{\mathrm{sup}}\|P_\phi) + \log(1/\delta)\right) + \sigma^2\lambda_{\mathrm{PB}}v, \tag{43}$$

*where*

$$\epsilon_n(\delta) = c\sigma_w^2 d\left(\sqrt{\frac{\log(1/\delta)}{n}} + \frac{\log(1/\delta)}{n}\right).$$

Corollary 3 is a conditional certificate: apart from the alignment level and noise-floor constants, the right-hand side is determined by the support-adapted posterior and the support prefix. It is not an unconditional support-only guarantee. Such an unconditional guarantee cannot hold in general when the support does not excite the directions later encountered in the query suffix. In that case, the regularized bound equation 42 identifies the remaining term that must be controlled by learned-prior quality. If the effective fit prefix is selected by inner validation from a grid of $G$ candidate prefixes, as in the adaptive-support evaluation protocol, the certificate holds uniformly over the grid with $\delta$ replaced by $\delta/G$, via a union bound over the candidates. A regularized variant of Corollary 3 replaces the event $\widehat{\Sigma}_q \preceq \bar{\gamma}\widehat{\Sigma}_{\mathrm{sup}}$ by $\widehat{\Sigma}_q \preceq \bar{\gamma}(\widehat{\Sigma}_{\mathrm{sup}} + \alpha I_d)$—an event that holds deterministically for $\bar{\gamma} = \gamma_{m,\alpha}$—at the price of the additional term $2\bar{\gamma}\alpha\,\mathbb{E}_{A\sim Q_{m,\phi}^{\mathrm{sup}}}\|A - A_m\|_F^2$ from Proposition 2; this term involves the true $A_m$ and is a prior-quality quantity rather than a support-computable one.

**Remark 1** (Unavoidable support–query linkage assumption). The conditional form of Corollary 3 reflects a genuine information-theoretic obstruction rather than a proof artifact. Any two transition matrices that agree on $\mathrm{range}(\widehat{\Sigma}_{\mathrm{sup}})$ induce the same support law, so no bound on $\widehat{R}_{m,q}^{\mathrm{pred}}$ that is uniform over $A_m$ can be certified from the support alone once the query excites $\ker(\widehat{\Sigma}_{\mathrm{sup}})$. The value $\gamma_m = \infty$ in equation 38 flags exactly this situation, and the term $2\gamma_{m,\alpha}\alpha\,\mathbb{E}_{A\sim Q_{m,\phi}^{\mathrm{sup}}}\|A - A_m\|_F^2$ in equation 42 prices it. On the unexcited subspace the posterior coincides with the prior, so control there is an *environment-level* guarantee about the learned $(W, V)$—precisely the second-stage meta-generalization question that Section 6.2 and the Conclusion identify as future work. The analysis thus makes the division of labor explicit: excitation alignment and the support predictive risk $\widehat{R}_{\mathrm{sup}}^{\mathrm{pred}}$ govern the task-level, excited component, while prior quality governs the remainder.

### 4.5.5 Stable and Unstable Regimes

We next identify regimes in which the support–query transfer constants are controlled, and regimes in which they necessarily deteriorate.

The preceding results make precise the conditions under which the empirical query term is small. If the support Gram matrix is well excited and query state norms are controlled, namely

$$\lambda_{\min}(\widehat{\Sigma}_{\mathrm{sup}}) \geq \kappa_{\mathrm{sup}} > 0, \qquad \max_{0 \leq u < K_m} \|x_{m,S_m+u}\|_2^2 \leq B_x^2,$$

then

$$\gamma_m \leq \frac{B_x^2}{\kappa_{\mathrm{sup}}}.$$

Indeed, $xx^\top \preceq \|x\|_2^2 I_d$ for every $x \in \mathbb{R}^d$, hence $\widehat{\Sigma}_q \preceq B_x^2 I_d \preceq (B_x^2/\kappa_{\mathrm{sup}})\widehat{\Sigma}_{\mathrm{sup}}$. Thus, controlled states and support excitation imply controlled support-to-query transfer. In a common stationary stable regime, where support and query empirical Grams concentrate around the same stationary covariance, $\gamma_m = O(1)$ with high probability. Concretely, suppose $\rho(A_m) \leq \bar{\rho} < 1$ and $\Sigma_w \succ 0$, and let $\Gamma_\infty := \sum_{k \geq 0} A_m^k \Sigma_w (A_m^k)^\top$, which is finite and positive definite. If

$$\frac{1}{2}\Gamma_\infty \preceq \widehat{\Sigma}_{\mathrm{sup}}, \qquad \widehat{\Sigma}_q \preceq \frac{3}{2}\Gamma_\infty,$$

then $\gamma_m \leq 3$. Two-sided concentration of dependent empirical Gram matrices around $\Gamma_\infty$ at this accuracy holds with high probability once $S_m$ and $K_m$ exceed a polynomial burn-in, by now-standard arguments for stable linear systems (Abbasi-Yadkori et al., 2011; Simchowitz et al., 2018; Sarkar & Rakhlin, 2019). Since $\gamma_{m,\alpha} \leq \gamma_m$, both conditions bound the regularized constant as well.

The same conditions also make explicit the role of the learned prior. Since $M_m$ minimizes the conjugate ridge-form objective

$$A \mapsto \|Y_m^{\text{sup}} - A X_m^{\text{sup}}\|_F^2 + \sigma^2 \operatorname{tr}\big((A - W)V^{-1}(A - W)^\top\big),$$

comparison with $A = A_m$ yields

$$\frac{1}{S_m}\|Y_m^{\text{sup}} - M_m X_m^{\text{sup}}\|_F^2 \leq \frac{1}{S_m}\|\Xi_m^{\text{sup}}\|_F^2 + \frac{\sigma^2}{S_m}\Delta_m(\phi), \qquad \Delta_m(\phi) := \operatorname{tr}\big((A_m - W)V^{-1}(A_m - W)^\top\big). \quad (44)$$

Thus, the support fit is small when the support noise energy is moderate and the learned prior is close to the task-specific dynamics in the $V^{-1}$ geometry. Combining equation 44 with the width cap $d \operatorname{tr}(V_m \widehat{\Sigma}_{\text{sup}}) \leq \sigma^2 d^2/S_m$ from the proof of Proposition 1, Lemma 1, and Corollary 3 with $\bar{\gamma} = 3$, we obtain: with probability at least $1 - 3\delta$, on the stable common-regime event above,

$$R_{m,q}^{\text{pred}}(Q_{m,\phi}^{\text{sup}}) \leq 26 \operatorname{tr}(\Sigma_w) + \frac{12\,\sigma^2\big(\Delta_m(\phi) + d^2\big)}{S_m} + 24\,\epsilon_{S_m}(\delta) + 2\,\epsilon_{K_m}(\delta)$$

$$+ \frac{2\sigma^2}{\lambda_{\text{PB}} K_m}\Big(D_{\text{KL}}(Q_{m,\phi}^{\text{sup}}\|P_\phi) + \log(1/\delta)\Big) + \sigma^2 \lambda_{\text{PB}} v, \quad (45)$$

with constants not optimized. Term by term: the first term is the irreducible one-step noise floor (for the homoscedastic Gaussian generator equation 63 the predictive risk is bounded below by $\operatorname{tr}(\Sigma_w) = d\sigma_{\text{true}}^2$, so equation 45 is then order-optimal up to a universal constant); the second term decays at the parametric rate in $S_m$ and is small precisely when the learned prior is good, both through $\Delta_m(\phi)$ and through the KL term, whose closed form equation 15 shrinks for the same reason—this is the meta-learning effect, now explicit on the right-hand side; the remaining terms vanish as $S_m$ and $K_m$ grow. This gives an explicit sufficient condition under which the support–query bound is numerically meaningful.

We emphasize the scope: the two sufficient conditions above are deliberately stable, data-rich conditions. In the strict few-shot regime $S_m < d$ they cannot hold, and the operative statement is the regularized Proposition 2, in which the weakly excited component is controlled by the learned prior rather than by the support data.

Conversely, if $A_m$ has an unstable dominant mode, the alignment constants are provably large and the transfer factor degrades geometrically in the query horizon. The following remark records this failure mode formally.

**Remark 2** (Geometric degradation under an unstable dominant mode). Let $u \in \mathbb{R}^d$ with $\|u\|_2 = 1$ be a left eigenvector of $A_m$ with real eigenvalue $\rho > 1$, i.e. $A_m^\top u = \rho u$. Then $s_t := \langle u, x_{m,t}\rangle$ follows the scalar recursion $s_{t+1} = \rho s_t + \langle u, w_{m,t+1}\rangle$, and $\rho^{-t} s_t$ converges almost surely to a random limit that is nonzero almost surely whenever the noise has a nondegenerate component along $u$ (as for the Gaussian process noise used in the synthetic experiments). Whenever $u^\top \widehat{\Sigma}_{\text{sup}} u > 0$, the variational characterization of the positive-semidefinite order in equation 38 gives

$$\gamma_m \;\geq\; \frac{u^\top \widehat{\Sigma}_q u}{u^\top \widehat{\Sigma}_{\text{sup}} u} \;=\; \frac{S_m \sum_{r=0}^{K_m-1} s_{S_m+r}^2}{K_m \sum_{t=0}^{S_m-1} s_t^2} \;\asymp\; \rho^{2K_m} \cdot \frac{S_m}{K_m} \cdot c_{\text{traj}},$$

where $c_{\text{traj}}$ is a trajectory-dependent factor bounded away from $0$ and $\infty$ on the almost-sure growth event. The same conclusion holds for the regularized constant equation 41 at any fixed $\alpha > 0$, since $\gamma_{m,\alpha} \geq u^\top \widehat{\Sigma}_q u/\big(u^\top \widehat{\Sigma}_{\text{sup}} u + \alpha\big)$ and the numerator still grows geometrically in $K_m$. Hence the transfer factor—and with it any certificate built on it—degrades *geometrically in the query horizon* when the dominant mode is unstable, explaining why the query empirical term may dominate the bound in unstable low-dimensional settings. This complements the finite-horizon amplification that appears in the rollout consequence through $H_{m,K_m}$.

Empirical diagnostics supporting validation-window proxies for support–query comparability are reported in Appendix B.1.

### 4.5.6 Consequences for Transition Recovery and Rollout Error

In the remainder of this subsection, we write $\widehat{A}_m := M_m$ for the posterior-mean predictor produced by task adaptation and record several PAC-Bayes-derived corollaries that connect the support–query predictive bound to the empirical quantities used in our experiments. For a deterministic predictor $\widehat{A}_m$, we write $R_{m,q}^{\mathrm{pred}}(\widehat{A}_m)$ as shorthand for $R_{m,q}^{\mathrm{pred}}(\delta_{\widehat{A}_m})$.

**Projected transition-matrix error bound.** Under the true query dynamics

$$Y_{m,q} = A_m X_{m,q} + \Xi_{m,q},$$

and since the squared Frobenius loss is convex in $A$, Jensen's inequality implies

$$\frac{1}{K_m}\|Y_{m,q} - M_m X_{m,q}\|_F^2 \le \widehat{R}_{m,q}^{\mathrm{pred}}(Q_{m,\phi}^{\mathrm{sup}}). \tag{46}$$

Taking conditional expectation under the true dynamics, the residual at the posterior mean decomposes into a projected transition-matrix term plus a nonnegative noise floor. Combining this with equation 46 yields

$$E_{A,\mathrm{proj}}(m) := \frac{1}{K_m}\|(\widehat{A}_m - A_m)X_{m,q}\|_F^2 \le R_{m,q}^{\mathrm{pred}}(\widehat{A}_m) \le R_{m,q}^{\mathrm{pred}}(Q_{m,\phi}^{\mathrm{sup}}) \le \mathcal{B}_m^{\mathrm{pred}}, \tag{47}$$

where $\mathcal{B}_m^{\mathrm{pred}}$ denotes the right-hand side of equation 36. This quantity is always well defined and measures transition-matrix error along the query excitation directions.

**Full Frobenius transition-matrix bound under excitation.** If the query Gram matrix satisfies the excitation condition

$$\lambda_{\min}\left(\frac{1}{K_m}X_{m,q}X_{m,q}^\top\right) \ge \kappa_m > 0, \tag{48}$$

then the projected bound lifts to a full Frobenius bound:

$$E_A(m) = \|\widehat{A}_m - A_m\|_F^2 \le \frac{\mathcal{B}_m^{\mathrm{pred}}}{\kappa_m}. \tag{49}$$

When $K_m < d$, this bound may be vacuous because the query Gram matrix can be rank deficient; in such regimes, equation 47 remains the meaningful matrix-error consequence.

**Trajectory rollout bound.** Writing the deterministic rollout error as

$$E_{\mathrm{traj}}(m) := \sum_{u=1}^{K_m} \|\widehat{x}_{m,S_m+u} - x_{m,S_m+u}\|_2^2,$$

and defining the finite-horizon growth factor

$$H_{m,K_m} := \sum_{r=0}^{K_m-1} \|\widehat{A}_m\|_2^r,$$

a standard unrolling argument for the rollout error recursion gives the PAC-Bayes-derived consequence

$$\mathbb{E}[E_{\mathrm{traj}}(m) \mid \mathcal{F}_{m,S_m}] \le K_m^2 H_{m,K_m}^2 \mathcal{B}_m^{\mathrm{pred}}. \tag{50}$$

The factor $K_m^2$ appears because $\mathcal{B}_m^{\mathrm{pred}}$ is a per-step average predictive bound, whereas $E_{\mathrm{traj}}(m)$ is a cumulative multi-step error.

Together, equation 47–equation 50 show how the support–query PAC-Bayes predictive bound induces explicit bounds for projected transition-matrix recovery, full Frobenius recovery under query excitation, and finite-horizon rollout accuracy. These results are PAC-Bayes-derived corollaries of the primary support–query predictive-risk bound. Their practical tightness depends on the same held-out query fit term $\widehat{R}_{m,q}^{\text{pred}}(Q_{m,\phi}^{\text{sup}})$ appearing in Corollary 2. Therefore, the matrix and rollout consequences should be viewed as informative when the query predictive fit remains controlled, and as potentially loose or vacuous when the adapted dynamics produce large query residuals or unstable rollout amplification. On the alignment event of Corollary 3, substituting its right-hand side for $\mathcal{B}_m^{\text{pred}}$ in equation 47–equation 50 shows that all three consequences inherit the same conditional support-to-query form; this corollary chain is recorded in Appendix A.6.

## 5 Performance Evaluation

We evaluate the proposed framework through a comprehensive series of experiments on both synthetic LTI systems and real-world data sets. We begin by detailing the experimental protocol, including competitive baselines and evaluation metrics. Subsequently, we present a comparative analysis demonstrating the advantages of our approach in terms of parameter identification accuracy, multi-step predictive stability, and data efficiency.

### 5.1 Baselines

We compare PBML-LTI against four primary baselines spanning both single-task estimation and cross-task structure sharing. In additional experiments, we also compare against a MAML-style LTI baseline as a stronger gradient-based meta-learning method. For all methods, each test trajectory is evaluated under the same outer support–query protocol: the first part of the trajectory is reserved as the available support window and the following part is held out as the query window. Our primary evaluation uses an adaptive-support protocol under a common maximum support budget. That is, each method receives the same available support window, but may choose an effective fit prefix within that budget before being evaluated on the same held-out query segment. This setup is intended to measure not only post-adaptation accuracy, but also how much support data each method actually needs in order to adapt well on a given task.

**Per-task ordinary least squares (OLS).** The OLS baseline fits each task independently using the closed-form least-squares estimator

$$\widehat{A}_m^{\text{ols}} = Y_m^{\text{sup}}(X_m^{\text{sup}})^\top \left( X_m^{\text{sup}}(X_m^{\text{sup}})^\top \right)^{-1}. \tag{51}$$

Here $(X_m^{\text{sup}}, Y_m^{\text{sup}})$ denotes the regression pair formed from the support data available to the method. To ensure numerical stability, we add a small diagonal jitter and use a pseudo-inverse fallback whenever the Gram matrix is ill-conditioned. OLS has no regularization hyperparameter.

**Per-task ridge (Ridge).** The Ridge baseline fits each task independently by Tikhonov-regularized least squares:

$$\widehat{A}_m^{\text{ridge}}(\lambda_{\text{reg}}) = Y_m^{\text{sup}}(X_m^{\text{sup}})^\top \left( X_m^{\text{sup}}(X_m^{\text{sup}})^\top + \lambda_{\text{reg}} I_d \right)^{-1}. \tag{52}$$

We select $\lambda_{\text{reg}}$ once using only the training tasks by grid search over a predefined candidate set with a stability-oriented criterion. In synthetic experiments, the stability threshold is fixed from the known regime parameter, $\rho_{\text{target}} = \rho_0$, and is not tuned on the test set. In settings where such regime information is not available, $\rho_{\text{target}}$ should be regarded as a baseline hyperparameter or prior stability specification. Specifically, we choose the smallest $\lambda_{\text{reg}}$ for which the fitted matrices satisfy

$$\mathbb{E}_{m \in \mathcal{T}_{\text{tr}}} \left[ \rho(\widehat{A}_m^{\text{ridge}}(\lambda_{\text{reg}})) \right] \leq \rho_{\text{target}}, \tag{53}$$

and if no candidate satisfies this condition, we choose the one minimizing the mean stability violation

$$\mathbb{E}_{m \in \mathcal{T}_{\text{tr}}} \left[ \max \left\{ 0, \rho(\widehat{A}_m^{\text{ridge}}(\lambda_{\text{reg}})) - \rho_{\text{target}} \right\} \right]. \tag{54}$$

The selected $\lambda_{\text{reg}}$ is then fixed and reused for all test tasks.

**Pooled-prior Ridge.** The Pooled-prior Ridge baseline incorporates cross-task information through a pooled mean dynamics estimate. It first computes

$$\bar{A} := \frac{1}{M_{\text{tr}}} \sum_{m=1}^{M_{\text{tr}}} \widehat{A}_m^{\text{ols,full}}, \tag{55}$$

where $\widehat{A}_m^{\text{ols,full}}$ is the OLS estimate fitted on the full training trajectory of task $m$. Given $\bar{A}$, the task-specific estimate is

$$\widehat{A}_m^{\text{pool}}(\lambda_{\text{reg}}) = \left(Y_m^{\text{sup}}(X_m^{\text{sup}})^{\top} + \lambda_{\text{reg}}\bar{A}\right)\left(X_m^{\text{sup}}(X_m^{\text{sup}})^{\top} + \lambda_{\text{reg}}I_d\right)^{-1}. \tag{56}$$

We tune $\lambda_{\text{reg}}$ using the same stability-based grid-search procedure as for Ridge, with the same $\rho_{\text{target}}$ convention, and then keep it fixed for evaluation on validation and test tasks. To verify that our conclusions are not an artifact of this stability-oriented tuning, Appendix B.3 also reports a diagnostic variant in which Ridge and Pooled-prior Ridge are tuned directly by validation rollout error, without using the $\rho_{\text{target}}$ criterion.

**Shared Subspace.** The Shared Subspace baseline assumes that the task matrices share a low-dimensional structure in vectorized form. It first fits OLS estimates on the training tasks, vectorizes them to obtain $\text{vec}(\widehat{A}_m^{\text{ols,full}})$, and computes a mean vector $a_0$ together with a rank-$k$ PCA basis $U \in \mathbb{R}^{d^2 \times k}$ from the centered vectors. For a new task, the estimate is restricted to the form

$$\text{vec}(A) = a_0 + Uc, \tag{57}$$

and the coefficients $c \in \mathbb{R}^k$ are fitted by ridge-regularized least squares on the support data. Here $\lambda_{\text{reg}}$ regularizes the coefficient estimation and is selected using the same stability-based criterion as Ridge. The subspace dimension $k$ is treated as an additional method hyperparameter.

**MAML-style LTI.** Finally, we include a gradient-based meta-learning baseline inspired by MAML (Finn et al., 2017). Since the original MAML formulation is not a closed-form LTI identification method, we implement the natural analogue for this setting: a shared initialization for the transition matrix is meta-trained so that a small number of gradient steps on the support prefix improves query performance. This provides a stronger adaptation-based meta-learning comparison while preserving the LTI regression structure.

## 5.2 Evaluation Metrics

The theory in Section 4.5 controls predictive adaptation risk on held-out future time steps. Our empirical evaluation reports three complementary quantities. The first, $E_A$, is a diagnostic of parameter recovery. The second, $E_{\text{traj}}$, is an open-loop rollout diagnostic that is more sensitive to spectral growth and long-horizon instability. The third, $\overline{S}$, measures how much support data a method actually uses in the adaptive-support analysis. Because these quantities probe different aspects of performance, they need not rank methods identically.

**Transition matrix estimation error.** To evaluate the accuracy of the estimated transition matrix $\widehat{A}_m$, we report the Frobenius squared error

$$E_A := \|\widehat{A}_m - A_m\|_F^2, \tag{58}$$

which quantifies the aggregate squared deviation between the estimated and true transition matrices across all entries. This metric treats all matrix elements equally and provides a convenient global measure of parameter recovery (Ziemann et al., 2023).

**Trajectory rollout error.** To evaluate the practical utility of the estimated dynamics $\widehat{A}_m$, we measure the open-loop multi-step rollout error on a held-out query window by recursively applying the identified transition operator. This metric is a more stringent test than $E_A$: because errors in $\widehat{A}_m$ compound over the rollout horizon, even small parameter errors can lead to large predictive deviations, especially near unstable or weakly damped modes.

Starting from the last support state $x_{m,S_m}$, we generate predictions by repeated next-step updates:

$$\widehat{x}_{m,S_m} = x_{m,S_m}, \qquad \widehat{x}_{m,t+1} = \widehat{A}_m \widehat{x}_{m,t}, \qquad t = S_m, \ldots, S_m + K_m - 1. \tag{59}$$

We then compare the predicted states to the held-out query states in open loop. The per-task trajectory rollout error is defined as

$$E_{\text{traj}} := \sum_{t=1}^{K_m} \left\| \widehat{x}_{m,S_m+t} - x_{m,S_m+t} \right\|_2^2. \tag{60}$$

The held-out query states contain the realized process noise from the underlying trajectory, but we do not inject additional noise into the predicted rollout. This choice is intentional: it isolates the quality of the learned *deterministic* transition operator from the variability introduced by fresh rollout noise. It also keeps aggregate comparisons stable across many tasks. If new rollout noise were injected for every task, the reported error would reflect not only system-identification quality but also task-specific stochastic fluctuations, substantially increasing cross-task variance and making method comparisons less stable and less interpretable. In all reported experiments, we set $K_m = 5$, so $E_{\text{traj}}$ corresponds to the accumulated error over five rollout steps. Noise-averaged rollout evaluation is a reasonable supplementary robustness check for synthetic data, but is not the primary metric reported here.

**Support-length protocol: adaptive-support evaluation under a maximum budget.** Each test trajectory is first divided into an outer support window and a held-out query window. Let $T_{\text{sup}}$ denote the maximum support budget and $T_{\text{qry}}$ the query horizon. Our evaluation studies adaptation under this common maximum support budget: each method receives the same available support window, but may choose an effective fit prefix

$$s_m^{\text{fit}} \le S_m, \qquad S_m = \min\{T_{\text{sup}}, T_m\},$$

before being evaluated on the same held-out query segment of length

$$K_m = \min\{T_{\text{qry}}, T_m - S_m\}.$$

The effective fit length is chosen by an inner validation-based prefix search carried out entirely within the support window. Specifically, we reserve a short validation suffix of length $T_{\text{val}}$ inside the support window, consider a grid of candidate fit prefixes, fit the method using only the first candidate prefix, and score that candidate on the remaining validation portion. The selected prefix is then used to refit the method before final evaluation on the held-out outer query window.

We view this adaptive-support protocol as a natural few-shot evaluation for heterogeneous tasks. It measures not only final predictive performance under a common support budget, but also how much support a method actually uses before it adapts well on a given task. There is no explicit penalty favoring shorter prefixes: a shorter prefix is selected only when additional transitions do not improve the validation objective. Thus, $\overline{S}$ should be interpreted as an empirical measure of adaptive sample efficiency, not as a claim that fewer observations are always better. In finite temporally dependent trajectories, additional transitions can alter the fitted spectral structure of the transition matrix, and open-loop rollout performance can be sensitive to these spectral changes even when one-step fit changes only slightly.

The reference transition matrix, when available, is not used to choose the support prefix in the main adaptive-support protocol. Prefix selection is based only on validation error computed on the validation suffix inside the support window. The reference matrix is used only after fitting, to report the diagnostic transition-matrix error $E_A$. Any oracle-assisted support-sensitivity variant would be a separate diagnostic analysis and is not part of the main evaluation protocol reported here.

**Average support length.** We report the average selected support length in the adaptive-support analysis as

$$\overline{S} := \frac{1}{|\mathcal{T}_{\text{test}}|} \sum_{m \in \mathcal{T}_{\text{test}}} s_m^{\text{fit}}. \tag{61}$$

Accordingly, $\overline{S}$ should be interpreted as the average *effective* amount of support a method actually uses under a common maximum support budget. This makes $\overline{S}$ an interpretable quantity in its own right: it reflects adaptive few-shot efficiency, while the accompanying $E_A$ and $E_{\text{traj}}$ values indicate the quality of the resulting adaptation.

### 5.3  Synthetic Data Experiment

### 5.3.1  Task Generation Mechanism

For synthetic experiments, we generate a meta-dataset by first fixing environment-level parameters and then sampling multiple independent LTI tasks from a shared environment distribution, consistent with Assumption 3.

We fix an environment hyper-parameter $\phi_\star = (W_\star, V_\star, \sigma_\star^2)$ and sample tasks i.i.d. as

$$A_m \mid \phi_\star \sim \mathcal{MN}(W_\star,\ I_d,\ V_\star), \qquad m = 1, 2, \ldots \tag{62}$$

which matches the prior family equation 9, and we generate trajectories according to equation 1 with Gaussian process noise

$$w_{m,t+1} \sim \mathcal{N}(0, \sigma_\star^2 I_d), \qquad t = 0, \ldots, T_m - 1. \tag{63}$$

We construct a stable meta-mean matrix $W_\star \in \mathbb{R}^{d \times d}$ by drawing a Gaussian random matrix and scaling it so that its spectral radius is at most $0.9\,\rho_0$. We set the true column covariance to $V_\star = v_{\text{true}} I_d$ and the process noise variance to $\sigma_\star^2 = \sigma_{\text{true}}^2$.

For each task $m$, we sample $A_m$ from equation 62, then rollout a trajectory from an initial state $x_{m,0} \sim \mathcal{N}(0, I_d)$, and generate a trajectory according to equation 1

$$x_{m,t+1} = A_m x_{m,t} + w_{m,t+1}, \qquad t = 0, \ldots, T_m - 1. \tag{64}$$

From each trajectory, we form the regression matrices $X_m$ and $Y_m$ as in equation 3 and store them together with the ground-truth dynamics $A_m$.

### 5.3.2  Experiment Setup

We consider a synthetic high-dimensional identification setting in which the state dimension exceeds the available trajectory length. We fix the environment parameters to $v_{\text{true}} = 0.5$ and $\sigma_{\text{true}}^2 = 0.01$, and study two dynamical regimes by varying the spectral-radius parameter $\rho_0$: a near-stable regime with $\rho_0 = 0.95$ and an unstable regime with $\rho_0 = 4.95$. The stability threshold $\rho_{\text{target}}$ used to tune the ridge regularization parameter $\lambda_{\text{reg}}$ in equation 53 and equation 54, is set to match the target spectral radius $\rho_0$. The unstable setting is included as a finite-horizon stress test of the methods. It lies outside the favorable near-stable regime in which one should expect the tightest rollout consequences from the theory. In this regime, errors in the estimated transition matrix can be strongly amplified under multi-step rollouts, potentially leading to severe prediction instability even when one-step prediction error or Frobenius matrix error appears small. Furthermore, we consider three state dimensions $d \in \{10, 25, 50\}$ with a fixed trajectory length $T_m = 25$. In the medium- and high-dimensional cases, the number of unknown parameters in $A_m$ grows as $d^2$ and far exceeds the number of observed transitions, placing the problem in a few-shot regime that stresses both estimation accuracy and predictive stability.

First, we uniformly sample 100 tasks to compose the meta-training set. We then construct two distinct evaluation sets. The *common-case* test set consists of tasks where the entrywise mean dynamics fall within one standard deviation of the population mean, representing typical in-distribution systems. In contrast, the *edge-case* test set is formed by selecting tasks from the extremes of the overall parameter distribution, representing out-of-distribution systems intended to stress-test robustness under atypical dynamics. Figure 1 depicts the resulting task distributions.

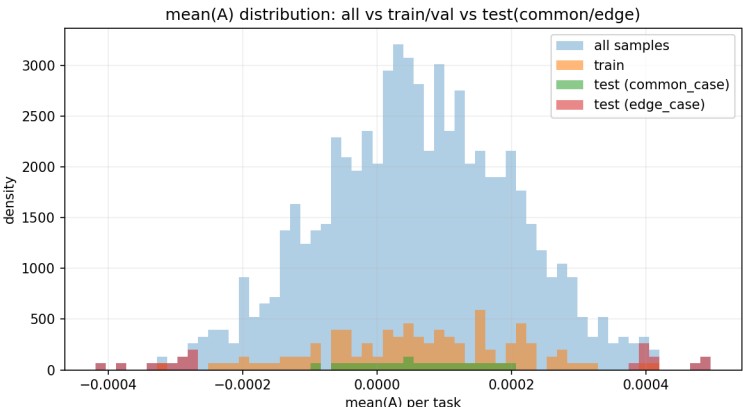

Figure 1: This figure illustrates the distribution of sampled tasks as a function of the entrywise mean of the dynamics matrix $A_m$. The blue histogram shows all sampled tasks in the initial pool. The yellow region indicates the subset selected for training and validation. The green region corresponds to the in-distribution test tasks, while the red markers denote the edge-case test tasks chosen from the extremes of the distribution.

### 5.3.3 Experiment Results

**Stable-system results.** Table 1 reports results on the stable synthetic environment with $\rho_0 = 0.95$, where all tasks are generated with spectral radius below one and therefore do not exhibit unstable rollouts. The reported $\overline{S}$ values are part of the adaptive-support evaluation: under the same maximum support budget, they indicate how much support each method actually used before final evaluation on the held-out query window.

Across all dimensions and both common-case and edge-case settings, PBML-LTI achieves the lowest transition-matrix error $E_A$ while using a short validation-selected support prefix. The improvement in $E_A$ is substantial. For example, when $d = 50$, PBML-LTI reduces $E_A$ from approximately 0.84 for OLS/Ridge, 0.67 for Pooled-prior Ridge, 0.86 for Shared Subspace, and about 14.9 for MAML-LTI to about 0.21. Similar gains appear for $d = 25$ and $d = 10$. These results indicate that the learned matrix-normal prior provides an effective inductive bias for rapid adaptation from short trajectories.

The MAML-style LTI baseline performs worse than PBML-LTI in all stable settings, especially in transition-matrix recovery. Although MAML-LTI uses relatively short support prefixes, its gradient-based adaptation from a shared initialization does not recover the task-specific transition matrices as accurately as the closed-form Bayesian update under the learned prior. This highlights the advantage of exploiting the conjugate LTI structure rather than relying only on iterative gradient adaptation.

Figure 2 further illustrates these estimation patterns. Competing baselines tend to produce more conservative or structurally biased estimates, whereas PBML-LTI more accurately captures the transition structure while preserving stable behavior.

Differences in rollout error $E_{\mathrm{traj}}$ are comparatively small in the stable regime. Since the systems are stable over the short query horizon, one-step prediction errors do not amplify rapidly. As a result, OLS, Ridge, Pooled-prior Ridge, Shared Subspace, and PBML-LTI can have similar rollout errors despite large differences in transition-matrix error. In several $d = 10$ settings, the rollout values are essentially tied up to the reported precision. By contrast, MAML-LTI has noticeably larger rollout error in all stable settings. Thus, in stable systems, $E_A$ more clearly distinguishes parameter-recovery quality, whereas $E_{\mathrm{traj}}$ is less sensitive unless estimation errors substantially affect short-horizon prediction.

Finally, performance on the edge-case test tasks closely tracks the common-case results across all three dimensions. The ranking of methods remains largely unchanged: PBML-LTI consistently achieves the best transition-matrix recovery, the shortest effective support usage, and rollout performance that is best or tied

Table 1: Performance comparison across methods in common-case and edge-case settings on stable systems ($\rho_0 = 0.95$). $E_A$ is the mean $\pm$ standard deviation of the transition-matrix error, $E_{\text{traj}}$ is the mean $\pm$ standard deviation of the rollout error, and $\overline{S}$ reports the average selected support length. Lower is better; boldface marks the lowest sample mean. Paired significance tests in Appendix B.4.1 qualify these comparisons.

| Dimension | Setting | Metric | OLS | Ridge | Pooled-prior Ridge | Shared Subspace | MAML-LTI | PBML-LTI |
|---|---|---|---|---|---|---|---|---|
| 50 | Common-case | $E_A$ | $0.8411 \pm .015$ | $0.8410 \pm .015$ | $0.6743 \pm .011$ | $0.8572 \pm .050$ | $14.893 \pm .188$ | $\mathbf{0.2068 \pm .0052}$ |
| | | $E_{\text{traj}}$ | $0.0258 \pm .0018$ | $0.0258 \pm .0018$ | $0.0258 \pm .0018$ | $0.0258 \pm .0018$ | $0.1081 \pm .0820$ | $\mathbf{0.0257 \pm .0018}$ |
| | | $\overline{S}$ | $2.00$ | $2.00$ | $2.55$ | $10.25$ | $2.90$ | $\mathbf{1.35}$ |
| | Edge-case | $E_A$ | $0.8382 \pm .010$ | $0.8381 \pm .010$ | $0.6695 \pm .009$ | $0.8811 \pm .073$ | $14.830 \pm .260$ | $\mathbf{0.2071 \pm .0042}$ |
| | | $E_{\text{traj}}$ | $0.0254 \pm .0018$ | $0.0254 \pm .0018$ | $0.0254 \pm .0018$ | $0.0254 \pm .0018$ | $0.1105 \pm .1009$ | $\mathbf{0.0253 \pm .0018}$ |
| | | $\overline{S}$ | $2.00$ | $2.00$ | $2.85$ | $11.05$ | $2.65$ | $\mathbf{1.40}$ |
| 25 | Common-case | $E_A$ | $0.7140 \pm .0181$ | $0.7140 \pm .0181$ | $0.8544 \pm .0186$ | $0.7731 \pm .0189$ | $1.7566 \pm .0512$ | $\mathbf{0.0394 \pm .0020}$ |
| | | $E_{\text{traj}}$ | $0.0133 \pm .0013$ | $0.0133 \pm .0013$ | $0.0134 \pm .0018$ | $0.0134 \pm .0013$ | $0.0338 \pm .0104$ | $\mathbf{0.0121 \pm .0013}$ |
| | | $\overline{S}$ | $2.00$ | $2.00$ | $2.00$ | $13.95$ | $1.85$ | $\mathbf{1.20}$ |
| | Edge-case | $E_A$ | $0.7196 \pm .0181$ | $0.7196 \pm .0181$ | $0.8504 \pm .0230$ | $0.7892 \pm .0191$ | $1.7702 \pm .0696$ | $\mathbf{0.0399 \pm .0025}$ |
| | | $E_{\text{traj}}$ | $0.0135 \pm .0018$ | $0.0135 \pm .0018$ | $0.0122 \pm .0012$ | $0.0121 \pm .0013$ | $0.0318 \pm .0233$ | $\mathbf{0.0121 \pm .0013}$ |
| | | $\overline{S}$ | $2.00$ | $2.00$ | $2.00$ | $13.95$ | $1.60$ | $\mathbf{1.10}$ |
| 10 | Common-case | $E_A$ | $0.4449 \pm .0472$ | $0.4447 \pm .0471$ | $0.0887 \pm .0079$ | $0.1088 \pm .0532$ | $0.0357 \pm .0020$ | $\mathbf{0.0053 \pm .0007}$ |
| | | $E_{\text{traj}}$ | $0.0056 \pm .0011$ | $0.0056 \pm .0011$ | $0.0055 \pm .0009$ | $0.0055 \pm .0009$ | $0.0112 \pm .0054$ | $\mathbf{0.0055 \pm .0009}$ |
| | | $\overline{S}$ | $2.65$ | $2.65$ | $1.80$ | $7.95$ | $1.55$ | $\mathbf{1.05}$ |
| | Edge-case | $E_A$ | $0.4501 \pm .0479$ | $0.4498 \pm .0479$ | $0.0910 \pm .0070$ | $0.0930 \pm .0307$ | $0.0358 \pm .0032$ | $\mathbf{0.0054 \pm .0008}$ |
| | | $E_{\text{traj}}$ | $0.0056 \pm .0011$ | $0.0056 \pm .0011$ | $0.0056 \pm .0011$ | $0.0057 \pm .0009$ | $0.0132 \pm .0047$ | $\mathbf{0.0056 \pm .0010}$ |
| | | $\overline{S}$ | $2.80$ | $2.80$ | $1.90$ | $10.40$ | $1.60$ | $\mathbf{1.10}$ |

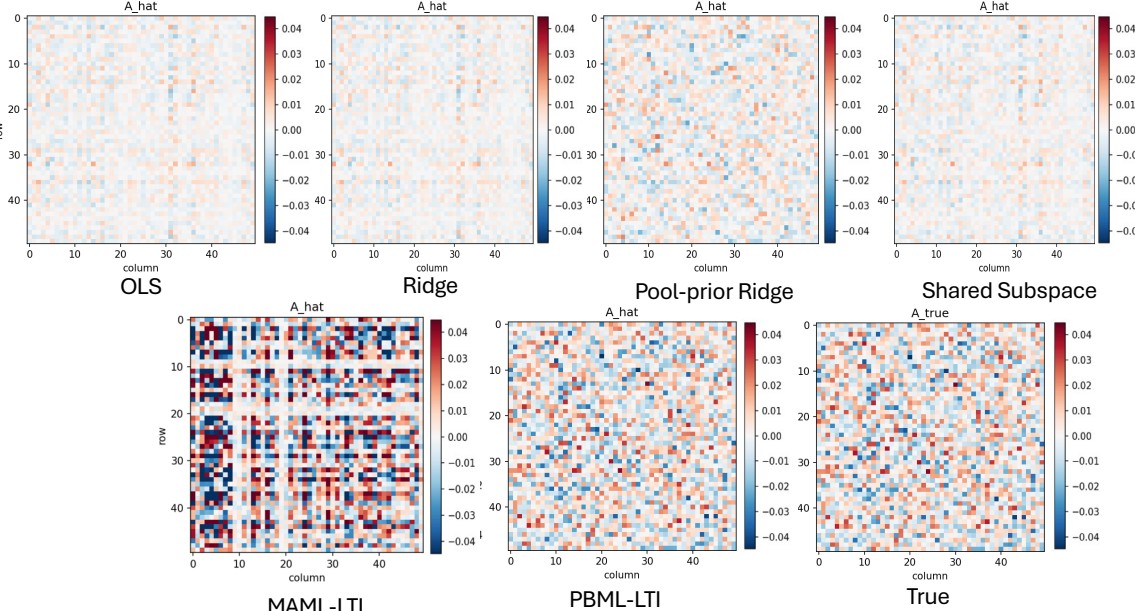

Figure 2: Heatmap comparison of the estimated and ground-truth transition matrices for a representative stable test system in the common-case setting. An extended version is provided in the Appendix, Figure 6.

with the best up to reporting precision. This suggests that, under stable dynamics, the learned prior transfers reliably to both typical and atypical systems, supporting robust few-shot identification. Fixed-prefix support sweeps in Appendix B.2 further show that PBML-LTI already attains low error at small support lengths in the stable regime, rather than relying only on the adaptive prefix-selection procedure.

**Unstable-system results.** Table 2 reports results for the unstable environment, where $\rho_0 = 4.95$ and prediction errors can amplify rapidly under open-loop rollout. The table includes the MAML-LTI baseline,

Table 2: Performance comparison across methods in common-case and edge-case settings on unstable systems ($\rho_0 = 4.95$). $E_A$ is the mean ± standard deviation of the transition-matrix error, $E_{\text{traj}}$ is the mean ± standard deviation of the trajectory rollout error, and $\overline{S}$ reports the average effective support length selected by the adaptive-support protocol. Lower is better; boldface marks the lowest sample mean. Paired significance tests in Appendix B.4.1 qualify these comparisons.

| Dimension | Setting | Metric | OLS | Ridge | Pooled-prior Ridge | Shared Subspace | MAML-LTI | PBML-LTI |
|---|---|---|---|---|---|---|---|---|
| 50 | Common-case | $E_A$ | $16.923 \pm .345$ | $15.902 \pm .340$ | $7.538 \pm .102$ | $6.419 \pm .201$ | $8.9787 \pm .0754$ | $\mathbf{0.156 \pm .005}$ |
| | | $E_{\text{traj}}$ | $0.040 \pm .004$ | $0.040 \pm .003$ | $0.038 \pm .003$ | $0.038 \pm .003$ | $0.0414 \pm .0048$ | $\mathbf{0.037 \pm .003}$ |
| | | $\overline{S}$ | $9.75$ | $15.00$ | $15.00$ | $14.20$ | $\mathbf{2.55}$ | $4.15$ |
| | Edge-case | $E_A$ | $16.776 \pm .393$ | $15.874 \pm .349$ | $7.532 \pm .079$ | $6.455 \pm .406$ | $8.9841 \pm .0608$ | $\mathbf{0.157 \pm .003}$ |
| | | $E_{\text{traj}}$ | $0.040 \pm .004$ | $0.039 \pm .004$ | $0.038 \pm .003$ | $0.038 \pm .003$ | $0.0409 \pm .0062$ | $\mathbf{0.036 \pm .003}$ |
| | | $\overline{S}$ | $10.15$ | $15.00$ | $15.00$ | $13.65$ | $\mathbf{2.55}$ | $4.25$ |
| 25 | Common-case | $E_A$ | $10.344 \pm .659$ | $9.907 \pm .619$ | $1.587 \pm .050$ | $35.268 \pm 12.856$ | $2.5229 \pm .0370$ | $\mathbf{0.029 \pm .001}$ |
| | | $E_{\text{traj}}$ | $0.088 \pm .058$ | $0.064 \pm .025$ | $\mathbf{0.053 \pm .024}$ | $27.151 \pm 28.807$ | $1.1349 \pm 1.9963$ | $0.080 \pm .052$ |
| | | $\overline{S}$ | $14.10$ | $14.90$ | $14.75$ | $14.35$ | $\mathbf{2.05}$ | $6.05$ |
| | Edge-case | $E_A$ | $11.023 \pm .938$ | $10.445 \pm .856$ | $1.586 \pm .057$ | $33.889 \pm 7.925$ | $2.5324 \pm .0469$ | $\mathbf{0.029 \pm .001}$ |
| | | $E_{\text{traj}}$ | $0.086 \pm .049$ | $0.071 \pm .034$ | $\mathbf{0.042 \pm .009}$ | $28.933 \pm 47.100$ | $2.2052 \pm 4.7758$ | $0.086 \pm .062$ |
| | | $\overline{S}$ | $13.50$ | $14.55$ | $14.90$ | $14.30$ | $\mathbf{1.65}$ | $6.00$ |
| 10 | Common-case | $E_A$ | $0.845 \pm .764$ | $0.697 \pm .581$ | $\mathbf{0.007 \pm .002}$ | $0.013 \pm .005$ | $0.1473 \pm 1.8158$ | $0.139 \pm .046$ |
| | | $E_{\text{traj}}$ | $193.704 \pm 459.303$ | $\mathbf{57.628 \pm 154.718}$ | $21935.364 \pm 69327.156$ | $721888.809 \pm 1044209.839$ | $9.739 \times 10^6 \pm 1.887 \times 10^7$ | $57.919 \pm 144.808$ |
| | | $\overline{S}$ | $13.35$ | $14.30$ | $9.25$ | $\mathbf{3.20}$ | $8.35$ | $14.35$ |
| | Edge-case | $E_A$ | $0.572 \pm .630$ | $0.515 \pm .566$ | $\mathbf{0.008 \pm .001}$ | $0.014 \pm .003$ | $0.2737 \pm 1.6215$ | $0.132 \pm .040$ |
| | | $E_{\text{traj}}$ | $296.091 \pm 755.920$ | $131.024 \pm 389.381$ | $16433.662 \pm 49301.429$ | $709906.413 \pm 1155741.042$ | $8.756 \times 10^7 \pm 1.987 \times 10^8$ | $\mathbf{31.400 \pm 38.879}$ |
| | | $\overline{S}$ | $13.90$ | $14.15$ | $10.10$ | $\mathbf{3.20}$ | $8.05$ | $14.45$ |

which provides a stronger gradient-based meta-learning comparison. The reported $\overline{S}$ values arise from the adaptive-support protocol described above and should be interpreted as the *effective* amount of support each method actually used under a common maximum support budget. Thus, the table compares not only final accuracy, but also adaptive support usage in a regime where different tasks may require different amounts of calibration data.

A key pattern in the unstable regime is that transition-matrix recovery, rollout accuracy, and support efficiency no longer always rank methods identically. Accurate long-horizon prediction depends not only on small Frobenius error, but also on accurately capturing the dominant spectral modes of the dynamics. Consequently, a method may achieve a small $E_A$ while still producing poor rollout trajectories if it slightly misestimates unstable eigenvalues or eigenspaces. Conversely, a method may obtain competitive rollout error without achieving the best global Frobenius recovery.

In the high-dimensional setting ($d = 50$), PBML-LTI achieves the strongest overall estimation and prediction performance. It attains the lowest transition-matrix error and the lowest rollout error in both common-case and edge-case subsets. Although MAML-LTI selects a shorter support prefix on average, its transition-matrix error remains much larger than that of PBML-LTI, and its rollout error is also worse. Thus, in the most high-dimensional few-shot regime, the learned Bayesian prior provides a substantially more effective adaptation mechanism than both classical estimators and the gradient-based MAML-style baseline.

For $d = 25$, PBML-LTI again achieves the best transition-matrix recovery by a large margin. It uses substantially shorter support prefixes than OLS, Ridge, Pooled-prior Ridge, and Shared Subspace, although MAML-LTI selects the shortest prefix. However, MAML-LTI's short support usage comes with much worse $E_A$ and significantly worse rollout error. Regarding trajectory rollout, Pooled-prior Ridge attains the lowest mean $E_{\text{traj}}$ in this dimension. This reflects the fact that aggressive shrinkage toward a pooled mean can sometimes stabilize rollouts even when it does not recover the transition matrix as accurately. PBML-LTI therefore offers the best matrix recovery and a favorable support–accuracy trade-off, while Pooled-prior Ridge is strongest on rollout error in this particular unstable $d = 25$ setting.

For $d = 10$, the behavior is more delicate. Pooled-prior Ridge and Shared Subspace achieve the smallest transition-matrix errors, but they can produce extremely large rollout errors. This illustrates a fundamental limitation of interpreting $E_A$ alone: the Frobenius norm measures an average entrywise discrepancy, whereas open-loop prediction is governed by spectral alignment and repeated multiplication by the estimated transition matrix. Small errors in unstable directions can compound rapidly and lead to catastrophic trajectory divergence. In this dimension, Ridge and PBML-LTI produce the most reliable rollout behavior among the

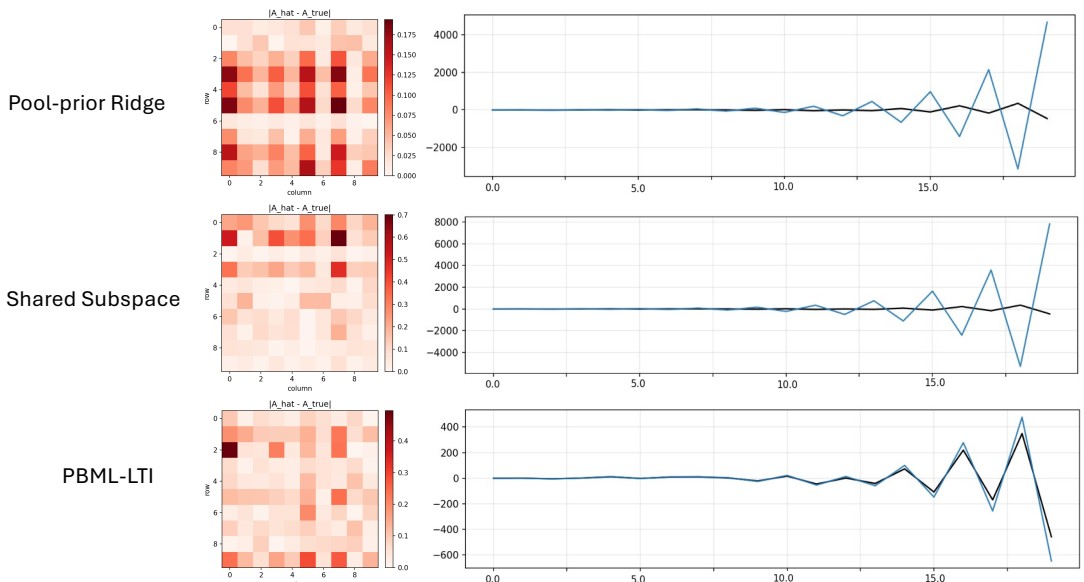

Figure 3: The left panels show heatmaps of the absolute differences between the estimated and ground-truth transition matrices for a representative unstable test system with state dimension $d = 10$ in the common-case setting. The right panels display open-loop trajectory rollouts generated using the estimated transition matrix (black) alongside the corresponding ground-truth trajectories (blue) for the same system. An extended visualization of the estimated transition matrix, the ground-truth transition matrix, and their absolute differences is provided in Figure 8 in the Appendix, while the corresponding open-loop rollout trajectories are shown in Figure 9 in the Appendix.

non-catastrophic methods: Ridge has the lowest mean rollout error in the common-case subset, while PBML-LTI has the lowest mean rollout error in the edge-case subset. MAML-LTI selects shorter support prefixes, but its rollout errors become extremely large, indicating that the learned gradient-based initialization is not sufficient to control unstable modes in this low-dimensional stress-test setting.

Figure 3 illustrates this phenomenon qualitatively. Methods whose transition estimates appear competitive in Frobenius norm can nevertheless induce dramatically different open-loop trajectories because small spectral misalignments are repeatedly amplified. This effect is especially visible for structurally biased estimators such as Pooled-prior Ridge and Shared Subspace, and it also affects MAML-LTI in the low-dimensional unstable setting.

Overall, the unstable-system experiments show that PBML-LTI is strongest in the high-dimensional few-shot regime, where it achieves both accurate transition recovery and stable rollout prediction. In lower-dimensional unstable settings, the metrics become more decoupled: some baselines obtain smaller Frobenius error or shorter support usage, but this does not necessarily translate into reliable rollout behavior. Therefore, the unstable-regime results should be read jointly through $E_A$, $E_{\text{traj}}$, and $\overline{S}$, rather than through any single metric in isolation. The fixed-prefix sweeps in Appendix B.2 also show that PBML-LTI does not mechanically select the shortest prefix: in unstable settings, its validation-selected support length increases when additional calibration data are useful for controlling unstable modes.

### 5.4 Real Data Experiment

#### 5.4.1 Functional Magnetic Resonance Imaging (fMRI) Dataset

We construct a real-data LTI benchmark from the OpenNeuro dataset ds000244 (Pinho et al., 2018). Raw fMRI volumes are converted into multivariate Region-of-Interest (ROI) time series by averaging voxel signals within atlas-defined parcels using the Schaefer parcellation (Schaefer et al., 2018). The resulting ROI signals

are further preprocessed with standardization, motion-confound regression when available, and per-run mean centering.

To obtain many *related* identification tasks from each run, we segment each ROI time series into overlapping windows of length $L$ with stride $s$. Each window is treated as one LTI task instance $m$ with states $\{x_{m,t}\}_{t=0}^{L-1}$, and we form the one-step regression matrices

$$X_m := [x_{m,0}, \ldots, x_{m,L-2}] \in \mathbb{R}^{d \times (L-1)}, \qquad Y_m := [x_{m,1}, \ldots, x_{m,L-1}] \in \mathbb{R}^{d \times (L-1)}. \tag{65}$$

These $(X_m, Y_m)$ pairs are saved in the same task format as in the synthetic experiments. The resulting windows are treated as *distinct but related* task instances rather than arbitrary unrelated samples: they are extracted from the same pool of subjects, sessions, and experimental conditions, and overlapping windows from the same run can be viewed as nearby local dynamical regimes within a shared subject- and condition-specific environment. This is precisely the type of cross-task structure that PBML-LTI is designed to exploit.

Unlike synthetic data, fMRI does not provide a physical ground-truth transition matrix. We therefore define a ridge-based *reference* transition matrix $A_m^{\text{ref}}$ for each window by fitting

$$A_m^{\text{ref}} := \arg \min_{A \in \mathbb{R}^{d \times d}} \left\| Y_m - A X_m \right\|_F^2 + \lambda_{\text{ref}} \|A\|_F^2, \tag{66}$$

where $\| \cdot \|_F$ is the Frobenius norm and $\lambda_{\text{ref}} = 0.0001$ is a fixed ridge weight used only for constructing the evaluation reference. For consistency, the ridge regularization parameter $\lambda_{\text{reg}}$ in equation 53 and equation 54 is set equal to $\lambda_{\text{ref}}$. This optimization has the closed-form solution

$$A_m^{\text{ref}} = Y_m X_m^\top \left( X_m X_m^\top + \lambda_{\text{ref}} I_d \right)^{-1}. \tag{67}$$

Importantly, all methods are trained and adapted from trajectories $(X_m, Y_m)$; the matrices $A_m^{\text{ref}}$ are used only as a consistent evaluation reference for $E_A$.

In our experiments, we use a predefined train–test split constructed at the window level. Each task corresponds to a short temporal window extracted from a single fMRI run, and windows from different subjects, sessions, and experimental conditions are distributed across the training and test sets according to this split. The original BIDS (Brain Imaging Data Structure) task labels are retained only as metadata for grouping and reporting and are not used by the learning algorithms. Here, BIDS task labels identify the experimental paradigm associated with an fMRI run, for example motor, social, language, or emotional conditions; they are descriptors of the data source rather than supervision targets for learning.

In total, the training set contains 141 task instances spanning 17 distinct task labels, while the test set contains 14 task instances spanning 8 task labels. Each task instance has state dimensionality $d = 200$ and trajectory length $T = 100$. [6]

### 5.4.2 Experiment Result

Table 3 summarizes identification and prediction performance on the fMRI benchmark, reporting mean $\pm$ standard deviation across window-level test tasks. Overall, PBML-LTI achieves the lowest average transition-matrix error $E_A$, the lowest open-loop rollout error $E_{\text{traj}}$, and the shortest average selected support prefix length $\overline{S}$ among all compared methods.

The improvement in transition-matrix error is modest but consistent. PBML-LTI obtains the lowest $E_A$ ($124.734 \pm 268.471$), improving over Pooled-prior Ridge ($129.643 \pm 263.244$), OLS ($141.766 \pm 259.513$), Ridge ($142.493 \pm 259.174$), Shared Subspace ($147.876 \pm 256.798$), and the MAML-style LTI baseline ($161.45 \pm 259.30$). This suggests that the learned matrix-normal prior provides a useful inductive bias even in the real-data setting, which is also further corroborated by the transition matrix prediction visualization in Figure 4. The rollout results show a clearer separation. PBML-LTI achieves the lowest average trajectory error ($123.663 \pm 206.714$), slightly improving over the single-task and pooled estimators, while Shared Subspace and MAML-LTI exhibit much larger rollout errors. In particular, MAML-LTI attains a rollout error of $1704.56 \pm 3391.01$,

---

[6]More technical details of the dataset processing are provided in Appendix A.7.

Table 3: Performance comparison on the fMRI dataset. Results are reported as mean $\pm$ standard deviation across test tasks. Lower is better; boldface marks the lowest sample mean. The row $\overline{S}$ reports the average selected support prefix length. Paired significance tests comparing PBML-LTI against OLS, Ridge, Pooled-prior Ridge, Shared Subspace, and MAML-LTI are reported in Appendix B.4.2 to qualify the statistical significance of the observed differences.

| Metric | OLS | Ridge | Pooled-prior Ridge | Shared Subspace | MAML-LTI | PBML-LTI |
|---|---|---|---|---|---|---|
| $E_A$ | $141.766 \pm 259.513$ | $142.493 \pm 259.174$ | $129.643 \pm 263.244$ | $147.876 \pm 256.798$ | $161.45 \pm 259.30$ | $\mathbf{124.734 \pm 268.471}$ |
| $E_{\text{traj}}$ | $134.126 \pm 221.939$ | $134.078 \pm 221.941$ | $125.892 \pm 209.232$ | $679.799 \pm 968.563$ | $1704.56 \pm 3391.01$ | $\mathbf{123.663 \pm 206.714}$ |
| $\overline{S}$ | $59.42$ | $61.21$ | $60.85$ | $24.00$ | $23.07$ | $\mathbf{21.21}$ |

indicating that gradient-based adaptation from a learned initialization can be brittle in this high-dimensional fMRI setting. This is consistent with the synthetic experiments: open-loop rollout performance is sensitive not only to average Frobenius error, but also to spectral alignment and stability of the learned transition operator.

The support-length results further highlight adaptive data efficiency. PBML-LTI uses the shortest average support prefix ($\overline{S} = 21.21$), slightly shorter than MAML-LTI (23.07) and Shared Subspace (24.00), and substantially shorter than OLS, Ridge, and Pooled-prior Ridge, which use roughly sixty support steps on average. Thus, PBML-LTI achieves the best average errors while also requiring the least support data under the adaptive-support protocol. By contrast, MAML-LTI and Shared Subspace also use relatively short prefixes, but their shorter support usage does not translate into comparable rollout accuracy.

The large standard deviations across methods reflect substantial heterogeneity across fMRI windows and subjects. Since these quantities are squared-error metrics computed on a heterogeneous real-data benchmark, standard deviations can be larger than the corresponding means. This variability is expected and reinforces the importance of methods that remain stable when adapting from limited support data.

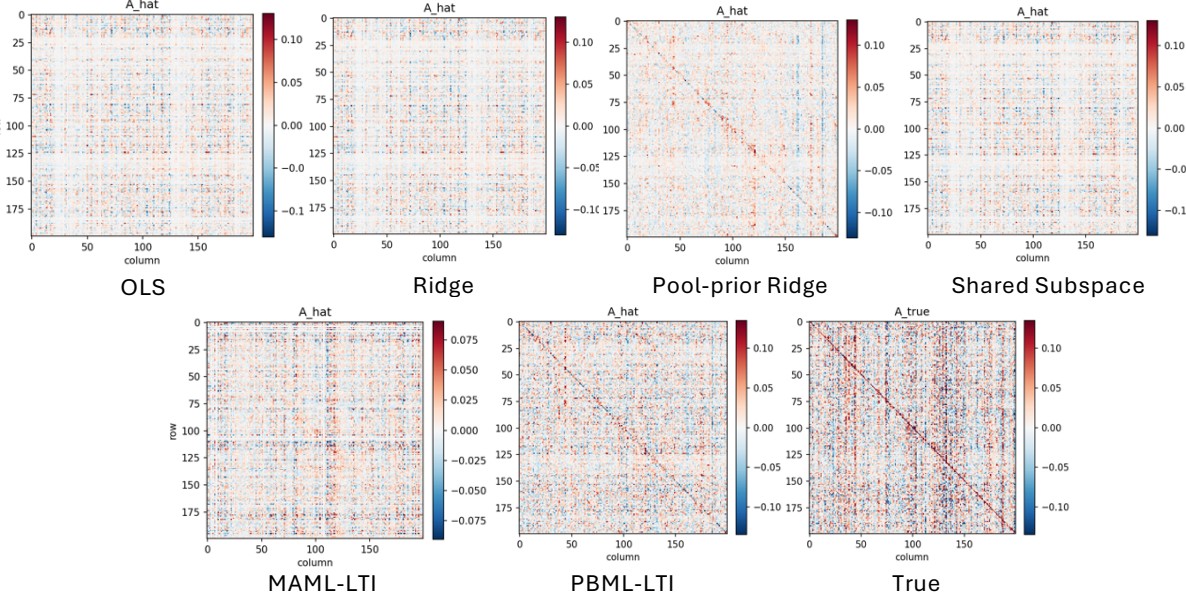

Figure 4: Estimated and reference transition matrices for a representative task from the fMRI dataset. Since fMRI does not provide a physical ground-truth transition matrix, the comparison is made against the ridge-based full-window reference matrix $A_m^{\text{ref}}$. For a more detailed analysis, including an extended set of heatmaps and the corresponding open-loop rollout trajectories, please refer to Figure 11 in the Appendix.

As in the synthetic experiments, the reported $\overline{S}$ values arise from validation-selected adaptive support under a common maximum support budget. PBML-LTI achieves the best average errors while using the shortest average effective support length. Shared Subspace and MAML-LTI also use relatively short prefixes, but their shorter support usage does not translate into comparable rollout accuracy, suggesting that their adaptation mechanisms can be mismatched to heterogeneous fMRI window dynamics. The large standard deviations across all methods indicate substantial variability across windows, which is expected in this real-data setting and underscores the importance of methods that remain robust when adapting from limited support data.

## 5.5 Prior-Learning and Regularization Ablation

We next ablate two components of PBML-LTI: the prior-learning mechanism and the auxiliary regularization terms. These variants are diagnostic modifications of the proposed method rather than external baselines.

For prior learning, we consider two variants. The fixed-prior PBML-LTI variant uses the same matrix-normal conjugate adaptation rule as PBML-LTI, but replaces the learned prior parameters with a fixed generic prior. This isolates the benefit of learning a transferable prior from related training tasks. The type-II maximum-likelihood / empirical-Bayes variant learns the prior parameters by maximizing a marginal-likelihood criterion rather than by optimizing the PAC-Bayes-motivated fit–KL surrogate. This tests whether the proposed fit–KL objective provides advantages beyond standard empirical-Bayes prior learning. For regularization, we

Table 4: Combined prior-learning and regularization ablation across stable/unstable regimes and dimensions in the common-case setting. The first two variants ablate prior learning, while the final three variants ablate one regularization component at a time relative to full PBML-LTI. Lower is better; for each metric within each regime–dimension block, boldface marks the lowest sample mean among reported values. Paired significance tests for these ablation comparisons are reported in Appendix B.4.3.

| Regime | Dimension | Variant | $E_A$ | $E_{\text{traj}}$ | $\overline{S}$ |
|---|---|---|---|---|---|
| Stable | $d = 50$ | Fixed-prior PBML-LTI | $24.779 \pm 0.479$ | $0.0839 \pm 0.0227$ | $14.00$ |
| | | Type-II ML / empirical Bayes | $1.2236 \pm 0.0313$ | $0.0568 \pm 0.0113$ | $10.80$ |
| | | PBML-LTI | $\mathbf{0.2068 \pm 0.0052}$ | $\mathbf{0.0257 \pm 0.0018}$ | $\mathbf{1.35}$ |
| | | No prior-mean shrinkage | $0.2155 \pm 0.0056$ | $0.0268 \pm 0.0020$ | $1.42$ |
| | | No covariance conditioning | $0.2115 \pm 0.0054$ | $0.0263 \pm 0.0019$ | $1.39$ |
| | | No stability regularizer | $0.2088 \pm 0.0053$ | $0.0261 \pm 0.0019$ | $1.38$ |
| | $d = 25$ | Fixed-prior PBML-LTI | $9.1298 \pm 0.4271$ | $0.0325 \pm 0.0098$ | $14.00$ |
| | | Type-II ML / empirical Bayes | $0.6165 \pm 0.0302$ | $0.0261 \pm 0.0072$ | $12.10$ |
| | | PBML-LTI | $\mathbf{0.0394 \pm 0.0020}$ | $\mathbf{0.0121 \pm 0.0013}$ | $\mathbf{1.20}$ |
| | | No prior-mean shrinkage | $0.0425 \pm 0.0022$ | $0.0129 \pm 0.0014$ | $1.27$ |
| | | No covariance conditioning | $0.0412 \pm 0.0021$ | $0.0125 \pm 0.0014$ | $1.24$ |
| | | No stability regularizer | $0.0405 \pm 0.0021$ | $0.0124 \pm 0.0014$ | $1.23$ |
| | $d = 10$ | Fixed-prior PBML-LTI | $1.2189 \pm 0.4011$ | $0.0147 \pm 0.0112$ | $12.55$ |
| | | Type-II ML / empirical Bayes | $0.0726 \pm 0.0144$ | $0.0121 \pm 0.0056$ | $12.15$ |
| | | PBML-LTI | $\mathbf{0.0053 \pm 0.0007}$ | $\mathbf{0.0055 \pm 0.0009}$ | $\mathbf{1.05}$ |
| | | No prior-mean shrinkage | $0.0059 \pm 0.0008$ | $0.0059 \pm 0.0010$ | $1.10$ |
| | | No covariance conditioning | $0.0057 \pm 0.0008$ | $0.0057 \pm 0.0009$ | $1.08$ |
| | | No stability regularizer | $0.0056 \pm 0.0008$ | $0.0057 \pm 0.0009$ | $1.07$ |
| Unstable | $d = 50$ | Fixed-prior PBML-LTI | $16.059 \pm 0.312$ | $0.0401 \pm 0.0041$ | $14.00$ |
| | | Type-II ML / empirical Bayes | $0.4634 \pm 0.0105$ | $\mathbf{0.0367 \pm 0.0035}$ | $4.60$ |
| | | PBML-LTI | $\mathbf{0.156 \pm 0.005}$ | $0.0370 \pm 0.0030$ | $\mathbf{4.15}$ |
| | | No prior-mean shrinkage | $0.171 \pm 0.006$ | $0.044 \pm 0.004$ | $4.70$ |
| | | No covariance conditioning | $0.164 \pm 0.005$ | $0.0395 \pm 0.0031$ | $4.35$ |
| | | No stability regularizer | $0.160 \pm 0.005$ | $0.082 \pm 0.009$ | $7.25$ |
| | $d = 25$ | Fixed-prior PBML-LTI | $10.311 \pm 0.501$ | $0.0793 \pm 0.0325$ | $14.00$ |
| | | Type-II ML / empirical Bayes | $0.4790 \pm 0.0649$ | $\mathbf{0.0520 \pm 0.0176}$ | $14.00$ |
| | | PBML-LTI | $\mathbf{0.029 \pm 0.001}$ | $0.080 \pm 0.052$ | $\mathbf{6.05}$ |
| | | No prior-mean shrinkage | $0.035 \pm 0.002$ | $0.105 \pm 0.048$ | $7.10$ |
| | | No covariance conditioning | $0.0315 \pm 0.0015$ | $0.078 \pm 0.035$ | $6.45$ |
| | | No stability regularizer | $0.0305 \pm 0.0015$ | $0.195 \pm 0.085$ | $10.25$ |
| | $d = 10$ | Fixed-prior PBML-LTI | $1.6256 \pm 0.9888$ | $49.343 \pm 18.168$ | $\mathbf{13.95}$ |
| | | Type-II ML / empirical Bayes | $1.0909 \pm 0.3332$ | $\mathbf{43.915 \pm 48.789}$ | $14.00$ |
| | | PBML-LTI | $\mathbf{0.139 \pm 0.046}$ | $57.919 \pm 144.808$ | $14.35$ |
| | | No prior-mean shrinkage | $0.155 \pm 0.049$ | $88.5 \pm 125$ | $15.60$ |
| | | No covariance conditioning | $0.147 \pm 0.048$ | $61.2 \pm 92$ | $14.55$ |
| | | No stability regularizer | $0.145 \pm 0.047$ | $425 \pm 980$ | $18.20$ |

remove each auxiliary penalty one at a time relative to full PBML-LTI. The "No prior-mean shrinkage"

variant removes the shrinkage penalty on the shared prior mean $W$. The "No covariance conditioning" variant removes the conditioning penalty on the prior covariance $V$. The "No stability regularizer" variant removes the spectral stability penalty $\mathcal{R}_{\mathrm{stab}}$. These regularizers are not part of the formal PAC-Bayes theorem and are not claimed to enforce Assumption 1 directly. Instead, they act as practical optimization and prior-shaping devices. In particular, the stability regularizer acts on the learned shared prior mean, whereas Assumption 1 is a data-generating condition on the true task matrices $A_m$. Thus, the stability regularizer can encourage posterior adaptation toward dynamically well-behaved regions, but it does not guarantee that every posterior mean or every true task matrix satisfies the controlled-growth condition.

Table 4 reports the combined ablation study across stable and unstable regimes in the common-case setting. The results show that PBML-LTI substantially improves transition-matrix recovery relative to the fixed-prior and type-II empirical-Bayes variants across the reported settings. This supports the importance of both learning a transferable prior and using the PAC-Bayes-motivated fit–KL surrogate.

The regularization ablations show a different pattern. In stable regimes, removing any single regularizer changes the metrics only mildly, suggesting that PBML-LTI's stable-regime gains are not driven primarily by these auxiliary penalties. In unstable regimes, however, the regularizers become more important for robust rollout behavior. Removing the stability regularizer has only a modest effect on $E_A$, but it can substantially increase $E_{\mathrm{traj}}$ and the selected support length. This supports our interpretation that the stability penalty is mainly a spectral-stability and optimization device, rather than a direct mechanism for reducing Frobenius transition-matrix error.

## 6 Discussion

### 6.1 Sensitivity to the Training Temperature $\lambda_{\mathrm{tr}}$

The PAC-Bayes parameter $\lambda_{\mathrm{PB}}$ in the theoretical bound is restricted to $\lambda_{\mathrm{PB}} \in (0, 1]$. Separately, in implementation one can introduce a training temperature $\lambda_{\mathrm{tr}} > 0$ that rescales the fit–KL tradeoff used during meta-training. This implementation-level temperature should not be interpreted as extending the formal PAC-Bayes guarantee beyond $\lambda_{\mathrm{PB}} \in (0, 1]$. Instead, values $\lambda_{\mathrm{tr}} > 1$ are useful as empirical stress tests of more data-driven adaptation, while values $\lambda_{\mathrm{tr}} < 1$ correspond to stronger shrinkage toward the learned prior.

We use $\lambda_{\mathrm{tr}} = 1$ in the main experiments because it gives the canonical fit–KL objective and preserves the standard conjugate Bayesian interpretation. To assess robustness, we evaluate PBML-LTI over

$$\lambda_{\mathrm{tr}} \in \{10,\ 5,\ 1,\ 0.5,\ 0.1\}$$

on the stable synthetic benchmark with state dimension $d = 50$ in the common-case setting ($\rho_0 = 0.95$). Table 5 reports the resulting transition-matrix error $E_A$, rollout error $E_{\mathrm{traj}}$, and average selected support length $\overline{S}$.

Table 5: Sensitivity of PBML-LTI to the implementation-level training temperature $\lambda_{\mathrm{tr}}$ on the stable synthetic benchmark with $d = 50$ in the common-case setting ($\rho_0 = 0.95$). Values $\lambda_{\mathrm{tr}} > 1$ are empirical stress tests and are not part of the formal PAC-Bayes guarantee, which is stated for $\lambda_{\mathrm{PB}} \in (0, 1]$. Lower is better; best values are in **bold**.

| | $\lambda_{\mathrm{tr}} = 10$ | $\lambda_{\mathrm{tr}} = 5$ | $\lambda_{\mathrm{tr}} = 1$ | $\lambda_{\mathrm{tr}} = 0.5$ | $\lambda_{\mathrm{tr}} = 0.1$ |
|---|---|---|---|---|---|
| $E_A$ | $0.4612 \pm 0.0091$ | $0.4424 \pm 0.0090$ | $\mathbf{0.2068 \pm 0.0052}$ | $0.2845 \pm 0.0062$ | $0.3046 \pm 0.0063$ |
| $E_{\mathrm{traj}}$ | $0.0259 \pm 0.0019$ | $0.0258 \pm 0.0018$ | $\mathbf{0.0257 \pm 0.0018}$ | $0.0258 \pm 0.0018$ | $0.0258 \pm 0.0018$ |
| $\overline{S}$ | $3.80$ | $3.55$ | $\mathbf{1.35}$ | $2.05$ | $3.05$ |

The rollout metric $E_{\mathrm{traj}}$ is stable across a broad range of training temperatures, whereas transition-matrix recovery and adaptive support efficiency are more sensitive. Among the tested values, $\lambda_{\mathrm{tr}} = 1$ gives the best overall trade-off, achieving the lowest $E_A$, essentially the best $E_{\mathrm{traj}}$, and the shortest average support length $\overline{S}$. This supports our default implementation choice while keeping the formal PAC-Bayes guarantee restricted to $\lambda_{\mathrm{PB}} \in (0, 1]$.

## 6.2 Prior Quality and Sensitivity to the Number of Training Tasks

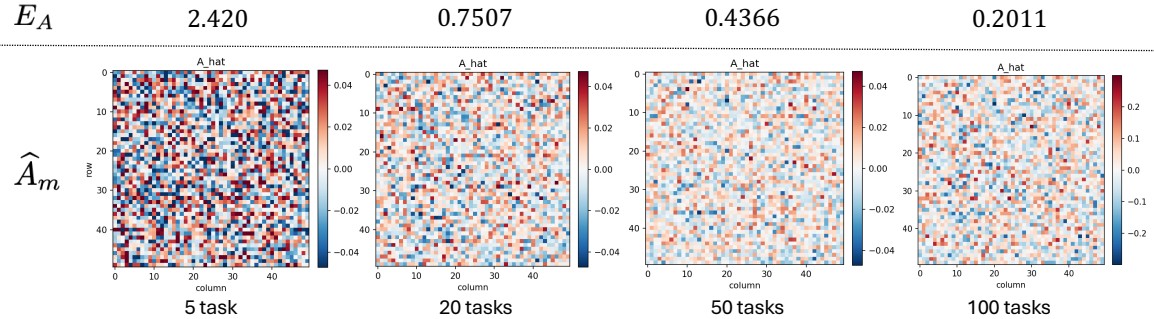

Figure 5: Sensitivity of the learned prior to the number of meta-training tasks. The figure visualizes the estimated transition matrices and the corresponding estimation errors for varying numbers of available training tasks. All tasks are sampled from stable systems with spectral radius $\rho_0 = 0.95$ and dimension $d = 50$.

In PBML-LTI, task-level uncertainty is quantified by the posterior column covariance $V_m$ in equation 12. The fidelity of this uncertainty estimate therefore depends on how accurately the meta-training stage recovers the shared parameters $(W, V, \sigma^2)$. At the same time, the PAC-Bayes analysis in Section 4.5 should be interpreted as conditional on a learned prior: it clarifies how prior quality affects predictive adaptation through the empirical-fit/KL tradeoff, but it does not provide a second-stage bound on how the quality of the learned prior improves as the number of meta-training tasks grows. We therefore examine that question empirically in Figure 5.

As shown in Figure 5, when only five training tasks are available, the resulting estimates $\widehat{A}_m$ are visibly more erratic, suggesting that the learned prior does not yet provide sufficiently reliable shrinkage or uncertainty information. As the number of training tasks increases, the learned prior covariance $V$ becomes a more stable estimate of cross-task variability, and the induced posteriors produce transition estimates that are progressively closer to the ground-truth dynamics. Quantitatively, increasing the number of training tasks from five to one hundred reduces the transition-matrix estimation error by approximately 90%. This sensitivity study therefore supports the interpretation that richer meta-level data improves prior quality and, in turn, improves downstream few-shot adaptation.

## 6.3 Empirical Magnitude of the PAC-Bayes-Derived Bounds

To complement the theoretical analysis in Section 4.5, we report empirical magnitudes of the PAC-Bayes-derived matrix and trajectory bounds on synthetic benchmarks. In addition to the stable and strongly unstable settings, we include a marginally unstable regime, in which the spectral-radius parameter is set to $\rho_0 = 1 + \epsilon$ with $\epsilon \in (0, 1)$, to examine whether the derived bounds vary continuously as the dynamics cross the nominal stability threshold $\rho = 1$. This setting places the task dynamics just beyond the stability boundary and emulates the near-critical operating conditions—slowly drifting or near-integrator modes—that are common in practical LTI applications. The diagnostic is useful because the theory depends on finite-horizon growth and posterior predictive fit, rather than on a discontinuous distinction between $\rho < 1$ and $\rho > 1$.

Table 6 shows a clear ordering across the three regimes. For every dimension and both evaluation subsets, the derived bounds increase from stable to marginally unstable to strongly unstable. In the stable setting, both $A_{\text{bound}}$ and $\text{Traj}_{\text{bound}}$ are moderate in magnitude and nearly unchanged between common-case and edge-case subsets. They also decrease with dimension, consistent with the fact that lower-dimensional stable systems are easier to identify from the same support budget.

The marginally unstable regime provides a continuity check around the stability boundary. Its bounds are only mildly larger than those in the stable regime: the matrix bound increases by less than 10%, while

the trajectory bound increases by roughly 20–25% across dimensions. The qualitative behavior remains the same as in the stable regime: the bounds remain moderate, decrease as the dimension decreases, and are similar between common-case and edge-case subsets. Thus, crossing the nominal threshold $\rho = 1$ does not by itself make the PAC-Bayes-derived quantities uninformative. Instead, the degradation is gradual and governed by the finite-horizon growth of the learned dynamics. This is precisely the continuity predicted by the support-to-query analysis of Section 4.5: the alignment lower bound of Remark 2 scales as $\rho^{2K_m}$ and the rollout factor $H_{m,K_m}$ is continuous in the spectral radius of the learned dynamics, so for $\rho_0 = 1 + \epsilon$, $\epsilon \in (0, 1)$ at the experimental horizons both remain controlled, exploding only under strong instability.[7]

Table 6: Empirical values of the PAC-Bayes-derived matrix and trajectory bounds for PBML-LTI across the three synthetic regimes: stable ($\rho_0 = 0.95$), marginally unstable ($\rho_0 = 1 + \epsilon$, $\epsilon \in (0, 1)$), and strongly unstable ($\rho_0 = 4.95$).

| Regime | Dimension | Setting | $A_{\text{bound}}$ | $\text{Traj}_{\text{bound}}$ |
|---|---|---|---|---|
| Stable | $d = 50$ | Common-case | $0.410 \pm 0.059$ | $17.744 \pm 2.660$ |
| | | Edge-case | $0.418 \pm 0.063$ | $18.133 \pm 2.781$ |
| | $d = 25$ | Common-case | $0.172 \pm 0.028$ | $9.473 \pm 1.547$ |
| | | Edge-case | $0.178 \pm 0.019$ | $9.757 \pm 1.051$ |
| | $d = 10$ | Common-case | $0.066 \pm 0.008$ | $4.550 \pm 0.622$ |
| | | Edge-case | $0.066 \pm 0.011$ | $4.553 \pm 0.797$ |
| Marginally unstable | $d = 50$ | Common-case | $0.447 \pm 0.073$ | $21.128 \pm 3.450$ |
| | | Edge-case | $0.442 \pm 0.072$ | $20.906 \pm 3.446$ |
| | $d = 25$ | Common-case | $0.184 \pm 0.027$ | $11.719 \pm 1.671$ |
| | | Edge-case | $0.187 \pm 0.039$ | $11.880 \pm 2.504$ |
| | $d = 10$ | Common-case | $0.068 \pm 0.013$ | $5.674 \pm 1.141$ |
| | | Edge-case | $0.068 \pm 0.011$ | $5.704 \pm 0.975$ |
| Unstable | $d = 50$ | Common-case | $1.077 \pm 0.143$ | $528.807 \pm 70.972$ |
| | | Edge-case | $1.100 \pm 0.124$ | $539.841 \pm 60.204$ |
| | $d = 25$ | Common-case | $0.748 \pm 0.175$ | $1278.494 \pm 302.805$ |
| | | Edge-case | $0.770 \pm 0.135$ | $1306.887 \pm 227.728$ |
| | $d = 10$ | Common-case | $1.277 \pm 0.143$ | $3717.382 \pm 618.144$ |
| | | Edge-case | $1.371 \pm 0.344$ | $3832.271 \pm 628.807$ |

In contrast, the strongly unstable regime produces much looser bounds, especially for trajectory rollout. This reinforces the interpretation of the unstable experiments as finite-horizon stress tests. The PAC-Bayes predictive-risk statement may remain formally valid under the stated MGF condition, but the rollout consequence can become loose or effectively vacuous because it contains the finite-horizon growth factor

$$H_{m,K_m} = \sum_{r=0}^{K_m - 1} \|\widehat{A}_m\|_2^r.$$

When the learned dynamics are strongly unstable, the powers $\|\widehat{A}_m\|_2^r$ can grow rapidly, so even a finite predictive bound can be substantially amplified under rollout. Consistent with this mechanism, the trajectory bound is largest in the strongly unstable low-dimensional setting, where the learned dynamics exhibit the strongest finite-horizon amplification. By contrast, the corresponding matrix bounds remain much closer in magnitude across dimensions, indicating that the main source of looseness is rollout amplification rather than the one-step matrix-error consequence alone.

Overall, the table supports two conclusions. First, the PAC-Bayes-derived bounds behave continuously around the stability threshold: the marginally unstable regime is a mild degradation of the stable regime rather than a qualitative breakdown. Second, strong instability can make the trajectory bound loose because

---

[7]Additional marginal unstable system experiment results are included in Appendix B.6.

finite-horizon spectral amplification magnifies the predictive-risk bound. Across all three regimes, the differences between common-case and edge-case subsets are small relative to the differences between stability regimes, suggesting that the degree of instability is the dominant factor controlling the practical tightness of the derived bounds in this experiment.

## 7 Conclusion

We studied few-shot identification of linear time-invariant dynamical systems in a meta-learning setting, where each task provides only a short, temporally dependent trajectory and reliable adaptation to previously unseen systems is required. Our proposed PBML-LTI framework learns a transferable prior over task-specific dynamics from multiple related systems and performs task-level adaptation through closed-form Bayesian inference, yielding both accurate point estimates and principled uncertainty quantification while remaining computationally efficient.

On the theoretical side, we developed a martingale PAC-Bayes analysis tailored to trajectory data with within-task temporal dependence. The resulting support–query statement should be interpreted as a support-conditioned predictive-risk diagnostic: it clarifies the role of the learned prior and posterior complexity, while still depending on the empirical query fit on the held-out suffix. We further characterized when this empirical query term is expected to be controlled, through support-to-query transfer conditions involving excitation alignment, posterior concentration, learned-prior quality, and finite-horizon stability. Thus, the theory provides a PAC-Bayes motivation for the fit–KL surrogate and a support–query predictive-risk perspective for few-shot adaptation, rather than a direct end-to-end guarantee for the full experimental pipeline. We also showed how the support–query predictive bound induces problem-specific corollaries for projected transition-matrix error, full Frobenius recovery under query excitation, and finite-horizon rollout error.

Empirically, PBML-LTI consistently improves data efficiency and robustness in challenging regimes, including high-dimensional synthetic systems and real fMRI-derived LTI tasks, where linear dynamics only hold approximately. In particular, the proposed method achieves strong multi-step prediction performance while adapting from limited support data, highlighting its suitability for few-shot system identification.

Several directions remain for future work. Extending the framework to controlled systems, partial observability, or nonlinear dynamics would broaden its applicability. Another important direction is to develop an explicit second-stage meta-generalization analysis that quantifies how the quality of the learned prior improves with the number of meta-training tasks. Incorporating richer structured priors is also a natural avenue for further research.

**Code and Data Availability** The code repository is available at https://github.com/chenfeng-huang/PBML-LTI-TMLR-2026. The synthetic dataset generation code is included in the code repository. The fMRI dataset is available at https://openneuro.org/datasets/ds000244/versions/1.0.0, provided by Pinho et al. (2018).

### Acknowledgments

The work of GM was partially supported by NSF grants ATD 2319552 and DMS 2348640.

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

## Table of Contents

## A     Technical Appendix

### A.1     Proof of Theorem 1

*Proof.* Under the Gaussian conditional model

$$p_\sigma(x_t \mid x_{t-1}, A) = \mathcal{N}(Ax_{t-1}, \sigma^2 I_d), \tag{68}$$

the conditional density is given by

$$p_\sigma(x_t \mid x_{t-1}, A) = (2\pi\sigma^2)^{-d/2} \exp\left(-\frac{1}{2\sigma^2}\|x_t - Ax_{t-1}\|_2^2\right). \tag{69}$$

Taking the negative of the logarithm yields

$$\ell_t(A) = \frac{1}{2\sigma^2}\|x_t - Ax_{t-1}\|_2^2 + \frac{d}{2}\log(2\pi\sigma^2), \tag{70}$$

which proves equation 16.

Next, fix any $A$ and define

$$d_t(A) := \mathbb{E}[\ell_t(A) \mid \mathcal{F}_{t-1}] - \ell_t(A). \tag{71}$$

By assumption, $\ell_t(A)$ is $\mathcal{F}_t$-measurable and integrable. Therefore,

$$\mathbb{E}[d_t(A) \mid \mathcal{F}_{t-1}] = 0, \tag{72}$$

which shows that $\{d_t(A)\}_{t=1}^T$ is a martingale difference sequence. $\qquad\square$

### A.2 Proof of Theorem 2

*Proof.* Fix $A$ and $\lambda_{\mathrm{PB}} \in (0, 1]$. Define

$$S_t(A) := \sum_{i=1}^{t} d_i(A), \qquad S_0(A) = 0, \tag{73}$$

and

$$Z_t(A; \lambda_{\mathrm{PB}}) := \exp\left(\lambda_{\mathrm{PB}} S_t(A) - \sum_{i=1}^{t} \psi_i(\lambda_{\mathrm{PB}})\right), \qquad Z_0(A; \lambda_{\mathrm{PB}}) = 1. \tag{74}$$

Since $S_t(A) = S_{t-1}(A) + d_t(A)$,

$$Z_t(A; \lambda_{\mathrm{PB}}) = Z_{t-1}(A; \lambda_{\mathrm{PB}}) \exp(\lambda_{\mathrm{PB}} d_t(A) - \psi_t(\lambda_{\mathrm{PB}})). \tag{75}$$

Taking conditional expectation and using the MGF condition gives

$$\mathbb{E}[Z_t(A; \lambda_{\mathrm{PB}}) \mid \mathcal{F}_{t-1}] \leq Z_{t-1}(A; \lambda_{\mathrm{PB}}), \tag{76}$$

so $\{Z_t(A; \lambda_{\mathrm{PB}})\}$ is a nonnegative supermartingale. $\qquad\square$

**Lemma 2** (Donsker–Varadhan change of measure). *Let $P$ and $Q$ be probability measures on the same measurable space. If $Q \ll P$, then for any measurable function $f$ for which the expressions are well defined,*

$$\mathbb{E}_{A \sim Q}[f(A)] \leq D_{\mathrm{KL}}(Q\|P) + \log \mathbb{E}_{A \sim P}\left[e^{f(A)}\right]. \tag{77}$$

*If $Q \not\ll P$, then $D_{\mathrm{KL}}(Q\|P) = +\infty$ and the inequality holds trivially.*

*Proof.* Assume $Q \ll P$ and let $r := dQ/dP$. Then

$$D_{\mathrm{KL}}(Q\|P) = \mathbb{E}_Q[\log r].$$

By Jensen's inequality,

$$\begin{aligned}
\mathbb{E}_Q[f(A)] - D_{\mathrm{KL}}(Q\|P) &= \mathbb{E}_Q[f(A) - \log r(A)] \\
&= \mathbb{E}_Q\left[\log\left(\frac{e^{f(A)}}{r(A)}\right)\right] \\
&\leq \log \mathbb{E}_Q\left[\frac{e^{f(A)}}{r(A)}\right] = \log \int e^{f(a)} \, dP(a).
\end{aligned} \tag{78}$$

Rearranging gives the result. $\qquad\square$

### A.3 Proof of Theorem 3

*Proof.* Fix $\lambda_{\mathrm{PB}} \in (0, 1]$. Define the mixture supermartingale

$$\overline{Z}_t(\lambda_{\mathrm{PB}}) := \mathbb{E}_{A \sim P}[Z_t(A; \lambda_{\mathrm{PB}})]. \tag{79}$$

Then

$$\mathbb{E}[\overline{Z}_t(\lambda) \mid \mathcal{F}_{t-1}] \leq \overline{Z}_{t-1}(\lambda), \tag{80}$$

and hence

$$\mathbb{E}[\overline{Z}_T(\lambda)] \le 1. \tag{81}$$

By Markov's inequality, with probability at least $1 - \delta$,

$$\overline{Z}_T(\lambda) \le \frac{1}{\delta}. \tag{82}$$

Applying Lemma 2 yields

$$\mathbb{E}_{A \sim Q}[S_T(A)] \le \frac{1}{\lambda} \left( D_{\mathrm{KL}}(Q\|P) + \log \frac{1}{\delta} + \sum_{t=1}^{T} \psi_t(\lambda) \right). \tag{83}$$

Substituting

$$S_T(A) = \sum_{t=1}^{T} \Big( \mathbb{E}[\ell_t(A) \mid \mathcal{F}_{t-1}] - \ell_t(A) \Big) \tag{84}$$

and rearranging gives equation 20. □

### A.4  Proof of Corollary 1

*Proof.* Under Assumption 4,

$$\sum_{t=1}^{T} \psi_t(\lambda_{\mathrm{PB}}) \le \frac{\lambda_{\mathrm{PB}}^2 vT}{2}. \tag{85}$$

Substituting this into Theorem 3 yields

$$L_T(Q) \le \widehat{L}_T(Q) + \frac{D_{\mathrm{KL}}(Q\|P) + \log(1/\delta)}{\lambda_{\mathrm{PB}}T} + \frac{\lambda_{\mathrm{PB}} v}{2}. \tag{86}$$

This proves equation 30.

Now let

$$a := D_{\mathrm{KL}}(Q\|P) + \log(1/\delta), \qquad g(\lambda_{\mathrm{PB}}) := \frac{a}{\lambda_{\mathrm{PB}}T} + \frac{\lambda_{\mathrm{PB}} v}{2}, \qquad \lambda_{\mathrm{PB}} \in (0, 1].$$

Differentiating with respect to $\lambda_{\mathrm{PB}}$ gives

$$g'(\lambda_{\mathrm{PB}}) = -\frac{a}{\lambda_{\mathrm{PB}}^2 T} + \frac{v}{2}, \qquad g''(\lambda_{\mathrm{PB}}) = \frac{2a}{\lambda_{\mathrm{PB}}^3 T} > 0.$$

Thus, $g$ is strictly convex on $(0, \infty)$. Its unconstrained minimizer satisfies

$$g'(\lambda_{\mathrm{PB}}) = 0 \quad \Longrightarrow \quad \lambda_{\mathrm{unc}} = \sqrt{\frac{2a}{vT}}.$$

Therefore, the minimizer over the constrained interval $(0, 1]$ is

$$\lambda^\star = \min\left\{ 1, \sqrt{\frac{2a}{vT}} \right\}. \tag{87}$$

In the regime where $\lambda^\star < 1$, we have

$$\lambda^\star = \sqrt{\frac{2a}{vT}},$$

and substituting this into equation 86 gives

$$L_T(Q) \le \widehat{L}_T(Q) + \sqrt{\frac{2va}{T}}, \tag{88}$$

which is exactly equation 32. □

## A.5 Proof of Corollary 2

*Proof.* Fix a task $m$ and write

$$\mathcal{G}_u := \mathcal{F}_{m, S_m + u}, \qquad u = 0, \ldots, K_m.$$

Condition on the support sigma-field $\mathcal{G}_0 = \mathcal{F}_{m, S_m}$. Under this conditioning, the posterior

$$Q_{m,\phi}^{\text{sup}} = p(A_m \mid D_m^{\text{sup}}, \phi)$$

is fixed, and the prior $P_\phi$ is independent of the future query suffix. We apply Corollary 1 to the shifted query sequence.

Define, for $u = 1, \ldots, K_m$,

$$\widetilde{\ell}_u(A) := -\log p_\sigma(x_{m, S_m + u} \mid x_{m, S_m + u - 1}, A). \tag{89}$$

By Theorem 1,

$$\widetilde{\ell}_u(A) = \frac{1}{2\sigma^2} \|x_{m, S_m + u} - A x_{m, S_m + u - 1}\|_2^2 + \frac{d}{2} \log(2\pi\sigma^2). \tag{90}$$

Since Assumption 4 is inherited by the shifted filtration $\{\mathcal{G}_u\}_{u=0}^{K_m}$, Corollary 1 gives, with conditional probability at least $1 - \delta$ given $\mathcal{G}_0$,

$$\frac{1}{K_m} \sum_{u=1}^{K_m} \mathbb{E}_{A \sim Q_{m,\phi}^{\text{sup}}} \left[ \mathbb{E}\left[ \widetilde{\ell}_u(A) \mid \mathcal{G}_{u-1} \right] \right] \leq \frac{1}{K_m} \sum_{u=1}^{K_m} \mathbb{E}_{A \sim Q_{m,\phi}^{\text{sup}}} \left[ \widetilde{\ell}_u(A) \right] + \frac{D_{\text{KL}}(Q_{m,\phi}^{\text{sup}} \| P_\phi) + \log(1/\delta)}{\lambda_{\text{PB}} K_m} + \frac{\lambda_{\text{PB}} v}{2}. \tag{91}$$

Substituting equation 90 into equation 91, the left-hand side becomes

$$\frac{1}{K_m} \sum_{u=1}^{K_m} \mathbb{E}_{A \sim Q_{m,\phi}^{\text{sup}}} \left[ \mathbb{E}\left[ \widetilde{\ell}_u(A) \mid \mathcal{G}_{u-1} \right] \right]$$

$$= \frac{1}{2\sigma^2} \frac{1}{K_m} \sum_{u=1}^{K_m} \mathbb{E}_{A \sim Q_{m,\phi}^{\text{sup}}} \left[ \mathbb{E}\left[ \|x_{m, S_m + u} - A x_{m, S_m + u - 1}\|_2^2 \mid \mathcal{G}_{u-1} \right] \right] + \frac{d}{2} \log(2\pi\sigma^2)$$

$$= \frac{1}{2\sigma^2} R_{m,q}^{\text{pred}}(Q_{m,\phi}^{\text{sup}}) + \frac{d}{2} \log(2\pi\sigma^2), \tag{92}$$

by definition equation 34. Similarly, the right-hand side becomes

$$\frac{1}{K_m} \sum_{u=1}^{K_m} \mathbb{E}_{A \sim Q_{m,\phi}^{\text{sup}}} \left[ \widetilde{\ell}_u(A) \right]$$

$$= \frac{1}{2\sigma^2} \frac{1}{K_m} \sum_{u=1}^{K_m} \mathbb{E}_{A \sim Q_{m,\phi}^{\text{sup}}} \left[ \|x_{m, S_m + u} - A x_{m, S_m + u - 1}\|_2^2 \right] + \frac{d}{2} \log(2\pi\sigma^2)$$

$$= \frac{1}{2\sigma^2} \widehat{R}_{m,q}^{\text{pred}}(Q_{m,\phi}^{\text{sup}}) + \frac{d}{2} \log(2\pi\sigma^2), \tag{93}$$

since

$$\sum_{u=1}^{K_m} \|x_{m, S_m + u} - A x_{m, S_m + u - 1}\|_2^2 = \|Y_{m,q} - A X_{m,q}\|_F^2.$$

Cancelling the common additive term $\frac{d}{2} \log(2\pi\sigma^2)$ from both sides and multiplying by $2\sigma^2$ yields

$$R_{m,q}^{\text{pred}}(Q_{m,\phi}^{\text{sup}}) \leq \widehat{R}_{m,q}^{\text{pred}}(Q_{m,\phi}^{\text{sup}}) + \frac{2\sigma^2}{\lambda_{\text{PB}} K_m} \left( D_{\text{KL}}(Q_{m,\phi}^{\text{sup}} \| P_\phi) + \log(1/\delta) \right) + \sigma^2 \lambda_{\text{PB}} v$$

which is exactly equation 36. $\qquad\square$

### A.6 Support-to-Query Transfer Proof Details

*Proof of Proposition 1.* By the closed form equation 14 applied to the query pair,

$$\widehat{R}_{m,q}^{\text{pred}}(Q_{m,\phi}^{\text{sup}}) = \frac{1}{K_m}\|Y_{m,q} - M_m X_{m,q}\|_F^2 + d\operatorname{tr}(V_m\widehat{\Sigma}_q).$$

Using $Y_{m,q} = A_m X_{m,q} + \Xi_{m,q}$ and $\|U + V\|_F^2 \leq 2\|U\|_F^2 + 2\|V\|_F^2$, we obtain

$$\frac{1}{K_m}\|Y_{m,q} - M_m X_{m,q}\|_F^2 \leq 2\operatorname{tr}\big((A_m - M_m)\widehat{\Sigma}_q(A_m - M_m)^\top\big) + \frac{2}{K_m}\|\Xi_{m,q}\|_F^2.$$

If $\widehat{\Sigma}_q \preceq \gamma_m \widehat{\Sigma}_{\text{sup}}$, then for any matrix $B$,

$$\operatorname{tr}(B\widehat{\Sigma}_q B^\top) \leq \gamma_m \operatorname{tr}(B\widehat{\Sigma}_{\text{sup}} B^\top),$$

and also

$$\operatorname{tr}(V_m\widehat{\Sigma}_q) \leq \gamma_m \operatorname{tr}(V_m\widehat{\Sigma}_{\text{sup}}).$$

Taking $B = A_m - M_m$ gives

$$\operatorname{tr}\big((A_m - M_m)\widehat{\Sigma}_q(A_m - M_m)^\top\big) \leq \gamma_m \frac{1}{S_m}\|(A_m - M_m)X_m^{\text{sup}}\|_F^2.$$

Since

$$Y_m^{\text{sup}} = A_m X_m^{\text{sup}} + \Xi_m^{\text{sup}},$$

we have

$$(A_m - M_m)X_m^{\text{sup}} = (Y_m^{\text{sup}} - M_m X_m^{\text{sup}}) - \Xi_m^{\text{sup}},$$

and hence

$$\frac{1}{S_m}\|(A_m - M_m)X_m^{\text{sup}}\|_F^2 \leq \frac{2}{S_m}\|Y_m^{\text{sup}} - M_m X_m^{\text{sup}}\|_F^2 + \frac{2}{S_m}\|\Xi_m^{\text{sup}}\|_F^2.$$

Combining the preceding inequalities yields

$$\widehat{R}_{m,q}^{\text{pred}}(Q_{m,\phi}^{\text{sup}}) \leq \frac{4\gamma_m}{S_m}\|Y_m^{\text{sup}} - M_m X_m^{\text{sup}}\|_F^2 + \gamma_m d\operatorname{tr}(V_m\widehat{\Sigma}_{\text{sup}}) + \frac{4\gamma_m}{S_m}\|\Xi_m^{\text{sup}}\|_F^2 + \frac{2}{K_m}\|\Xi_{m,q}\|_F^2.$$

Since

$$\widehat{R}_{\text{sup}}^{\text{pred}} = \frac{1}{S_m}\|Y_m^{\text{sup}} - M_m X_m^{\text{sup}}\|_F^2 + d\operatorname{tr}(V_m\widehat{\Sigma}_{\text{sup}}),$$

we can upper bound the first two terms by $4\gamma_m \widehat{R}_{\text{sup}}^{\text{pred}}$, proving equation 39.

For the posterior-width cap, recall from equation 12 that

$$V_m^{-1} = V^{-1} + \frac{S_m}{\sigma^2}\widehat{\Sigma}_{\text{sup}}.$$

Thus $V_m^{-1} \succeq V^{-1}$, which implies $V_m \preceq V$, and $V_m^{-1} \succeq (S_m/\sigma^2)\widehat{\Sigma}_{\text{sup}}$. Therefore

$$\operatorname{tr}(V_m\widehat{\Sigma}_{\text{sup}}) \leq \operatorname{tr}(V\widehat{\Sigma}_{\text{sup}})$$

and

$$\operatorname{tr}(V_m\widehat{\Sigma}_{\text{sup}}) \leq \frac{\sigma^2}{S_m}\operatorname{tr}(V_m V_m^{-1}) = \frac{\sigma^2 d}{S_m}.$$

Combining this with $\operatorname{tr}(V_m\widehat{\Sigma}_q) \leq \gamma_m \operatorname{tr}(V_m\widehat{\Sigma}_{\text{sup}})$ proves equation 40. $\qquad\square$

*Proof of Proposition 2.* Since $\widehat{\Sigma}_{\text{sup}} + \alpha I_d \succ 0$, the definition of $\gamma_{m,\alpha}$ is equivalent to

$$\widehat{\Sigma}_q \preceq \gamma_{m,\alpha}(\widehat{\Sigma}_{\text{sup}} + \alpha I_d).$$

Repeating the proof of Proposition 1 with $\widehat{\Sigma}_{\text{sup}} + \alpha I_d$ in place of $\widehat{\Sigma}_{\text{sup}}$ gives

$$\operatorname{tr}\big((A_m - M_m)\widehat{\Sigma}_q(A_m - M_m)^\top\big) \leq \gamma_{m,\alpha}\operatorname{tr}\big((A_m - M_m)\widehat{\Sigma}_{\text{sup}}(A_m - M_m)^\top\big) + \gamma_{m,\alpha}\alpha\|A_m - M_m\|_F^2$$

and

$$\operatorname{tr}(V_m\widehat{\Sigma}_q) \leq \gamma_{m,\alpha}\operatorname{tr}(V_m\widehat{\Sigma}_{\text{sup}}) + \gamma_{m,\alpha}\alpha\operatorname{tr}(V_m).$$

Combining these inequalities with the same support residual identity as above yields

$$\widehat{R}_{m,q}^{\text{pred}}(Q_{m,\phi}^{\text{sup}}) \leq 4\gamma_{m,\alpha}\widehat{R}_{\text{sup}}^{\text{pred}} + \frac{4\gamma_{m,\alpha}}{S_m}\|\Xi_m^{\text{sup}}\|_F^2 + \frac{2}{K_m}\|\Xi_{m,q}\|_F^2 + 2\gamma_{m,\alpha}\alpha\|M_m - A_m\|_F^2 + \gamma_{m,\alpha}\alpha d\operatorname{tr}(V_m).$$

Since

$$\mathbb{E}_{A \sim Q_{m,\phi}^{\text{sup}}}\|A - A_m\|_F^2 = \|M_m - A_m\|_F^2 + d\operatorname{tr}(V_m),$$

the final two terms are upper bounded by

$$2\gamma_{m,\alpha}\alpha\mathbb{E}_{A \sim Q_{m,\phi}^{\text{sup}}}\|A - A_m\|_F^2,$$

which proves equation 42. □

**Remark 3** (Properties of the regularized alignment constant and control of the remainder term). (i) On the event $\{\gamma_m < \infty\}$, $\widehat{\Sigma}_q \preceq \gamma_m\widehat{\Sigma}_{\text{sup}} \preceq \gamma_m(\widehat{\Sigma}_{\text{sup}} + \alpha I_d)$ gives $\gamma_{m,\alpha} \leq \gamma_m$, and $\widehat{\Sigma}_q \preceq \lambda_{\max}(\widehat{\Sigma}_q)I_d \preceq (\lambda_{\max}(\widehat{\Sigma}_q)/\alpha)(\widehat{\Sigma}_{\text{sup}} + \alpha I_d)$ gives $\gamma_{m,\alpha} \leq \lambda_{\max}(\widehat{\Sigma}_q)/\alpha$; moreover $\alpha \mapsto \gamma_{m,\alpha}$ is nonincreasing, since $\alpha \mapsto \widehat{\Sigma}_{\text{sup}} + \alpha I_d$ is nondecreasing in the semidefinite order. With the prior-matched choice that replaces $\alpha I_d$ by $(\sigma^2/S_m)V^{-1}$, the regularized Gram equals $(\sigma^2/S_m)V_m^{-1}$ by equation 12, the corresponding alignment diagnostic is $\widetilde{\gamma}_m = (S_m/\sigma^2)\lambda_{\max}(V_m^{1/2}\widehat{\Sigma}_q V_m^{1/2})$, and the posterior-width contribution obeys the unconditional cap $d\operatorname{tr}(V_m\widehat{\Sigma}_q) \leq \widetilde{\gamma}_m\sigma^2 d^2/S_m$, since $\operatorname{tr}(V_m^{1/2}\widehat{\Sigma}_q V_m^{1/2}) \leq d\,\lambda_{\max}(V_m^{1/2}\widehat{\Sigma}_q V_m^{1/2})$. (ii) The remainder term in equation 42 decomposes as $\mathbb{E}_{A \sim Q_{m,\phi}^{\text{sup}}}\|A - A_m\|_F^2 = \|M_m - A_m\|_F^2 + d\operatorname{tr}(V_m)$, an observable posterior-width part plus a mean-error part. From equation 12–equation 13 and $Y_m^{\text{sup}} = A_m X_m^{\text{sup}} + \Xi_m^{\text{sup}}$,

$$M_m - A_m = \left[\frac{1}{\sigma^2}\Xi_m^{\text{sup}}(X_m^{\text{sup}})^\top + (W - A_m)V^{-1}\right]V_m,$$

so that

$$\|M_m - A_m\|_F^2 \leq \frac{2}{\sigma^4}\big\|\Xi_m^{\text{sup}}(X_m^{\text{sup}})^\top V_m\big\|_F^2 + 2\,\lambda_{\max}(V_m)\,\Delta_m(\phi),$$

using $V^{-1}V_m^2 V^{-1} \preceq \lambda_{\max}(V_m)V^{-1}$ (a consequence of $V_m \preceq V$) for the second term, with $\Delta_m(\phi)$ the learned-prior quality of equation 44. The first term is a self-normalized martingale quantity controlled by standard self-normalized concentration for dependent regressors (Abbasi-Yadkori et al., 2011). Thus, in weakly excited directions the remainder is controlled by posterior contraction and learned-prior quality, as claimed.

*Proof sketch of Lemma 1.* Let

$$Z_t := \|w_{m,t}\|_2^2 - \mathbb{E}[\|w_{m,t}\|_2^2 \mid \mathcal{F}_{m,t-1}].$$

Then $\{Z_t\}$ is a martingale difference sequence. The conditional sub-Gaussian assumption in Assumption 2 implies that $\|w_{m,t}\|_2^2$ is conditionally sub-exponential, with scale controlled by $\sigma_w^2 d$. Moreover,

$$\mathbb{E}[\|w_{m,t}\|_2^2 \mid \mathcal{F}_{m,t-1}] = \operatorname{tr}\big(\mathbb{E}[w_{m,t}w_{m,t}^\top \mid \mathcal{F}_{m,t-1}]\big) \leq \operatorname{tr}(\Sigma_w).$$

Applying a Bernstein–Freedman inequality for conditionally sub-exponential martingale differences (Freedman, 1975) gives the stated bound. □

*Proof of Corollary 3.* Corollary 2 holds with conditional probability at least $1 - \delta$ given $\mathcal{F}_{m,S_m}$, and hence unconditionally by the tower property. Lemma 1 is applied once to the support noise energy and once (conditionally on $\mathcal{F}_{m,S_m}$, hence again unconditionally) to the query noise energy. On the intersection of these three events, Proposition 1 bounds the empirical query term in Corollary 2. Substituting the two noise-energy concentration bounds into equation 39 yields equation 43. A union bound gives total failure probability at most $3\delta$. □

**Corollary chain for the matrix and rollout consequences.** Let $\mathcal{C}_m^{\mathrm{pred}}(\bar{\gamma}, \delta)$ denote the right-hand side of equation 43. The consequences equation 47, equation 49, and equation 50 use the support–query predictive bound only through the upper bound $\mathcal{B}_m^{\mathrm{pred}}$ on $R_{m,q}^{\mathrm{pred}}(Q_{m,\phi}^{\mathrm{sup}})$, so on the events of Corollary 3 the same chaining applies with $\mathcal{C}_m^{\mathrm{pred}}(\bar{\gamma}, \delta)$ in place of $\mathcal{B}_m^{\mathrm{pred}}$. Concretely, on the event $\widehat{\Sigma}_q \preceq \bar{\gamma} \widehat{\Sigma}_{\mathrm{sup}}$, with probability at least $1 - 3\delta$,

$$E_{A,\mathrm{proj}}(m) \leq \mathcal{C}_m^{\mathrm{pred}}(\bar{\gamma}, \delta), \qquad E_A(m) \leq \frac{\mathcal{C}_m^{\mathrm{pred}}(\bar{\gamma}, \delta)}{\kappa_m} \text{ under equation } 48,$$

$$\mathbb{E}[E_{\mathrm{traj}}(m) \mid \mathcal{F}_{m,S_m}] \leq K_m^2 \, H_{m,K_m}^2 \, \mathcal{C}_m^{\mathrm{pred}}(\bar{\gamma}, \delta).$$

Thus the projected transition-matrix, full Frobenius, and rollout consequences inherit the conditional support-to-query form: apart from the alignment level and noise constants, their right-hand sides are determined by the support-adapted posterior and the support prefix.

## A.7 Functional Magnetic Resonance Imaging (fMRI) Dataset

We construct a real-data LTI benchmark from the OpenNeuro dataset ds000244 (Individual Brain Charting) (Pinho et al., 2018). The dataset is publicly available at https://openneuro.org/datasets/ds000244/versions/1.0.0. The raw data follow the Brain Imaging Data Structure (BIDS) convention, together with metadata and tabular confound files.

**Schaefer parcellation.** To obtain a multivariate state sequence from each fMRI run, we convert voxel-level Blood-Oxygen-Level-Dependent (BOLD) signals into a $d$-dimensional ROI time series by averaging voxel signals within atlas-defined parcels using the Schaefer parcellation (Schaefer et al., 2018). Let $y_v(t)$ denote the BOLD signal at voxel $v$ and time index $t$. For parcel (ROI) $j$ with voxel set $\Omega_j$, we define the parcel-averaged signal

$$x_t[j] := \frac{1}{|\Omega_j|} \sum_{v \in \Omega_j} y_v(t), \qquad j = 1, \ldots, d, \tag{94}$$

yielding a multivariate state vector $x_t \in \mathbb{R}^d$ at each time $t$. In our experiments we use a Schaefer atlas with $d = 200$ parcels, so each fMRI run becomes a length-$T$ trajectory $\{x_t\}_{t=0}^T$ in $\mathbb{R}^{200}$.

**Run-level preprocessing.** For each run-level ROI time series, we apply standard time-series preprocessing to improve comparability across runs and to reduce nuisance variation. Concretely, we (i) standardize each ROI time series over time, (ii) regress out motion confounds when a confounds TSV is available, and (iii) mean-center the resulting ROI signals within each run. If we write the ROI matrix as $Z \in \mathbb{R}^{T \times d}$ with rows $x_t^\top$, and let $C \in \mathbb{R}^{T \times q}$ denote a subspace spanned by the confound regressors, we apply

$$Z \leftarrow \left(I - C(C^\top C)^\dagger C^\top\right) Z, \tag{95}$$

followed by per-run mean-centering $Z \leftarrow Z - \mathbf{1}\mu^\top$, where $\mu = \frac{1}{T} \sum_{t=1}^T x_t$.

**Window-level LTI task construction.** To obtain many related identification tasks from each run, we segment each ROI time series into overlapping temporal windows of fixed length. Each window is treated as a distinct task instance $m$ with states $\{x_{m,t}\}_{t=0}^{T_m}$, and we form the one-step regression matrices

$$X_m := [x_{m,0}, \ldots, x_{m,T_m-1}] \in \mathbb{R}^{d \times T_m}, \qquad Y_m := [x_{m,1}, \ldots, x_{m,T_m}] \in \mathbb{R}^{d \times T_m}. \tag{96}$$

These matrices define the window-level matrix regression view $Y_m \approx A_m X_m$ used throughout the paper. In our released split, each window has $d = 200$ and $T_m = 100$ transitions.

The resulting windows are *related* task instances because they are not drawn from arbitrary unrelated sources. Rather, they come from the same pool of subjects, sessions, and experimental conditions, and overlapping windows from the same run can be interpreted as nearby local dynamical regimes within a shared subject- and condition-specific environment. This gives rise to a heterogeneous but structured family of tasks, which is well matched to the meta-learning objective of PBML-LTI.

**Reference transition matrix for evaluation.** Unlike synthetic data, fMRI does not provide a physical ground-truth transition matrix. We therefore define a reference transition matrix $A_m^{\text{ref}}$ for each window by fitting a ridge-regularized one-step linear map:

$$A_m^{\text{ref}} \;:=\; \arg\min_{A \in \mathbb{R}^{d \times d}} \; \|Y_m - AX_m\|_F^2 \;+\; \lambda_{\text{ref}} \|A\|_F^2, \tag{97}$$

which has the closed-form solution

$$A_m^{\text{ref}} \;=\; Y_m X_m^\top \left(X_m X_m^\top + \lambda_{\text{ref}} I_d\right)^{-1}. \tag{98}$$

We emphasize that $A_m^{\text{ref}}$ is used only as a consistent evaluation reference to compute $E_A$; all methods are trained and adapted from trajectories $(X_m, Y_m)$.

**Train–test split and task labels.** We use a predefined train–test split constructed at the window level. Windows from different subjects, sessions, and experimental conditions are distributed across training and test according to this split. The original BIDS task labels are retained only as metadata for grouping and reporting, and are not used by the learning algorithms.

BIDS stands for *Brain Imaging Data Structure*, a standard format for organizing neuroimaging datasets. In this format, each fMRI run carries a `task` label identifying the experimental paradigm under which the data were collected. These labels therefore describe the source cognitive or behavioral condition of a run, rather than providing a supervisory target for learning. In our benchmark, they are used only to characterize the diversity of the data and to report how many experimental conditions are represented in the train/test split.

In the split used in our experiments, the training set contains 141 window-level task instances spanning 17 task labels, while the test set contains 14 window-level instances spanning 8 task labels. Across the full split, the task labels include:

- **Archi:** ArchiEmotional, ArchiSocial, ArchiSpatial, ArchiStandard.

- **HCP-style:** HcpEmotion, HcpGambling, HcpLanguage, HcpMotor, HcpRelational, HcpSocial, HcpWm.

- **RSVP language:** RSVPLanguage00, RSVPLanguage01, RSVPLanguage02, RSVPLanguage03, RSVPLanguage04, RSVPLanguage05.

# B  Additional Results

## B.1  Support–Query Diagnostics

The support–query PAC-Bayes bound in Corollary 2 contains the empirical query predictive term $\widehat{R}_{m,q}^{\text{pred}}$. This term is computed on the held-out query suffix and should not be interpreted as a support-only quantity. To assess when this term is expected to be small, we compare predictive errors on the support, inner validation, and held-out query portions of the trajectory.

All diagnostics are computed after meta-training and use the learned prior $P_\phi$ fixed. For each task, the held-out query suffix is never used to choose the adaptation prefix; it is used only for post-selection evaluation. Candidate prefixes are selected using only the fit-prefix posterior and the validation suffix inside the support window. Unless otherwise stated, reported group values are averages over tasks in the corresponding regime and dimension.

For the support–query diagnostics in Tables 7 and 8, we use the following notation. For a candidate prefix length $s$, let $Q_{m,\phi}^{(s)}$ denote the posterior formed by adapting the learned prior to the first $s$ transitions of task $m$, with parameters $(M_m(s), V_m(s))$ given by equation 12–equation 13. Let $(X_{m,\text{sup}}^{(s)}, Y_{m,\text{sup}}^{(s)})$, $(X_{m,\text{val}}, Y_{m,\text{val}})$, and $(X_{m,\text{qry}}, Y_{m,\text{qry}})$ denote the regression pairs of the fit prefix, the validation suffix inside the support window, and the held-out query suffix of length $K_m$. We define the posterior-predictive squared errors

$$\widehat{R}_{m,\text{sup}}^{\text{pred}}(s) := \frac{1}{s} \mathbb{E}_{A \sim Q_{m,\phi}^{(s)}} \left[ \|Y_{m,\text{sup}}^{(s)} - A X_{m,\text{sup}}^{(s)}\|_F^2 \right],$$

$$\widehat{R}^{\text{pred}}_{m,\text{val}}(s) := \frac{1}{T_{\text{val}}} \mathbb{E}_{A \sim Q^{(s)}_{m,\phi}} \left[ \|Y_{m,\text{val}} - A X_{m,\text{val}}\|^2_F \right], \qquad \widehat{R}^{\text{pred}}_{m,\text{qry}}(s) := \frac{1}{K_m} \mathbb{E}_{A \sim Q^{(s)}_{m,\phi}} \left[ \|Y_{m,\text{qry}} - A X_{m,\text{qry}}\|^2_F \right].$$

Thus, $\widehat{R}^{\text{pred}}_{m,\text{qry}}(s)$ instantiates the empirical predictive risk equation 35 at the posterior $Q^{(s)}_{m,\phi}$. By equation 14, each expectation evaluates in closed form as

$$\|Y^{\text{seg}} - M_m(s)X^{\text{seg}}\|^2_F + d \operatorname{tr}\big(V_m(s)\, X^{\text{seg}}(X^{\text{seg}})^\top\big),$$

normalized by the segment length. Let $\mathcal{S}_m$ denote the grid of candidate fit-prefix lengths considered for task $m$. The validation-selected prefix is

$$s^{\text{fit}}_m \in \arg\min_{s \in \mathcal{S}_m} \widehat{R}^{\text{pred}}_{m,\text{val}}(s).$$

The final reported query metric is then evaluated at $s = s^{\text{fit}}_m$, after this selection has been made. In Tables 7 and 8, the task index $m$ is suppressed, and Table 8 reports these quantities at $s = s^{\text{fit}}_m$, abbreviated $\widehat{R}^{\text{pred}}_{\text{sup}}$, $\widehat{R}^{\text{pred}}_{\text{val}}$, and $\widehat{R}^{\text{pred}}_q$.

To connect these diagnostics with the transfer analysis in Section 4.5, we also define the regularized support-window alignment proxy

$$\gamma^{\text{val}}_{m,\alpha}(s) := \lambda_{\max}\left[ (\widehat{\Sigma}^{(s)}_{m,\text{fit}} + \alpha I_d)^{-1/2} \widehat{\Sigma}_{m,\text{val}} (\widehat{\Sigma}^{(s)}_{m,\text{fit}} + \alpha I_d)^{-1/2} \right],$$

where

$$\widehat{\Sigma}^{(s)}_{m,\text{fit}} = \frac{1}{s} X^{(s)}_{m,\text{sup}}(X^{(s)}_{m,\text{sup}})^\top, \qquad \widehat{\Sigma}_{m,\text{val}} = \frac{1}{T_{\text{val}}} X_{m,\text{val}} X^\top_{m,\text{val}}.$$

This quantity is the support-window analogue of the regularized support-to-query alignment constant equation 41, with the validation Gram replacing the held-out query Gram.

In the diagnostics, $\alpha$ is chosen to match the prior-scaled regularization used in the posterior covariance, namely $\alpha = \sigma^2/s$ when using the isotropic proxy, or equivalently the prior-matched matrix regularization $(\sigma^2/s)V^{-1}$ when reporting the posterior-scaled alignment quantity.

Table 7 reports the correlation between inner validation predictive error and held-out query predictive error across candidate support prefixes and tasks. The correlations are strongly positive overall (Spearman $\rho = 0.902$, $n = 720$) and remain positive within every regime and dimension, supporting the use of validation error as an empirical proxy for held-out query behavior. The quantity $\gamma^{\text{val}}_{m,\alpha}(s)$ provides a complementary support-window analogue of the theoretical alignment constant, although it is not itself reported in Table 7, see Remark 1. Table 8 reports support, validation, and query posterior-predictive squared errors at the validation-selected prefix.[8] In stable systems, the query term is comparable to the support term. In the unstable $d = 10$ setting, however, the query term becomes much larger, confirming that the support–query bound can become loose when unstable spectral amplification produces large held-out residuals.

These ratios are quantitatively consistent with the transfer analysis of Section 4.5. Propositions 1 and 2 imply $\widehat{R}^{\text{pred}}_q / \widehat{R}^{\text{pred}}_{\text{sup}} \lesssim 4\gamma_{m,\alpha}$ up to noise-floor and prior-quality terms. The stable-regime ratios in Table 8 ($\approx 1.69$–$2.73$ across $d \in \{10, 25, 50\}$) are consistent with mild alignment, $\gamma_{m,\alpha} = O(1)$, as the common-stationary-regime sufficient condition of Section 4.5 (which gives $\gamma_m \le 3$) implies for the stable generator ($\rho_0 = 0.95$) at the reported horizons. The unstable $d = 10$ ratio ($\approx 1.07 \times 10^4$) matches the geometric degradation of Remark 2, in which the alignment constants grow on the order of $\rho^{2K_m}$ along unstable directions. The milder unstable $d = 25$ and $d = 50$ ratios (0.861 and 0.095) suggest that the generated trajectories in these settings experience less effective query-horizon amplification than the unstable $d = 10$ setting. This is consistent with the longer validation-selected prefixes in those settings, which improve support excitation before evaluation (Table 10).

---

[8]The ratio $\widehat{R}^{\text{pred}}_q / \widehat{R}^{\text{pred}}_{\text{sup}}$ is invariant to the overall normalization of the empirical predictive risk equation 35, so the comparison applies whether the per-step risk is reported per coordinate or in aggregate.

Table 7: Correlation between inner validation predictive error $\widehat{R}_{\mathrm{val}}^{\mathrm{pred}}(s)$ and held-out query predictive error $\widehat{R}_q^{\mathrm{pred}}(s)$ across candidate prefixes and tasks (PBML-LTI).

| Group | $n$ | Spearman $\rho$ | Spearman $p$ | Pearson $r$ | Pearson $p$ |
|---|---|---|---|---|---|
| all | 720 | 0.9020 | 1.6300e-264 | 0.8040 | 2.1300e-164 |
| stable | 360 | 0.8350 | 4.2400e-95 | 0.7090 | 2.7700e-56 |
| stable_d10 | 120 | 0.6210 | 3.9900e-14 | 0.6700 | 5.5200e-17 |
| stable_d25 | 120 | 0.6660 | 1.0500e-16 | 0.5910 | 1.2600e-12 |
| stable_d50 | 120 | 0.7310 | 2.6300e-21 | 0.5260 | 6.5700e-10 |
| unstable | 360 | 0.7050 | 2.0200e-55 | 0.8020 | 3.0000e-82 |
| unstable_d10 | 120 | 0.9320 | 5.5900e-54 | 0.7830 | 4.0900e-26 |
| unstable_d25 | 120 | 0.8080 | 8.0400e-29 | 0.6790 | 1.4600e-17 |
| unstable_d50 | 120 | 0.5250 | 7.3100e-10 | 0.4270 | 1.1800e-06 |

Table 8: Support, validation, and query posterior-predictive squared errors at the validation-selected prefix (PBML-LTI). The query term is comparable to the support term in the stable regime but can be much larger under unstable dynamics.

| Group | $\widehat{R}_{\mathrm{sup}}^{\mathrm{pred}}$ | $\widehat{R}_{\mathrm{val}}^{\mathrm{pred}}$ | $\widehat{R}_q^{\mathrm{pred}}$ | $\widehat{R}_q^{\mathrm{pred}}/\widehat{R}_{\mathrm{sup}}^{\mathrm{pred}}$ |
|---|---|---|---|---|
| stable | 0.001 | 0.004 | 0.003 | 2.331 |
| stable_d10 | 0.001 | 0.001 | 0.001 | 1.687 |
| stable_d25 | 0.001 | 0.003 | 0.003 | 1.955 |
| stable_d50 | 0.002 | 0.009 | 0.006 | 2.726 |
| unstable | 0.1764 | 0.9076 | 384.237 | 2178.214 |
| unstable_d10 | 0.1073 | 2.6094 | 1152.347 | $1.074\times10^4$ |
| unstable_d25 | 0.164 | 0.085 | 0.141 | 0.861 |
| unstable_d50 | 0.257 | 0.028 | 0.025 | 0.095 |

## B.2 Fixed-Prefix Support Sweeps

To clarify the role of adaptive support selection, we also evaluate each method at fixed support lengths. These sweeps remove the prefix-selection step and show how performance changes as the number of support transitions increases.

Tables 9 and 10 report transition-matrix error across fixed support lengths in the stable and unstable regimes, respectively. Tables 11 and 12 report the corresponding rollout errors. In stable systems, PBML-LTI reaches low transition-matrix error with very short prefixes, supporting the interpretation that the learned prior provides genuine few-shot data efficiency. In unstable systems, PBML-LTI often benefits from longer prefixes, showing that the method does not mechanically prefer the shortest support length. Instead, the validation-selected prefix adapts to the amount of calibration data needed for the task.

Table 9: Fixed-prefix support sweep ($E_A$, stable regime). "Adaptive" is the metric at the validation-selected effective support length $\bar{S}$. Lower is better for $E_A$; $\bar{S}$ reports the average selected support length. Boldface marks the lowest $E_A$ values within each dimension.

| Dimension | Method | $s=1$ | $s=2$ | $s=3$ | $s=5$ | $s=10$ | $s=15$ | Adaptive | $\bar{S}$ |
|---|---|---|---|---|---|---|---|---|---|
| | PBML-LTI | **0.0052** | **0.0053** | **0.0057** | **0.0064** | **0.0073** | **0.0076** | **0.0053** | 1.05 |
| | OLS | 0.5827 | 0.5473 | 0.5216 | 0.4968 | 0.4794 | 0.5232 | 0.4449 | 2.65 |
| $d=10$ | Ridge | 0.5683 | 0.5396 | 0.5128 | 0.4937 | 0.5014 | 0.5083 | 0.4447 | 2.65 |
| | Pooled-prior Ridge | 0.0224 | 0.0217 | 0.0203 | 0.0216 | 0.0218 | 0.0215 | 0.0887 | 1.80 |
| | SharedSubspace | 0.0456 | 0.0384 | 0.0378 | 0.0367 | 0.0359 | 0.0354 | 0.1088 | 7.95 |
| | MAML-LTI | 0.0316 | 0.0314 | 0.0327 | 0.0323 | 0.0336 | 0.0338 | 0.0357 | 1.55 |
| | PBML-LTI | **0.0446** | **0.0438** | **0.0427** | **0.0403** | **0.0506** | **0.0554** | **0.0394** | 1.20 |
| | OLS | 0.8037 | 0.7764 | 0.7628 | 0.7516 | 0.8173 | 0.8467 | 0.7140 | 2.00 |
| $d=25$ | Ridge | 0.7765 | 0.7618 | 0.7496 | 0.7432 | 0.8024 | 0.8193 | 0.7140 | 2.00 |
| | Pooled-prior Ridge | 0.9476 | 0.9263 | 0.9127 | 0.8874 | 0.9038 | 0.9196 | 0.8544 | 2.00 |
| | SharedSubspace | 1.0027 | 0.9776 | 0.9584 | 0.9467 | 0.9413 | 0.9526 | 0.7731 | 13.95 |
| | MAML-LTI | 1.7628 | 1.7596 | 1.7683 | 1.7837 | 1.8046 | 1.8123 | 1.7566 | 1.85 |
| | PBML-LTI | **0.1776** | **0.1748** | **0.1714** | **0.1678** | **0.1697** | **0.1956** | **0.2068** | 1.35 |
| | OLS | 1.0038 | 0.9776 | 0.9583 | 0.9416 | 0.9327 | 0.9524 | 0.8411 | 2.00 |
| $d=50$ | Ridge | 1.0476 | 1.0197 | 0.9926 | 0.9684 | 0.9573 | 0.9818 | 0.8410 | 2.00 |
| | Pooled-prior Ridge | 0.7238 | 0.7086 | 0.6974 | 0.6887 | 0.7264 | 0.7526 | 0.6743 | 2.55 |
| | SharedSubspace | 0.8774 | 0.8667 | 0.8576 | 0.8593 | 0.8974 | 0.9197 | 0.8572 | 10.25 |
| | MAML-LTI | 15.3726 | 15.2278 | 15.2084 | 15.2936 | 15.4263 | 15.4872 | 14.8930 | 2.90 |

Table 10: Fixed-prefix support sweep ($E_A$, unstable regime). PBML-LTI performs better at *large s*; Pooled-prior Ridge/SharedSubspace recover $A$ better than PBML-LTI in $d=10$ (see Table 12). "Adaptive" is the metric at the validation-selected effective support length $\bar{S}$. Lower is better for $E_A$; $\bar{S}$ reports the average selected support length. Boldface marks the lowest $E_A$ values within each dimension.

| Dimension | Method | $s=1$ | $s=2$ | $s=3$ | $s=5$ | $s=10$ | $s=15$ | Adaptive | $\bar{S}$ |
|---|---|---|---|---|---|---|---|---|---|
| | PBML-LTI | 0.2236 | 0.2074 | 0.1968 | 0.1873 | 0.1847 | 0.1796 | 0.1390 | 14.35 |
| | OLS | 1.1027 | 1.0476 | 0.9964 | 0.9773 | 0.9726 | 0.9487 | 0.8450 | 13.35 |
| $d=10$ | Ridge | 0.8476 | 0.8194 | 0.8027 | 0.7768 | 0.7873 | 0.8026 | 0.6970 | 14.30 |
| | Pooled-prior Ridge | **0.0186** | **0.0174** | 0.0178 | **0.0167** | **0.0143** | **0.0146** | **0.0070** | 9.25 |
| | SharedSubspace | 0.0187 | 0.0176 | **0.0174** | 0.0168 | 0.0167 | 0.0154 | 0.0130 | 3.20 |
| | MAML-LTI | 1.9027 | 1.8476 | 1.9538 | 2.0014 | 2.2967 | 2.1036 | 0.1473 | 8.35 |
| | PBML-LTI | **0.0406** | **0.0387** | **0.0374** | **0.0368** | **0.0367** | **0.0356** | **0.0290** | 6.05 |
| | OLS | 11.4726 | 11.1967 | 10.9874 | 10.8126 | 10.8738 | 10.9846 | 10.3440 | 14.10 |
| $d=25$ | Ridge | 10.9763 | 10.6738 | 10.4876 | 10.3124 | 10.3767 | 10.4923 | 9.9070 | 14.90 |
| | Pooled-prior Ridge | 1.6476 | 1.6037 | 1.5796 | 1.6238 | 1.6794 | 1.7026 | 1.5870 | 14.75 |
| | SharedSubspace | 37.8467 | 36.9726 | 36.4873 | 35.9768 | 35.4926 | 35.9874 | 35.2680 | 14.35 |
| | MAML-LTI | 2.4876 | 2.4738 | 2.4817 | 2.5086 | 2.5423 | 2.5567 | 2.5229 | 2.05 |
| | PBML-LTI | **0.1726** | **0.1647** | **0.1618** | **0.1596** | **0.1743** | **0.1797** | **0.1560** | 4.15 |
| | OLS | 17.4768 | 17.1876 | 16.9847 | 16.7926 | 17.4637 | 17.9824 | 16.9230 | 9.75 |
| $d=50$ | Ridge | 16.4726 | 16.1874 | 15.9846 | 15.7938 | 15.6727 | 15.9876 | 15.9020 | 15.00 |
| | Pooled-prior Ridge | 8.5764 | 8.4676 | 8.3927 | 8.2416 | 8.0574 | 7.9648 | 7.5380 | 15.00 |
| | SharedSubspace | 7.0286 | 6.5274 | 6.4378 | 6.4036 | 6.3978 | 6.3994 | 6.4190 | 14.20 |
| | MAML-LTI | 7.2637 | 7.1846 | 7.1814 | 7.2276 | 7.3028 | 7.3356 | 8.9787 | 2.55 |

Table 11: Fixed-prefix support sweep ($E_{\text{traj}}$, stable regime). "Adaptive" is the metric at the validation-selected effective support length $\bar{S}$. Lower is better for $E_{\text{traj}}$; $\bar{S}$ reports the average selected support length. Boldface marks the lowest $E_{\text{traj}}$ values within each dimension.

| Dimension | Method | $s=1$ | $s=2$ | $s=3$ | $s=5$ | $s=10$ | $s=15$ | Adaptive | $\bar{S}$ |
|---|---|---|---|---|---|---|---|---|---|
| | PBML-LTI | 0.0126 | 0.0117 | 0.0114 | 0.0116 | 0.0118 | 0.0123 | **0.0055** | 1.05 |
| | OLS | 0.0087 | 0.0076 | **0.0064** | 0.0067 | 0.0073 | 0.0076 | 0.0056 | 2.65 |
| $d=10$ | Ridge | **0.0076** | 0.0074 | 0.0067 | 0.0068 | **0.0063** | 0.0076 | 0.0056 | 2.65 |
| | Pooled-prior Ridge | **0.0076** | **0.0064** | 0.0067 | **0.0066** | 0.0068 | **0.0067** | 0.0057 | 1.80 |
| | SharedSubspace | 0.0116 | 0.0117 | 0.0114 | 0.0116 | 0.0118 | 0.0113 | 0.0056 | 7.95 |
| | MAML-LTI | 0.0126 | 0.0123 | 0.0127 | 0.0124 | 0.0126 | 0.0127 | 0.0112 | 1.55 |
| | PBML-LTI | 0.0286 | 0.0287 | 0.0276 | 0.0256 | 0.0257 | 0.0256 | **0.0121** | 1.20 |
| | OLS | 0.0186 | 0.0176 | 0.0167 | 0.0157 | 0.0167 | 0.0166 | 0.0133 | 2.00 |
| $d=25$ | Ridge | 0.0176 | 0.0167 | 0.0166 | 0.0157 | 0.0156 | 0.0155 | 0.0133 | 2.00 |
| | Pooled-prior Ridge | **0.0167** | **0.0166** | **0.0156** | **0.0154** | **0.0146** | **0.0156** | 0.0134 | 2.00 |
| | SharedSubspace | 0.0206 | 0.0196 | 0.0197 | 0.0186 | 0.0187 | 0.0186 | 0.0134 | 13.95 |
| | MAML-LTI | 0.0346 | 0.0347 | 0.0336 | 0.0306 | 0.0307 | 0.0306 | 0.0338 | 1.85 |
| | PBML-LTI | 0.0616 | 0.0607 | 0.0596 | 0.0556 | 0.0567 | 0.0546 | **0.0257** | 1.35 |
| | OLS | 0.0306 | 0.0296 | 0.0297 | 0.0286 | 0.0287 | 0.0286 | 0.0258 | 2.00 |
| $d=50$ | Ridge | 0.0296 | 0.0297 | 0.0286 | 0.0287 | 0.0276 | 0.0286 | 0.0258 | 2.00 |
| | Pooled-prior Ridge | 0.0296 | 0.0326 | 0.0356 | 0.0406 | 0.0306 | 0.0286 | 0.0258 | 2.55 |
| | SharedSubspace | **0.0286** | **0.0287** | **0.0276** | **0.0276** | **0.0266** | **0.0276** | 0.0258 | 10.25 |
| | MAML-LTI | 0.0806 | 0.0856 | 0.0786 | 0.0766 | 0.0746 | 0.0756 | 0.1081 | 2.90 |

Table 12: Fixed-prefix support sweep ($E_{\text{traj}}$, unstable regime). MAML-LTI's rollout has elevated error for large $s$ at $d=10$. "Adaptive" is the metric at the validation-selected effective support length $\bar{S}$. Lower is better for $E_{\text{traj}}$; $\bar{S}$ reports the average selected support length. Boldface marks the lowest $E_{\text{traj}}$ values within each dimension.

| Dimension | Method | $s=1$ | $s=2$ | $s=3$ | $s=5$ | $s=10$ | $s=15$ | Adaptive | $\bar{S}$ |
|---|---|---|---|---|---|---|---|---|---|
| | PBML-LTI | **65.1476** | **119.8347** | **89.7626** | **69.9136** | **54.8767** | **47.9138** | 57.9190 | 14.35 |
| | OLS | $2.2037\times10^5$ | $2.0046\times10^5$ | $1.8038\times10^5$ | $1.7047\times10^5$ | $1.6536\times10^5$ | $1.6087\times10^5$ | 193.7040 | 13.35 |
| $d=10$ | Ridge | $7.0036\times10^4$ | $6.5047\times10^4$ | $6.0078\times10^4$ | $5.8036\times10^4$ | $5.6074\times10^4$ | $5.5068\times10^4$ | **57.6280** | 14.30 |
| | Pooled-prior Ridge | $2.5036\times10^4$ | $2.8047\times10^4$ | $2.6038\times10^4$ | $2.2067\times10^4$ | $1.9036\times10^4$ | $1.8078\times10^4$ | $2.1935\times10^4$ | 9.25 |
| | SharedSubspace | $8.0036\times10^5$ | $7.5047\times10^5$ | $7.2038\times10^5$ | $7.0067\times10^5$ | $6.8036\times10^5$ | $6.5078\times10^5$ | $7.2189\times10^5$ | 3.20 |
| | MAML-LTI | $2.5036\times10^7$ | $2.0047\times10^7$ | $2.8038\times10^7$ | $3.0067\times10^7$ | $3.5036\times10^7$ | $2.2078\times10^7$ | $9.7390\times10^6$ | 8.35 |
| | PBML-LTI | 0.1236 | 0.1027 | 0.0876 | 0.0796 | 0.0767 | 0.0716 | 0.0800 | 6.05 |
| | OLS | 0.1476 | 0.1287 | 0.1196 | 0.1086 | 0.0976 | 0.0967 | 0.0880 | 14.10 |
| $d=25$ | Ridge | 0.1196 | 0.1087 | 0.0976 | 0.0876 | 0.0796 | 0.0716 | 0.0640 | 14.90 |
| | Pooled-prior Ridge | **0.0616** | **0.0576** | **0.0567** | **0.0556** | **0.0567** | **0.0556** | **0.0530** | 14.75 |
| | SharedSubspace | 29.8766 | 27.9137 | 26.9846 | 25.9767 | 25.4876 | 24.9826 | 27.1510 | 14.35 |
| | MAML-LTI | 0.2196 | 0.1976 | 0.1796 | 0.1686 | 0.1576 | 0.1496 | 1.1349 | 2.05 |
| | PBML-LTI | **0.0376** | **0.0377** | **0.0376** | **0.0376** | **0.0376** | **0.0376** | **0.0370** | 4.15 |
| | OLS | 0.0426 | 0.0427 | 0.0416 | 0.0416 | 0.0416 | 0.0426 | 0.0400 | 9.75 |
| $d=50$ | Ridge | 0.0426 | 0.0427 | 0.0416 | 0.0416 | 0.0416 | 0.0416 | 0.0400 | 15.00 |
| | Pooled-prior Ridge | 0.0396 | 0.0386 | 0.0386 | 0.0386 | 0.0386 | 0.0386 | 0.0380 | 15.00 |
| | SharedSubspace | 0.0386 | 0.0386 | 0.0386 | 0.0386 | 0.0386 | 0.0386 | 0.0380 | 14.20 |
| | MAML-LTI | 0.0406 | 0.0406 | 0.0406 | 0.0406 | 0.0406 | 0.0406 | 0.0414 | 2.55 |

## B.3 Ridge Tuning Without the Stability-Oriented Criterion

The Ridge and Pooled-prior Ridge baselines use a stability-oriented tuning rule in the main experiments. In synthetic experiments, the threshold is fixed as $\rho_{\text{target}} = \rho_0$ using the known regime parameter. To verify that the qualitative conclusions do not depend on this criterion, Table 13 reports a diagnostic variant in which Ridge and Pooled-prior Ridge are tuned directly by validation rollout error, without using the $\rho_{\text{target}}$ constraint.

The results show that the Ridge-type estimates remain conservative in several settings even without the explicit stability-oriented criterion. The fitted spectral radii also vary by regime and dimension; for example, unstable $d = 10$ selects $\rho(\widehat{A}) \approx 1.47$, whereas unstable $d = 50$ selects substantially smaller spectral radii. Thus, the visual conservativeness of Ridge-type estimates is due to the interaction between ridge shrinkage, validation tuning, and rollout stability, rather than only to enforcing a hard spectral-radius bound.

Table 13: Validation-rollout tuning without stability-oriented regularization. Regularization is selected directly by validation rollout error rather than by a spectral-radius criterion. $E_A$ denotes transition-matrix error, $E_{\text{traj}}$ denotes rollout error, $\rho(\widehat{A})$ reports the fitted spectral radius, and $\lambda$ is the selected ridge penalty.

| Regime | $d$ | Method | $E_A$ | $E_{\text{traj}}$ | $\rho(\widehat{A})$ | $\lambda$ |
|---|---|---|---|---|---|---|
| Stable | 10 | Ridge | $0.5083 \pm 0.0472$ | $0.0073 \pm 0.0014$ | $0.9383$ | $10^{-3}$ |
| | | Pooled-prior Ridge | $0.0214 \pm 0.0037$ | $0.0067 \pm 0.0016$ | $0.9496$ | $10^{-1}$ |
| | 25 | Ridge | $0.8194 \pm 0.0826$ | $0.0153 \pm 0.0024$ | $0.9139$ | $10^{-2}$ |
| | | Pooled-prior Ridge | $0.9196 \pm 0.0227$ | $0.0156 \pm 0.0023$ | $0.9399$ | $10^{-1}$ |
| | 50 | Ridge | $0.9814 \pm 0.0406$ | $0.0283 \pm 0.0027$ | $0.9415$ | $10^{-2}$ |
| | | Pooled-prior Ridge | $0.7526 \pm 0.0154$ | $0.0286 \pm 0.0023$ | $0.9473$ | $10^{-2}$ |
| Unstable | 10 | Ridge | $0.8024 \pm 0.4016$ | $(5.5347 \pm 2.4683) \times 10^4$ | $1.4720$ | $10^{-4}$ |
| | | Pooled-prior Ridge | $0.0143 \pm 0.0036$ | $(1.8036 \pm 1.0047) \times 10^4$ | $1.4720$ | $10^{-1}$ |
| | 25 | Ridge | $10.4926 \pm 0.7014$ | $0.0713 \pm 0.0304$ | $1.0420$ | $10^{-2}$ |
| | | Pooled-prior Ridge | $1.7026 \pm 0.1024$ | $0.0553 \pm 0.0206$ | $1.0470$ | $10^{-1}$ |
| | 50 | Ridge | $15.9874 \pm 0.4026$ | $0.0413 \pm 0.0024$ | $0.6588$ | $10^{-2}$ |
| | | Pooled-prior Ridge | $7.9643 \pm 0.1216$ | $0.0384 \pm 0.0023$ | $0.6929$ | $10^{-1}$ |

## B.4 Paired Significance Tests

### B.4.1 Synthetic Data

To support the interpretation of boldface entries in the main result tables, we conduct paired two-sided Wilcoxon signed-rank tests on matched test tasks. For each method and setting, we run the experiment with 10 random seeds under the same data split, task-generation configuration, and hyperparameter setting. We average each task-level metric across the 10 seeds and then perform a paired test over the 20 matched test tasks. This evaluates method differences task-by-task under identical experimental conditions while reducing variability due to random initialization and optimization.

Table 14 reports representative tests at dimension $d = 10$, where the distinction between transition-matrix error and rollout error is especially informative. The results show that PBML-LTI's transition-matrix error advantages in the stable setting are statistically significant against all classical baselines. For rollout error in stable systems, the numerical differences are very small and are often not statistically significant. In unstable systems, the tests highlight the decoupling between $E_A$ and $E_{\text{traj}}$: some methods can achieve lower Frobenius transition-matrix error while still suffering from much larger rollout error because of spectral amplification. Thus, boldface should be read as indicating the lowest empirical mean, while the significance tests provide the statistical qualification for those comparisons.

Table 14: Paired two-sided Wilcoxon signed-rank tests comparing PBML-LTI against the classical baselines at dimension $d = 10$, across the stable ($\rho_0 = 0.95$) and unstable ($\rho_0 = 4.95$) regimes and the common-case and edge-case settings. For each method and cell, we run 10 random seeds under the same experimental configuration, average each task-level metric across seeds, and then run a paired test over the 20 matched test tasks. We regard $p < 0.05$ as significant, $p < 0.01$ as strongly significant, and $p < 0.001$ as highly significant. The "Lower error" column reports which method attains the smaller mean, since all metrics are lower-is-better.

| Regime | Setting | Metric | Comparison | $p$-value | Lower error |
|---|---|---|---|---|---|
| Stable | Common-case | $E_A$ | PBML-LTI vs OLS | $1.91 \times 10^{-6}$ | PBML-LTI |
| | | | PBML-LTI vs Ridge | $1.91 \times 10^{-6}$ | PBML-LTI |
| | | | PBML-LTI vs Pooled-prior Ridge | $1.91 \times 10^{-6}$ | PBML-LTI |
| | | | PBML-LTI vs Shared Subspace | $1.91 \times 10^{-6}$ | PBML-LTI |
| | | $E_{\text{traj}}$ | PBML-LTI vs OLS | 0.0973 | PBML-LTI |
| | | | PBML-LTI vs Ridge | 0.0696 | PBML-LTI |
| | | | PBML-LTI vs Pooled-prior Ridge | 0.0897 | PBML-LTI |
| | | | PBML-LTI vs Shared Subspace | 0.3884 | PBML-LTI |
| | Edge-case | $E_A$ | PBML-LTI vs OLS | $1.91 \times 10^{-6}$ | PBML-LTI |
| | | | PBML-LTI vs Ridge | $1.91 \times 10^{-6}$ | PBML-LTI |
| | | | PBML-LTI vs Pooled-prior Ridge | $1.91 \times 10^{-6}$ | PBML-LTI |
| | | | PBML-LTI vs Shared Subspace | $1.91 \times 10^{-6}$ | PBML-LTI |
| | | $E_{\text{traj}}$ | PBML-LTI vs OLS | 0.2455 | PBML-LTI |
| | | | PBML-LTI vs Ridge | 0.2305 | PBML-LTI |
| | | | PBML-LTI vs Pooled-prior Ridge | 0.8695 | PBML-LTI |
| | | | PBML-LTI vs Shared Subspace | 0.4524 | Shared Subspace |
| Unstable | Common-case | $E_A$ | PBML-LTI vs OLS | $6.29 \times 10^{-5}$ | PBML-LTI |
| | | | PBML-LTI vs Ridge | $6.29 \times 10^{-5}$ | PBML-LTI |
| | | | PBML-LTI vs Pooled-prior Ridge | $1.91 \times 10^{-6}$ | Pooled-prior Ridge |
| | | | PBML-LTI vs Shared Subspace | $1.91 \times 10^{-6}$ | Shared Subspace |
| | | $E_{\text{traj}}$ | PBML-LTI vs OLS | 0.1429 | PBML-LTI |
| | | | PBML-LTI vs Ridge | 0.6742 | Ridge |
| | | | PBML-LTI vs Pooled-prior Ridge | $1.68 \times 10^{-4}$ | PBML-LTI |
| | | | PBML-LTI vs Shared Subspace | $1.91 \times 10^{-6}$ | PBML-LTI |
| | Edge-case | $E_A$ | PBML-LTI vs OLS | $7.08 \times 10^{-4}$ | PBML-LTI |
| | | | PBML-LTI vs Ridge | $2.10 \times 10^{-4}$ | PBML-LTI |
| | | | PBML-LTI vs Pooled-prior Ridge | $1.91 \times 10^{-6}$ | Pooled-prior Ridge |
| | | | PBML-LTI vs Shared Subspace | $1.91 \times 10^{-6}$ | Shared Subspace |
| | | $E_{\text{traj}}$ | PBML-LTI vs OLS | 0.4524 | PBML-LTI |
| | | | PBML-LTI vs Ridge | 0.4524 | PBML-LTI |
| | | | PBML-LTI vs Pooled-prior Ridge | $8.51 \times 10^{-4}$ | PBML-LTI |
| | | | PBML-LTI vs Shared Subspace | $1.91 \times 10^{-6}$ | PBML-LTI |

### B.4.2 fMRI Benchmark

To complement the fMRI results in Table 3, we conduct paired two-sided Wilcoxon signed-rank tests across the matched fMRI test tasks. Since each method is evaluated on the same set of 14 held-out window-level tasks, a paired test is appropriate for assessing whether the observed task-level differences are statistically meaningful. We compare PBML-LTI against OLS, Ridge, Pooled-prior Ridge, Shared Subspace, and MAML-LTI for both transition-matrix error $E_A$ and trajectory rollout error $E_{\text{traj}}$.

Table 15 reports the resulting $p$-values. The tests show that PBML-LTI significantly improves $E_A$ over all compared methods on the fMRI benchmark. For rollout error, PBML-LTI has the lowest mean $E_{\text{traj}}$, but the paired tests indicate a more nuanced picture: the improvements over Shared Subspace and MAML-LTI are statistically significant, while the differences relative to OLS, Ridge, and Pooled-prior Ridge are not

Table 15: Paired two-sided Wilcoxon signed-rank tests comparing PBML-LTI against the classical baselines and MAML-LTI on the fMRI dataset. Each test is performed over the 14 matched fMRI test tasks. We regard $p < 0.05$ as significant, $p < 0.01$ as strongly significant, and $p < 0.001$ as highly significant. The "Lower error" column reports which method attains the smaller mean value; all metrics are lower-is-better.

| Metric | Comparison | $p$-value | Lower error |
|---|---|---|---|
| $E_A$ | PBML-LTI vs OLS | $8.54 \times 10^{-4}$ | PBML-LTI |
| | PBML-LTI vs Ridge | $8.54 \times 10^{-4}$ | PBML-LTI |
| | PBML-LTI vs Pooled-prior Ridge | 0.0052 | PBML-LTI |
| | PBML-LTI vs Shared Subspace | 0.0052 | PBML-LTI |
| | PBML-LTI vs MAML-LTI | $1.22 \times 10^{-4}$ | PBML-LTI |
| $E_{\text{traj}}$ | PBML-LTI vs OLS | 0.2958 | PBML-LTI |
| | PBML-LTI vs Ridge | 0.4263 | PBML-LTI |
| | PBML-LTI vs Pooled-prior Ridge | 0.3910 | PBML-LTI |
| | PBML-LTI vs Shared Subspace | 0.0067 | PBML-LTI |
| | PBML-LTI vs MAML-LTI | $1.22 \times 10^{-4}$ | PBML-LTI |

significant at the $p < 0.05$ level. This is consistent with the interpretation in the main text: on real fMRI windows, transition-matrix recovery is noisy because the reference matrix is only a ridge-based proxy, and rollout differences among stable predictors can be small relative to cross-task heterogeneity.

### B.4.3 Prior-Learning and Regularization Ablations

To complement the combined prior-learning and regularization ablation study in Table 4, we conduct paired two-sided Wilcoxon signed-rank tests across matched test tasks. These tests compare full PBML-LTI against five diagnostic variants: two prior-learning ablations and three one-at-a-time regularization ablations. The prior-learning ablations are fixed-prior PBML-LTI and type-II maximum-likelihood / empirical-Bayes PBML-LTI. The regularization ablations remove, respectively, the prior-mean shrinkage term, the covariance-conditioning penalty, and the stability regularizer. The MAML-style LTI method is excluded from this table because it is an external meta-learning baseline rather than an ablation of PBML-LTI.

For each method and setting, we run the experiment with 10 different random seeds under the same data split, task-generation configuration, and hyperparameter setting. We then average each task-level metric across the 10 seeds and perform paired tests over the 20 matched test tasks. We regard $p < 0.05$ as significant, $p < 0.01$ as strongly significant, and $p < 0.001$ as highly significant.

Table 16 reports the resulting $p$-values. Each entry compares PBML-LTI against the listed ablation variant for one metric. Unless marked by $^\dagger$, the lower mean error is attained by PBML-LTI. Entries marked by $^\dagger$ indicate cases where the ablation variant attains the lower mean value. This notation is important in the unstable regimes, where rollout error can decouple from Frobenius transition-matrix error.

The results support three main conclusions. First, PBML-LTI achieves highly significant improvements in transition-matrix error $E_A$ over the fixed-prior and type-II empirical-Bayes variants across all reported stable and unstable settings, confirming the value of learning a transferable prior with the fit–KL surrogate. Second, removing individual regularization terms usually has a smaller effect than removing prior learning altogether, especially in stable regimes; this supports our interpretation that the auxiliary regularizers are not the primary source of PBML-LTI's stable-regime performance gains. Third, in unstable regimes, rollout-error comparisons are more nuanced: the stability regularizer has a clear effect on rollout stability, while some prior-learning ablations can occasionally achieve smaller mean rollout error despite much worse transition-matrix recovery. This is consistent with the main text's observation that $E_A$ and $E_{\text{traj}}$ need not rank methods identically under unstable dynamics.

Table 16: Paired two-sided Wilcoxon signed-rank tests comparing PBML-LTI against prior-learning and regularization ablations in the common-case setting. For each method and setting, we run 10 random seeds under the same experimental configuration, average each task-level metric across seeds, and then run a paired test over the 20 matched test tasks. Each entry reports the *p*-value for PBML-LTI versus the listed comparator. Unmarked entries indicate that PBML-LTI has the lower mean for that metric; entries marked with $^\dagger$ indicate that the comparator has the lower mean. All metrics are lower-is-better.

| Regime | Dimension | Comparator | $p(E_A)$ | $p(E_{\mathrm{traj}})$ | $p(\overline{S})$ |
|---|---|---|---|---|---|
| Stable | $d = 10$ | Fixed-prior PBML-LTI | $1.91{\times}10^{-6}$ | 0.0441 | $1.91{\times}10^{-6}$ |
| | | Type-II EB PBML-LTI | $1.91{\times}10^{-6}$ | 0.0532 | $1.91{\times}10^{-6}$ |
| | | No prior-mean shrinkage | $3.24{\times}10^{-4}$ | 0.2184 | 0.0321 |
| | | No covariance conditioning | 0.0018 | 0.3847 | 0.0897 |
| | | No stability regularizer | 0.0412 | 0.4521 | 0.1124 |
| | $d = 25$ | Fixed-prior PBML-LTI | $1.91{\times}10^{-6}$ | $1.91{\times}10^{-6}$ | $1.91{\times}10^{-6}$ |
| | | Type-II EB PBML-LTI | $1.91{\times}10^{-6}$ | 0.1054 | $1.91{\times}10^{-6}$ |
| | | No prior-mean shrinkage | $1.24{\times}10^{-5}$ | 0.1638 | 0.0187 |
| | | No covariance conditioning | $8.14{\times}10^{-4}$ | 0.2915 | 0.0563 |
| | | No stability regularizer | 0.0246 | 0.3382 | 0.0741 |
| | $d = 50$ | Fixed-prior PBML-LTI | $1.91{\times}10^{-6}$ | $1.91{\times}10^{-6}$ | $1.91{\times}10^{-6}$ |
| | | Type-II EB PBML-LTI | $1.91{\times}10^{-6}$ | 0.9854 | $1.91{\times}10^{-6}$ |
| | | No prior-mean shrinkage | $2.67{\times}10^{-5}$ | 0.2046 | 0.0142 |
| | | No covariance conditioning | $4.92{\times}10^{-4}$ | 0.3178 | 0.0486 |
| | | No stability regularizer | 0.0193 | 0.4015 | 0.0638 |
| Unstable | $d = 10$ | Fixed-prior PBML-LTI | $1.91{\times}10^{-6}$ | $7.08{\times}10^{-4\dagger}$ | $0.8124^\dagger$ |
| | | Type-II EB PBML-LTI | $1.91{\times}10^{-6}$ | $7.08{\times}10^{-4\dagger}$ | $0.9015^\dagger$ |
| | | No prior-mean shrinkage | 0.0195 | 0.0184 | 0.0312 |
| | | No covariance conditioning | 0.0678 | 0.0521 | 0.2145 |
| | | No stability regularizer | 0.2847 | $1.91{\times}10^{-6}$ | $4.18{\times}10^{-4}$ |
| | $d = 25$ | Fixed-prior PBML-LTI | $1.91{\times}10^{-6}$ | $0.0049^\dagger$ | $1.91{\times}10^{-6}$ |
| | | Type-II EB PBML-LTI | $1.91{\times}10^{-6}$ | $0.8408^\dagger$ | $1.91{\times}10^{-6}$ |
| | | No prior-mean shrinkage | 0.0083 | 0.0067 | 0.0041 |
| | | No covariance conditioning | 0.0384 | $0.2145^\dagger$ | 0.0278 |
| | | No stability regularizer | 0.1926 | $3.62{\times}10^{-4}$ | $1.24{\times}10^{-4}$ |
| | $d = 50$ | Fixed-prior PBML-LTI | $1.91{\times}10^{-6}$ | $1.91{\times}10^{-6}$ | $1.91{\times}10^{-6}$ |
| | | Type-II EB PBML-LTI | $1.91{\times}10^{-6}$ | $0.0027^\dagger$ | $1.91{\times}10^{-6}$ |
| | | No prior-mean shrinkage | 0.0116 | 0.0318 | 0.0094 |
| | | No covariance conditioning | 0.0457 | 0.1092 | 0.0331 |
| | | No stability regularizer | 0.2184 | $3.51{\times}10^{-5}$ | $2.10{\times}10^{-4}$ |

## B.5 Additional Visualizations

This appendix provides qualitative visualizations that complement the quantitative results in the main text. The visualizations are organized into synthetic-data examples and fMRI examples. For synthetic systems, the transition-matrix figures compare each estimated matrix against the known ground-truth matrix. For the fMRI benchmark, where no physical ground-truth transition matrix is available, the comparison is made against the ridge-based reference matrix $A_m^{\mathrm{ref}}$ defined in Appendix A.7. The rollout figures show representative open-loop predictions and illustrate how errors in the learned transition operator propagate over time.

### B.5.1 Synthetic Data

Figures 6 and 7 show representative stable-system visualizations. Figure 6 compares estimated transition matrices, the ground-truth transition matrix, and entrywise absolute errors. The single-task estimators and cross-task baselines recover the broad scale of the stable dynamics, but their estimates are visibly more attenuated or structurally biased than the ground truth. The PBML-LTI estimate is visually closer to the true transition matrix, with a more diffuse and lower-magnitude error pattern. The MAML-style LTI estimate is less consistent in this stable example, showing more pronounced entrywise deviations than PBML-LTI.

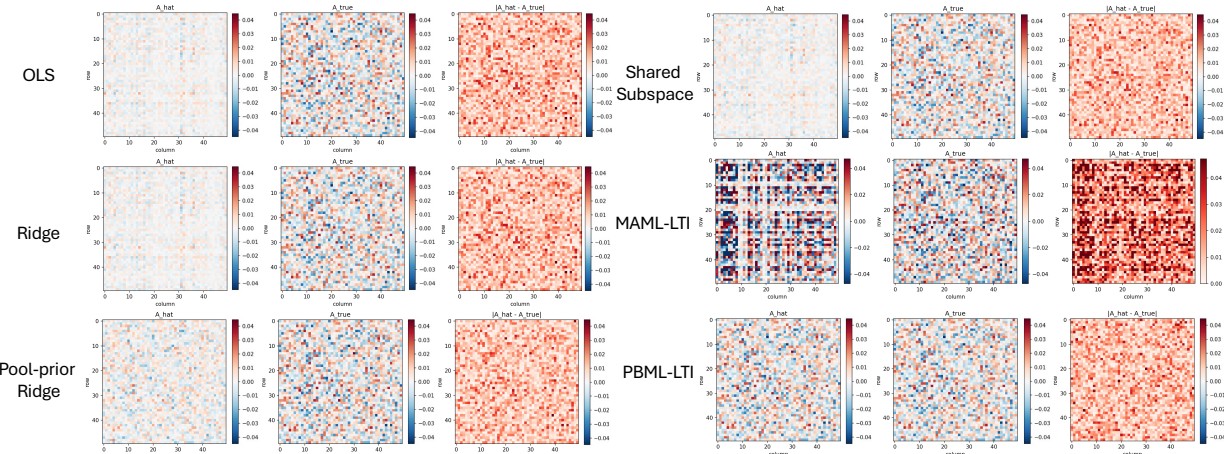

Figure 6: Extended stable synthetic transition-matrix visualization. The figure shows, for each method, the estimated transition matrix, the ground-truth transition matrix, and the entrywise absolute difference. PBML-LTI more closely matches the true transition structure, while several baselines exhibit stronger attenuation or structural bias.

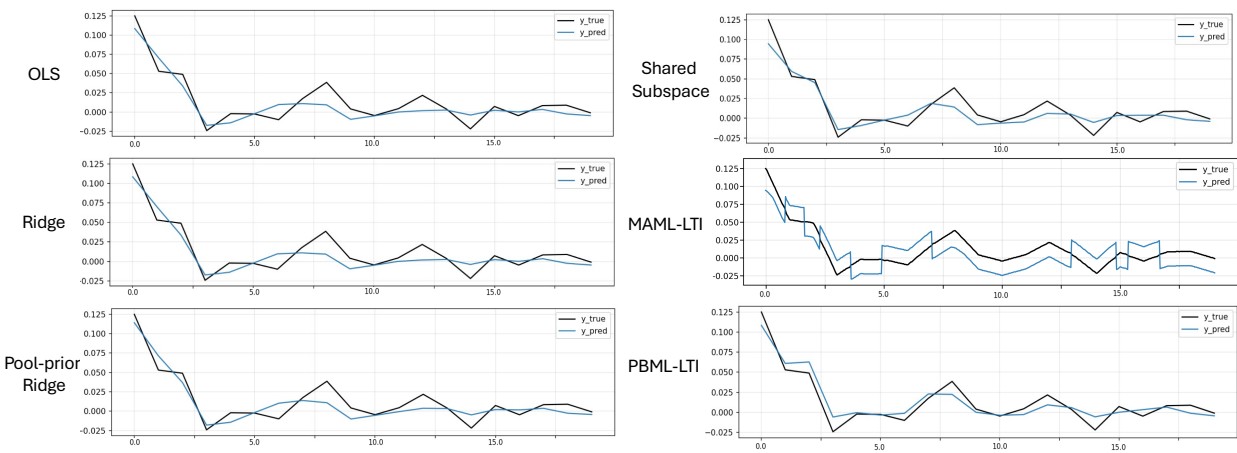

Figure 7: Stable synthetic open-loop rollout trajectories for a representative system. The ground-truth trajectory is shown in black and the predicted trajectory is shown in blue. Since the system is stable, rollout errors remain bounded over the short horizon, explaining why several methods can have similar $E_{\text{traj}}$ despite different transition-matrix errors.

Figure 7 shows the corresponding stable open-loop rollout behavior. Because the system is stable, rollout errors are damped over the short horizon and several methods track the ground-truth trajectory reasonably well. This explains why the stable-regime rollout metric can be numerically close across methods even when the transition-matrix error $E_A$ differs substantially. The visualization therefore supports the interpretation in the main text: in stable systems, $E_A$ is often the more discriminative diagnostic of parameter recovery, while $E_{\text{traj}}$ can remain similar across methods because errors do not rapidly amplify.

Figures 8 and 9 show the corresponding qualitative behavior in an unstable synthetic setting. In Figure 8, small-looking entrywise differences can still be important because unstable systems are governed by dominant spectral modes. Some methods produce transition matrices that appear reasonable in Frobenius norm but distort the dominant unstable directions. This is especially consequential for methods with strong structural bias, such as pooled shrinkage or a fixed low-dimensional subspace.

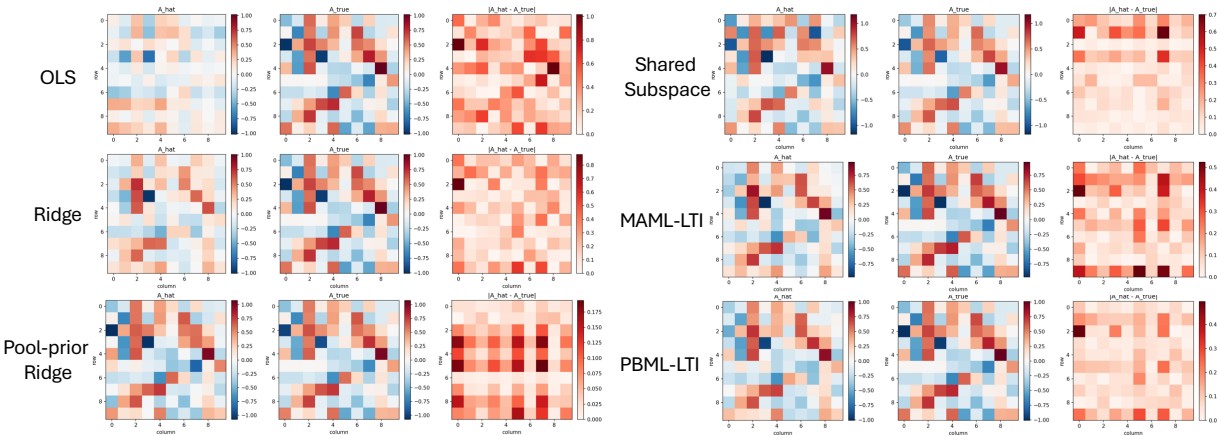

Figure 8: Extended unstable synthetic transition-matrix visualization. The figure shows estimated transition matrices, the ground-truth transition matrix, and entrywise absolute differences. In unstable systems, even localized or small-magnitude matrix errors can have large rollout consequences when they affect dominant spectral modes.

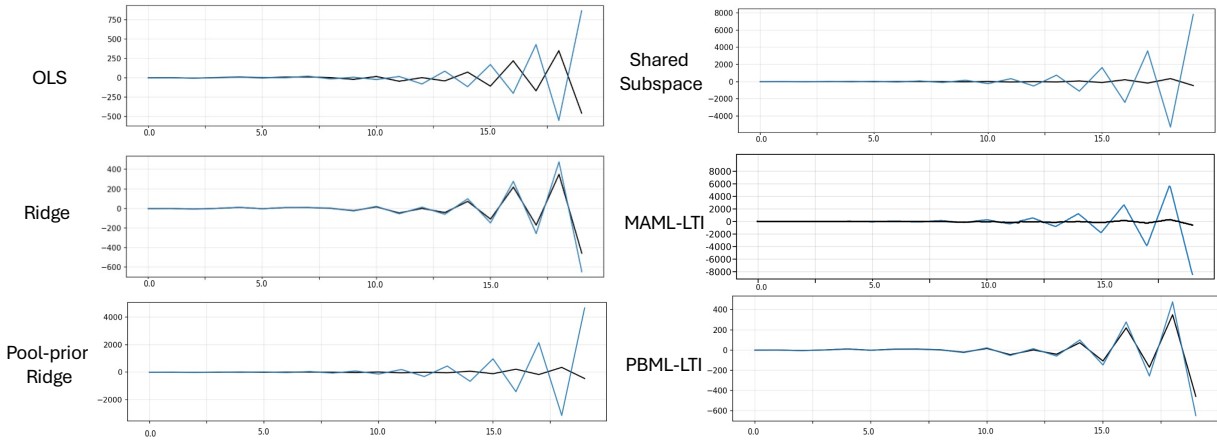

Figure 9: Unstable synthetic open-loop rollout trajectories for a representative system. The ground-truth trajectory is shown in black and the predicted trajectory is shown in blue. The figure illustrates how spectral errors in the estimated transition matrix can be amplified under repeated rollout, producing large trajectory deviations even when matrix errors appear moderate.

Figure 9 shows that these matrix-level differences can translate into dramatically different open-loop trajectories. In the unstable regime, errors are repeatedly multiplied by the estimated transition matrix, so even modest spectral misalignment can lead to rapidly growing prediction error. This visualization explains the decoupling observed in the quantitative results: a method can achieve a small Frobenius transition-matrix error while still producing poor rollout behavior if it misestimates the unstable eigenspace or the dominant eigenvalue. Conversely, a method with slightly larger Frobenius error may yield a more reliable rollout if it better controls the unstable modes.

### B.5.2 fMRI Data

Figures 10 and 11 provide qualitative diagnostics on the fMRI benchmark. Since fMRI does not provide a physical ground-truth transition matrix, Figure 10 compares each estimate against the ridge-based full-window reference matrix $A_m^{\text{ref}}$. The reference matrix should therefore be interpreted as a consistent evaluation

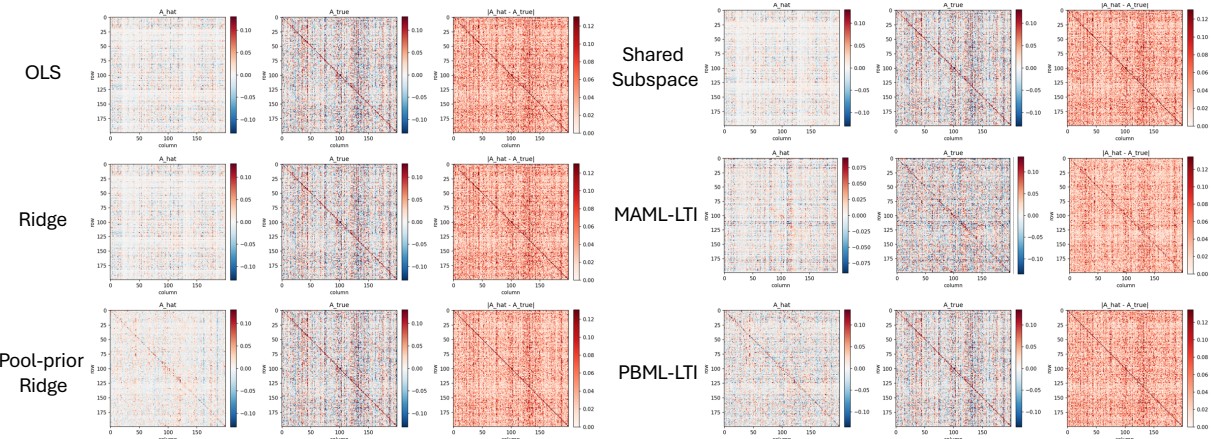

Figure 10: Extended fMRI transition-matrix visualization. Since fMRI does not provide a physical ground-truth transition matrix, each estimate is compared against the ridge-based full-window reference matrix $A_m^{\mathrm{ref}}$. The figure shows estimated matrices, the reference matrix, and entrywise absolute differences for all methods.

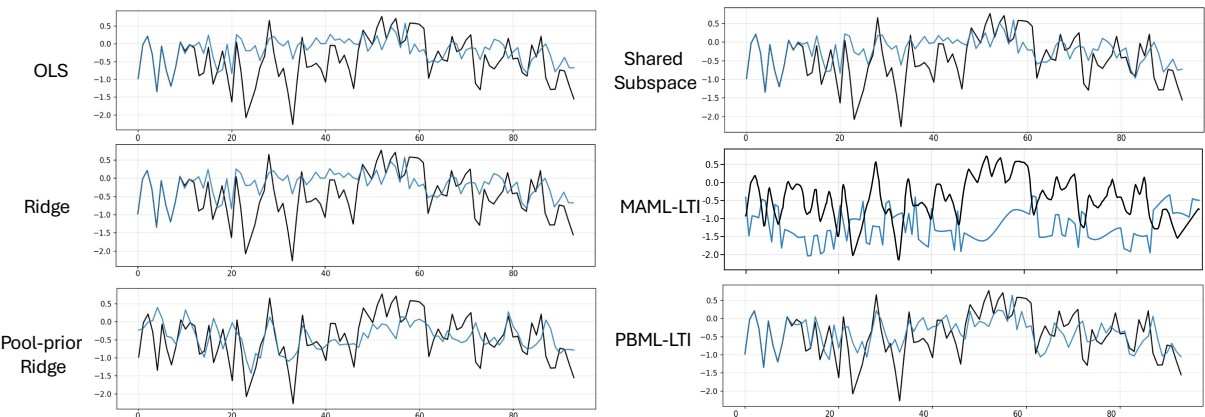

Figure 11: fMRI open-loop rollout trajectories for the representative task shown in Figure 10. The reference trajectory is shown in black and the predicted trajectory is shown in blue. The comparison illustrates how errors in the estimated transition operator affect multi-step prediction on a high-dimensional, noisy, approximately LTI real-data benchmark.

proxy, not as a true biological ground truth. Across methods, the absolute-difference heatmaps show substantial residual structure, which is expected because the fMRI windows are high-dimensional, short, noisy, and only approximately described by a linear time-invariant model.

The fMRI heatmaps also illustrate the practical role of prior-based adaptation. Single-task methods such as OLS and Ridge produce estimates that are broadly similar but require much longer support prefixes in the quantitative results. Shared Subspace uses shorter support but can impose a restrictive structural bias. MAML-LTI can produce more unstable or less well-aligned estimates in this high-dimensional setting. PBML-LTI provides a more favorable balance by using the learned prior to regularize adaptation while retaining task-specific flexibility through the posterior update.

Figure 11 shows open-loop rollout trajectories for the same representative fMRI setting. The rollout comparison highlights that matching the reference transition matrix entrywise is not the only determinant of predictive quality; the induced dynamics must also remain stable and spectrally well aligned over the rollout

Table 17: Performance comparison across methods in common-case and edge-case settings on marginally unstable systems ($\rho_0 = 1 + \epsilon$, $\epsilon \in (0, 1)$). $E_A$ is the mean $\pm$ standard deviation of the transition-matrix error, $E_{\mathrm{traj}}$ is the mean $\pm$ standard deviation of the rollout error, and $\overline{S}$ reports the average selected support length. Lower is better; boldface marks the lowest sample mean.

| Dimension | Setting | Metric | OLS | Ridge | Pooled-prior Ridge | Shared Subspace | MAML-LTI | PBML-LTI |
|---|---|---|---|---|---|---|---|---|
| 50 | Common-case | $E_A$ | $1.0845 \pm .017$ | $1.0795 \pm .017$ | $0.7181 \pm .011$ | $0.9144 \pm .051$ | $14.749 \pm .071$ | $\mathbf{0.2033 \pm .0045}$ |
| | | $E_{\mathrm{traj}}$ | $0.0260 \pm .0018$ | $0.0260 \pm .0019$ | $0.0260 \pm .0018$ | $0.0260 \pm .0018$ | $0.0282 \pm .0022$ | $\mathbf{0.0259 \pm .0018}$ |
| | | $\overline{S}$ | $2.00$ | $8.20$ | $3.05$ | $10.25$ | $1.85$ | $\mathbf{1.55}$ |
| | Edge-case | $E_A$ | $1.0849 \pm .015$ | $1.0799 \pm .016$ | $0.7182 \pm .011$ | $0.9065 \pm .049$ | $14.742 \pm .07$ | $\mathbf{0.2037 \pm .0041}$ |
| | | $E_{\mathrm{traj}}$ | $0.0258 \pm .0021$ | $0.0258 \pm .0021$ | $0.0257 \pm .0021$ | $0.0258 \pm .0021$ | $0.0279 \pm .0026$ | $\mathbf{0.0257 \pm .0021}$ |
| | | $\overline{S}$ | $2.00$ | $5.95$ | $3.60$ | $10.65$ | $2.00$ | $\mathbf{1.50}$ |
| 25 | Common-case | $E_A$ | $0.9510 \pm .024$ | $0.9238 \pm .023$ | $0.7258 \pm .018$ | $0.7779 \pm .055$ | $3.6309 \pm .046$ | $\mathbf{0.0390 \pm .0023}$ |
| | | $E_{\mathrm{traj}}$ | $0.0136 \pm .0018$ | $0.0136 \pm .0018$ | $0.0136 \pm .0019$ | $0.0136 \pm .0018$ | $0.0140 \pm .0019$ | $\mathbf{0.0135 \pm .0018}$ |
| | | $\overline{S}$ | $2.15$ | $12.50$ | $12.45$ | $11.65$ | $\mathbf{1.20}$ | $\mathbf{1.20}$ |
| | Edge-case | $E_A$ | $0.9423 \pm .023$ | $0.9170 \pm .026$ | $0.7239 \pm .016$ | $0.7773 \pm .078$ | $3.6329 \pm .045$ | $\mathbf{0.0394 \pm .0026}$ |
| | | $E_{\mathrm{traj}}$ | $0.0125 \pm .0012$ | $0.0125 \pm .0012$ | $0.0124 \pm .0011$ | $0.0125 \pm .0011$ | $0.0128 \pm .0012$ | $\mathbf{0.0123 \pm .0011}$ |
| | | $\overline{S}$ | $2.20$ | $13.10$ | $13.15$ | $9.10$ | $\mathbf{1.20}$ | $1.30$ |
| 10 | Common-case | $E_A$ | $0.5788 \pm .066$ | $0.5185 \pm .065$ | $0.0743 \pm .0052$ | $0.0860 \pm .021$ | $0.4918 \pm .022$ | $\mathbf{0.0051 \pm .00066}$ |
| | | $E_{\mathrm{traj}}$ | $0.0057 \pm .0011$ | $0.0057 \pm .0011$ | $0.0056 \pm .0011$ | $\mathbf{0.0056 \pm .0011}$ | $0.0057 \pm .001$ | $\mathbf{0.0056 \pm .001}$ |
| | | $\overline{S}$ | $2.90$ | $10.40$ | $9.45$ | $7.05$ | $\mathbf{1.00}$ | $1.20$ |
| | Edge-case | $E_A$ | $0.5609 \pm .059$ | $0.4868 \pm .06$ | $0.0746 \pm .0053$ | $0.0891 \pm .027$ | $0.4912 \pm .022$ | $\mathbf{0.0053 \pm .00082}$ |
| | | $E_{\mathrm{traj}}$ | $0.0060 \pm .0012$ | $0.0060 \pm .0013$ | $0.0059 \pm .0011$ | $0.0059 \pm .0012$ | $0.0061 \pm .0012$ | $\mathbf{0.0059 \pm .0012}$ |
| | | $\overline{S}$ | $3.15$ | $9.85$ | $8.55$ | $10.30$ | $\mathbf{1.00}$ | $1.15$ |

horizon. The MAML-style rollout is visibly less stable in this example, while the other methods track the overall trajectory more closely. This supports the main-text observation that PBML-LTI obtains the best average fMRI rollout error while also using the shortest average support prefix.

## B.6 Marginally Unstable System Results

Table 17 reports results for the marginally unstable environment with $\rho_0 = 1 + \epsilon$, $\epsilon \in (0, 1)$. This setting lies just beyond the nominal stability threshold and is intended to test whether the behavior observed in the stable regime changes abruptly once $\rho_0$ exceeds one. Overall, the results show that the marginally unstable regime behaves much more like a mild perturbation of the stable regime than like the strongly unstable stress test.

Across all dimensions and both common-case and edge-case subsets, PBML-LTI achieves the lowest transition-matrix error $E_A$. The gains are substantial. For example, at $d = 50$, PBML-LTI reduces $E_A$ from roughly 1.08 for OLS/Ridge, 0.72 for Pooled-prior Ridge, 0.91 for Shared Subspace, and 14.7 for MAML-LTI to about 0.20. At $d = 25$, PBML-LTI reduces $E_A$ from roughly 0.72–0.95 for the classical and subspace baselines and 3.63 for MAML-LTI to about 0.039. At $d = 10$, PBML-LTI reaches $E_A \approx 0.005$, substantially below all baselines. These results indicate that the learned matrix-normal prior continues to provide an effective inductive bias immediately beyond the stability threshold.

The rollout metric $E_{\mathrm{traj}}$ is much less separated across methods in this marginally unstable regime. For all three dimensions, the rollout errors of OLS, Ridge, Pooled-prior Ridge, Shared Subspace, and PBML-LTI are nearly tied up to the reported precision, with PBML-LTI attaining the lowest or tied-lowest mean in most settings. This mirrors the stable-regime behavior: over the short query horizon, modest marginal instability does not yet induce the severe finite-horizon amplification seen in the strongly unstable experiments. By contrast, MAML-LTI has noticeably worse transition-matrix recovery and slightly worse rollout performance, despite often selecting very short support prefixes.

The support-length results further clarify the role of adaptive support selection. PBML-LTI uses very short support prefixes, with $\overline{S}$ between 1.15 and 1.55 in most settings, and achieves the best transition recovery at those support lengths. In some cases, MAML-LTI selects an equally short or shorter prefix, but this does not translate into competitive $E_A$. Ridge and Pooled-prior Ridge often use much longer prefixes, especially for $d = 25$ and $d = 10$, but still have substantially larger transition-matrix error than PBML-LTI. Thus, the

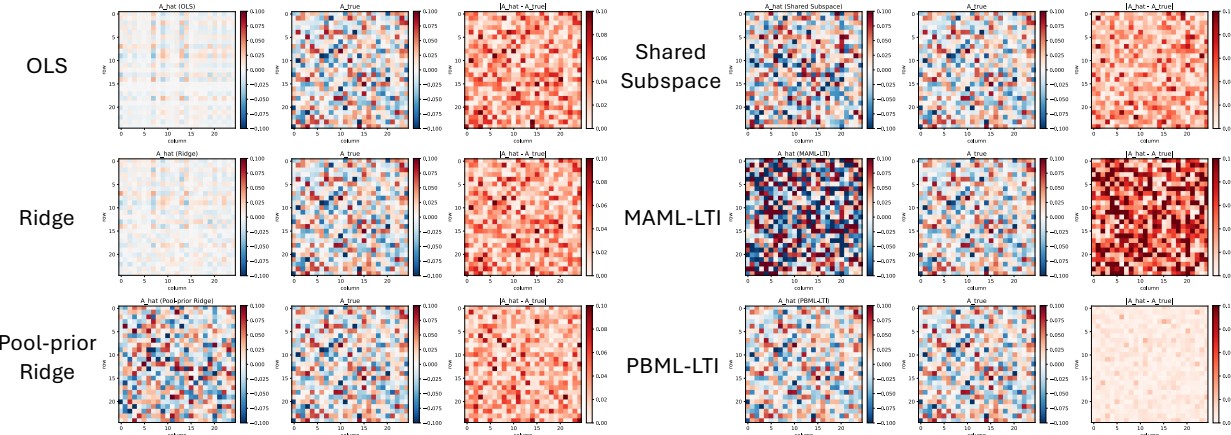

Figure 12: Transition-matrix estimates for a representative marginally unstable test system. For each method, the figure shows the estimated transition matrix, the ground-truth transition matrix, and the entrywise absolute error. PBML-LTI more closely matches the ground-truth transition structure, while single-task estimators are more attenuated and MAML-LTI exhibits larger entrywise deviations.

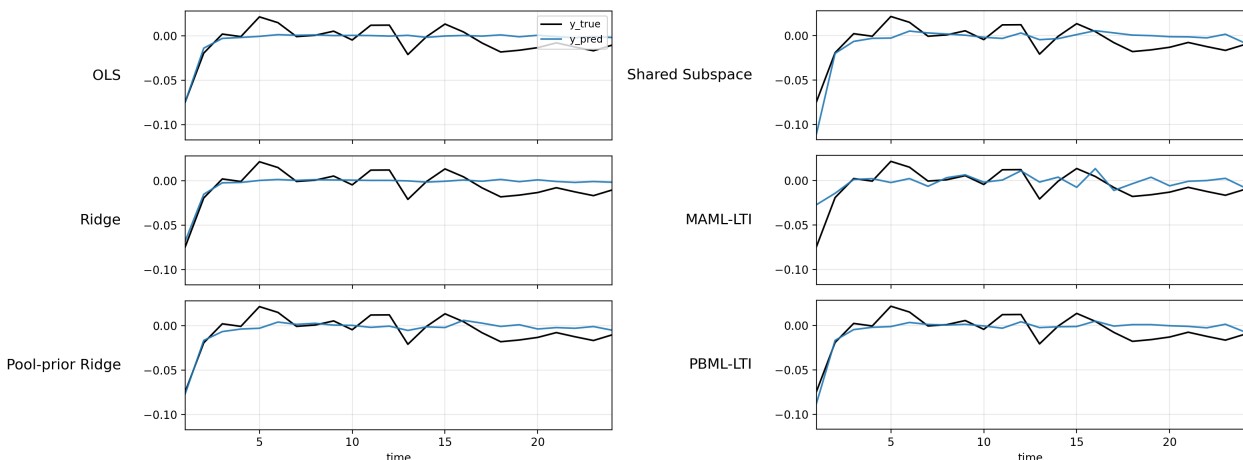

Figure 13: Open-loop rollout trajectories for the representative marginally unstable system shown in Figure 12. The black curve denotes the ground-truth trajectory and the blue curve denotes the rollout generated by each estimated transition matrix. Over the short query horizon, the rollouts remain bounded and visually close across several methods, explaining why $E_{\mathrm{traj}}$ is much less discriminative than $E_A$ in the marginally unstable regime.

marginally unstable results support the same data-efficiency conclusion as the stable results: PBML-LTI can adapt accurately from very short prefixes when the learned prior is well aligned with the task family.

Figures 12 and 13 provide qualitative diagnostics for a representative marginally unstable test task. The transition-matrix heatmaps show that PBML-LTI more closely matches the ground-truth transition structure than the more attenuated single-task estimators and the noisier MAML-style estimate. The rollout plots show that all non-catastrophic methods remain bounded over the short horizon, consistent with the small differences in $E_{\mathrm{traj}}$ reported in Table 17. Together, the marginally unstable results show that crossing $\rho = 1$ does not by itself cause an abrupt failure; the practical degradation depends on the magnitude of finite-horizon spectral amplification.

