# OpenReview forum: "PAC-Bayesian Meta-Learning for Few-Shot Identification of Linear Dynamical Systems"
_TMLR — Decision pending for TMLR_

### Review · Reviewer_71N6 · 2026-03-25

**Summary Of Contributions:**

This submission studies the problem of identifying the transition matrices for linear time-invariant (or LTI) dynamical systems, focusing specifically on "few-shot" identification where the goal is to perform the identification (learning) given a short trajectory prefix, and access to a set of trajectories from "related" LTI systems. The notion of relatedness is defined based on a hierarchical generative process.

Viewing the LTI system identification problem as a multi-input multi-output supervised regression problem, the proposed solution is based on meta-learning, where the meta-learning objective for this LTI system identification is motivated through an existing hierarchical Bayesian formulation and a novel PAC-Bayesian bound, and the adaptation to a new task is through a closed-form posterior. The novelty of the PAC-Bayesian bound is the adaptation of standard PAC-Bayesian analysis to this non-IID (independent and identically distributed) setting where the "examples" in the above supervised regression problem are no longer IID, but rather temporally dependent. The bound is also used to obtain an upperbound on the rollout error -- the difference between the estimated LTI system trajectory and the ground truth trajectory.

The proposed scheme is evaluated empirically on two sets of synthetic datasets, one for stable systems and another form unstable systems, where the ground truth LTI system is known. Another set of experiments evaluate the proposed scheme on a very practically motivated problem with fMRI data. The proposed scheme is compared against various regression baselines, and the results show that the proposed scheme is able to better estimate the transition matrix, or have better trajectory rollout error under various conditions.


Strengths:

- **[S1]** The submission studies a very important problem of LTI system identification in the few-shot (short trajectory prefix) setting, and proposes a solution that not provides point estimates, but also allows for uncertainty quantification, which is often critical in LTI systems.
- **[S2]** The submission clearly describes the problem setup, and has a very well motivated development of the meta-learning objective utilizing novel theoretical analysis under the PAC-Bayes framework.
- **[S3]** The proposed solution admits a simple adaptation scheme for any new task with a closed-form expressions for the posterior given the short prefix trajectory (few-shot data), obviating the need for gradient descent that is standard in usual meta-learning solutions.
- **[S4]** The empirical evaluation considers two different forms of test tasks (common vs edge-case) both for stable and unstable systems, allowing the evaluation of the robustness of the considered schemes across test tasks.


Weaknesses:

- **[W1]** In my opinion, one of the main weaknesses of the submission is the lack of a clear connection between the theoretical results in Theorem 3 (and Corollary 1) and any guarantees for few-shot adaptation. The loss on left-hand side of (18) is the expectation (over the posterior) of the conditional expectated loss at time $t$ averaged over all time steps $t \in [T]$. It is not clear how that loss connects to the few-shot adaptation loss, where the support trajectory is up to some time $T_{\text{sup}}$ used to generate the posterior, and we care about the loss from time $t = T_{\text{sup}} + 1, \ldots, T_m$. It is not clear what the bound in (18) says about the few-shot adaptation as both the left and right sides are considering the same trajectory time steps, where we can optimize for the RHS (right hand side) during the adaptation. However, the LHS (left-hand side) should be about future time steps --- this is not an issue with supervised learning because of the IID assumption and thus can extend to any number of examples; here the number of steps $T$ serve as the number of example. However, in the absence of the IID assumption this novel result does not appear to directly tell us much. Similar issue holds for $L_T(Q)$ and $\hat{L}_T(Q)$ in equation (29) for Corollary 1.
  - **[W1a]** Given the closed form of $Q_{m, \phi}$ from equations (12) and (13), is it not possible to get a bound in equation (27) or (29) that is specifically about $Q_{m, \phi}$ (which is being used in PBML-LTI) instead of a generic $Q$ and generic prior $P$?
  - **[W1b]** The main focus of this submission seems to be the LTI system identification. If by identification, we mean the few-shot estimation of the transition matrix $A_m$, then it would be useful to have bounds on $E_A$ defined in equation (37) in the empirical evaluation. To the best of my understanding, the results in Theorem 3 and Corollary 1 correspond to bounds on rollout error (as in  equation (39)).


- **[W2]** Another weakness is that the empirical evaluation is unable to clearly demonstrate the benefits of meta-learning. For example, the results in Table 1 seem unintuitive, potentially making the $E_{\text{traj}}$  or $E_A$ metric hard to interpret:
  - High values of $E_A$ (loss in estimation of the transition matrix) for the baselines do not appear to translate to high values of $E_{\text{traj}}$ (trajectory rollout error) -- more than $4 \times$ smaller $E_A$ amounts to $<1\%$ improvement in the $E_{\text{traj}}$ in both common and edge cases.
  - Pool-prior Ridge (which uses data from all the available tasks) performs significantly better than per-task OLS/Ridge for $d = 10$, and slightly better for $d = 50$, but appears worse for $d = 25$, appearing unintuitive that performance should get worse then better, warranting better explanation.
  - Similar issues are present in the results in Table 2, especially in cases where the $E_A$ value is significantly smaller for the proposed PBML-LTI scheme, but the $E_{\text{traj}}$ values of some other baselines with higher $E_A$ values is better (for example with $d=25$).
  - An explanation is provided for the stable systems, but then it is not clear whether either of the metrics ($E_A$ or $E_{\text{traj}}$) are useful metrics to compare methods in this setup. Furthermore, these explanations do not seem to apply to the similar issues in Table 2.
  - Results in Table 3 appear unintuitive:
    - This is no clear understanding of how $E_A$ is computed. If $E_A$ is computed using the $A_m^{\text{ref}}$ described in equation (46) using Ridge, it is not clear why the "Ridge" column in Table 3 does not have the lowest value of $E_A$.
    - The standard deviations are much larger than the means, making these results somewhat meaningless. Given these metrics are squared norms, these values are not valid.
  - Finally, a critical metric in all these evaluation is the "average support length" $\overline{S}$. However, the procedure for selecting the per-task support length $S_m$ for the proposed scheme and the baselines is not discussed anywhere (to the best of my understanding; I have also checked the appendix). Neither Algorithm 1 nor the description of the baselines in Section 5.1 discusses the adaptive support length selection scheme. I think it is important to discuss this metric better.

**Audience:**

Yes

**Audience Explanation:**

This submission studies are a very important problem of few-shot LTI system identification with uncertainty quantification. I think this is a very important problem, and would be of interest to a significant portion of the TMLR audience (assuming appropriate updates are made to the submission to sufficiently support the claims).

**Broader Impact Concerns:**

No broader impact concerns.

**Claims And Evidence:**

No

**Claims Explanation:**

The main claims of this submission involve the extension of the PAC-Bayesian meta-learning framework to a non-IID setting of LTI dynamical system identification, and the development of a meta-learning scheme which is capable of few-shot adaptation. The novel theoretical results do consider a non-IID setup.
- However, as I discuss in weakness **[W1]**, it is not clear to me how the main theoretical results presented in this paper translate to guarantees for the goal of few-shot adaptation.
- Furthermore, the empirical results appear to be somewhat unintuitive as I discuss in weakness **[W2]**, which is not able to clearly demonstrate the value of the proposed meta-learning scheme.

Therefore, I do not believe that the claims of this paper are sufficiently supported by the presented evidence (both theoretical and empirical).

**Requested Changes:**

To me, the following are somewhat important changes:

- It would be great to have a concrete explicit connection between the few-shot adaptation loss and the guarantees presented in Theorem 3 and Corollary 1. For example, bounding expected future trajectory loss $\sum_{t=T_{\text{supp}} + 1}^T \mathbb E_{A \sim Q_{m, \phi} } [ \mathbb E[ \ell_t(A) | \mathcal F_{t-1}] ]$  with the empirical support trajectory loss $\sum_{t=1}^{T_{\text{supp}}} \mathbb E_{A \sim Q_{m, \phi} } [ \ell_t(A) ]$, or show how the $L_T(Q)$ is related to the future trajectory.
- It is critical to explain how the per-task support length $S_m$ is selected for each of the methods. Alternately (and a better option in my opinion), we should control for the support length $T_{\text{supp}}$ across all baselines (for the test tasks, and maybe even for the training tasks), and compare the error metrics $E_A/E_{\text{traj}}$ for a given support length. In this case, we can consider different support lengths, and instead of a single table, we can even have a figure where the horizontal axis corresponds to the support length while the vertical axis presents the error metrics. If the proposed scheme is actually useful for smaller support lengths (demonstrating quicker adaptation), this will be clear in the figure.


In my opinion, the following changes will be beneficial to the submission, but are not necessary to demonstrate technical correctness:

- While the discussion following Corollary 1 claims that it makes explicit the role of meta-learning but tying the result to the quality of the (learned) prior $P$, it would be nice to have a more technical presentation of this connection. Furthermore, usual meta-learning results also incorporate the effect of the number of training tasks $M_{\text{tr}}$ (in equation (26)) available. It would be useful to have a similar result connecting the quality of the learned prior to the number of training tasks.
- A explicit discussion highlighting the nontrivial analytical steps needed to extend the PAC-Bayes LTI analysis of Sarkar & Rakhlin (2019) to the few-shot meta-learning setup would be very beneficial to readers.
- Can we detail how the posterior expectation in (21) and the KL divergence in (22) efficiently computed given that they correspond respectively to an expectation over the posterior, and a KL divergence computation between high dimensional distributions.
- In section 5.4 where we evaluate the proposed scheme on fMRI data, why does this choice of "we segment the ROI time series into overlapping windows of length $L$ with a stride of $s$" give us "related identification tasks"? Can this be explained better? On a related note, what are the "BIDS task labels"?
- The original MAML algorithm (Finn et al. 2017) was developed for both supervised learning (with the IID assumption within tasks) as well as reinforcement learning, where the within-task IID assumption does not hold. From that perspective, the novelty of "non-IID assumption within tasks" can be seen as somewhat limited. So "handling temporally dependent trajectory data" (as mentioned in "Meta-learning and hierarchical Bayes" paragraph in page 3) is not specific to this paper, though the focus on LTI systems might be. This should be appropriately qualified in the related works section, highlighting what precisely is novel here.
- Minor typo: Page 2, Paragraph "However, most existing PAC-Bayes meqt-learning rely ...", meqt -> meta.
- Minor: Why is ignoring the noise term in the rollout for computing the trajectory rollout error (equations 38-39) justified since the ground truth rollout includes the noise term? One can instead perform multiple rollouts and average the trajectory rollout error over the noise distribution (as the noise distribution is known for the synthetic problems). Can this be better explained? Alternately (probably more intensive), the evaluation can be instead modified to do multiple rollouts (for both the ground truth and the estimated systems), with the noise term incorporated in both, and we average the performances across the rollouts.

---

> ### Author Response · Authors · 2026-04-09
> **Response to W1 (1)**
>
> We thank the reviewer for their careful reading and constructive feedback. We agree that the manuscript should more clearly delineate the distinct roles of our theoretical and empirical contributions, specifically: (i) what the martingale PAC-Bayes analysis formally establishes, (ii) how that analysis motivates the fit--KL surrogate objective used by PBML-LTI, and (iii) what is exactly evaluated in the empirical support--adapt--query protocol.
>
> To address this, our central clarification in the revision is that PBML-LTI is designed as a PAC-Bayes-motivated meta-learning method. The martingale PAC-Bayes analysis yields a principled fit--complexity tradeoff under temporal dependence, which directly motivates our surrogate training objective.
>
> To make the connection between the theory and the few-shot protocol explicit, we have added a predictive-risk PAC-Bayes bound on the future query suffix following adaptation. Furthermore, we now demonstrate how this predictive bound induces problem-specific corollaries for three key metrics: (i) projected transition-matrix error, (ii) full Frobenius transition-matrix error under a query excitation condition, and (iii) deterministic trajectory rollout error via a finite-horizon growth factor. We believe these additions establish a substantially more direct and transparent link between our theoretical foundations, the few-shot evaluation protocol, and the reported empirical results.
>
> Response to [W1]: Connection between Theorem 3 / Corollary 1 and few-shot adaptation
> We agree that the original presentation of Theorem 3 and Corollary 1 was overly generic, and that the connection to few-shot support--query adaptation required explicit clarification. In the empirical protocol, the few-shot pattern already takes the form “support → adapt → query”: the posterior is derived from a support prefix, and predictive performance is evaluated on a held-out future query suffix. What the original draft lacked was an explicit specialization of the generic martingale PAC-Bayes result to this specific setting. In the revised manuscript, we have added this specialization. Further, we clarify that PBML-LTI optimizes a PAC-Bayes-motivated surrogate, rather than directly optimizing the exact support--query bound.
>
> For a task $m$, let
>
> $$D_m^{\mathrm{sup}} = \{x_{m,t}\}_{t=0}^{S_m}$$
>
> denote the support trajectory, and let
>
> $$Q_{m,\phi}^{\mathrm{sup}} := p(A_m \mid D_m^{\mathrm{sup}}, \phi)$$
>
> be the posterior derived using only this support prefix. Conditioning on the support filtration $\mathcal{F}_{m,S_m}$ fixes the posterior, and Corollary 1 can then be applied to the shifted filtration on the future query suffix:
>
> $$\mathcal{G}u := \mathcal{F}{m,S_m+u}, \qquad u = 0, \dots, Q_m.$$
>
> Writing
>
> $$X_{m,q} := [x_{m,S_m}, \dots, x_{m,S_m+Q_m-1}], \qquad Y_{m,q} := [x_{m,S_m+1}, \dots, x_{m,S_m+Q_m}],$$
>
> we define the support--query predictive risk
>
> $$R_{m,q}^{\mathrm{pred}}(Q) = \frac{1}{Q_m} \sum_{u=1}^{Q_m} \mathbb{E}{A \sim Q} \left[ \mathbb{E} \left[ \|x{m,S_m+u} - A x_{m,S_m+u-1}\|2^2 \mid \mathcal{G}{u-1} \right] \right]$$
>
> with empirical counterpart
>
> $$\hat{R}\_{m,q}^{pred}(Q) = (1/Q\_m) E\_Q [ ||Y\_{m,q} - A X\_{m,q}||\_F^2 ].$$
>
> Then Corollary 1 yields the support--query version
>
> $$R_{m,q}^{\mathrm{pred}}(Q_{m,\phi}^{\mathrm{sup}}) \le \widehat{R}{m,q}^{\mathrm{pred}}(Q{m,\phi}^{\mathrm{sup}}) + \frac{2\sigma^2}{\lambda Q_m} \left( \mathrm{KL}(Q_{m,\phi}^{\mathrm{sup}} \| P_{\phi}) + \log(1/\delta) \right) + \sigma^2 \lambda v =: \mathcal{B}_m^{\mathrm{pred}}.$$
>
> This is a PAC-Bayes bound whose left-hand side is the predictive risk on future query time steps after adaptation. We now state this support--query specialization explicitly in the paper.
>
> Relation to support empirical loss
>
> The reviewer also raises the question of whether future query loss can be bounded directly in terms of empirical support loss. In the temporally dependent setting, this does not follow without additional assumptions linking the support and query segments, for example stronger stationarity, mixing, or excitation assumptions. We therefore do not claim a direct support-loss-to-future-loss guarantee in the current paper. Instead, the natural few-shot interpretation of the theorem is the support--query bound above, where the posterior is built from the support prefix and the predictive risk is evaluated on future time steps.

---

> ### Author Response · Authors · 2026-04-09
> **Response to W1a (2)**
>
> We agree that the generic prior/posterior statement requires explicit specialization, given that PBML-LTI employs a closed-form matrix-normal posterior. In the specific formulation utilized by PBML-LTI, the prior and posterior can be expressed as
>
> $$ P_{\phi}=\mathcal{MN}(W,I_d,V), \qquad Q_{m,\phi}^{\mathrm{sup}}=\mathcal{MN}(M_m,I_d,V_m). $$
>
> Under this specialization, the KL term admits the closed-form expression
>
> $$ \mathrm{KL}(Q_{m,\phi}^{\mathrm{sup}}\|P_{\phi}) = \frac{1}{2} \left[ d\big(\mathrm{tr}(V^{-1}V_m)-\log\det(V^{-1}V_m)-d\big) + \mathrm{tr}\big((M_m-W)V^{-1}(M_m-W)^\top\big) \right]. $$
>
> Moreover, the posterior-expected support-query squared loss also admits a closed-form expression:
>
> $$ \hat{R}\_{m,q}^{\mathrm{pred}}(Q{m,\phi}^{\mathrm{sup}}) = \frac{1}{Q_m} \left[ \lVert Y_{m,q}-M_m X_{m,q} \rVert_F^2 + d\,\mathrm{tr}(V_m X_{m,q}X_{m,q}^\top) \right]. $$
>
> Since the squared Frobenius loss is convex in $A$, Jensen's inequality implies
>
> $$ \frac{1}{Q_m}\lVert Y_{m,q}-M_m X_{m,q} \rVert_F^2 \le \widehat{R}{m,q}^{\mathrm{pred}}(Q{m,\phi}^{\mathrm{sup}}). $$
>
> Accordingly, the empirical query-fit term entering the support-query PAC-Bayes bound directly controls the posterior-mean predictor $\widehat{A}_m=M_m$ used by PBML-LTI. In the paper, this specialization follows directly from the closed-form matrix-normal formulas now stated in the method section together with the support-query notation introduced in the theory section.

---

> ### Author Response · Authors · 2026-04-09
> **Response to W1b - Part 1 (3)**
>
> ### Response to [W1b]: Extending the PAC-Bayes analysis to transition-matrix and trajectory error
>
> We agree that the distinction between predictive-risk control and transition-matrix recovery requires more explicit clarification. In the original draft, $E_A$ was included primarily as a complementary empirical diagnostic of parameter recovery, whereas Theorem 3 / Corollary 1 were stated as predictive-risk results. In the revised manuscript, we bridge this gap by deriving new corollaries directly from the support--query predictive bound stated above, explicitly extending our theoretical guarantees to encompass both transition-matrix estimation and trajectory rollout error.
>
> First, under the true dynamics
>
> $$Y_{m,q} = A_m X_{m,q} + W_{m,q},$$
>
> the conditional expected query residual at the posterior mean decomposes into a projected transition-matrix term plus a noise floor. Combining this with the Jensen step above yields
>
> $$E\_{A,proj}(m) := (1/Q\_m) ||(A'\_m - A\_m) X\_{m,q}||\_F^2 \leq R\_{m,q}^{pred}(A'\_m) \leq R\_{m,q}^{pred}(Q\_{m,\phi}^{sup}) \leq B\_m^{pred}.$$
> This quantity is always well defined and directly measures matrix error along the query excitation directions.
>
> Second, under an additional excitation condition on the query design matrix,
>
> $$\\lambda_{\\min}\\!\\left(\\frac{1}{Q_m}X_{m,q}X_{m,q}^\\top\\right)\\ge \\kappa_m > 0,$$
>
> we obtain a full Frobenius transition-matrix bound
>
> $$E_A(m) = \\|\\widehat A_m-A_m\\|_F^2 \\le \\frac{\\mathcal B_m^{\\mathrm{pred}}}{\\kappa_m}.$$
>
> We note that in regimes where $Q_m<d$, this full Frobenius bound may be vacuous because the query Gram matrix can be rank deficient. In such settings, the projected bound above remains the meaningful matrix-error extension.
>
> Third, writing the deterministic rollout error as
>
> $$E_{\\mathrm{traj}}(m) := \\sum_{u=1}^{Q_m}\\|\\widehat x_{m,S_m+u}-x_{m,S_m+u}\\|_2^2,$$
>
> and defining the finite-horizon growth factor
>
> $$H_{m,Q_m}:=\\sum_{r=0}^{Q_m-1}\\|\\widehat A_m\\|_2^r,$$
>
> we obtain the trajectory extension
>
> $$\\mathbb{E}\\!\\left[E_{\\mathrm{traj}}(m)\\mid \\mathcal F_{m,S_m}\\right] \\le Q_m^2\\,H_{m,Q_m}^2\\,\\mathcal B_m^{\\mathrm{pred}}.$$
>
> The factor $Q_m^2$ appears because $\\mathcal B_m^{\\mathrm{pred}}$ is a *per-step average* predictive bound, whereas $E_{\\mathrm{traj}}(m)$ is a cumulative multi-step error.
>
> To assess whether these PAC-Bayes-derived corollaries are numerically meaningful, we evaluated the resulting positive bounds for PBML-LTI on both the stable and unstable synthetic benchmarks. The table below reports the empirical mean and standard deviation of the transition-matrix bound and the trajectory bound.
>
> | Regime | Dimension | Setting | A_bound | Traj_bound |
> |---|---:|---|---:|---:|
> | Stable | 50 | Common-case | 0.410 ± 0.059 | 17.744 ± 2.660 |
> | Stable | 50 | Edge-case | 0.418 ± 0.063 | 18.133 ± 2.781 |
> | Stable | 25 | Common-case | 0.172 ± 0.028 | 9.473 ± 1.547 |
> | Stable | 25 | Edge-case | 0.178 ± 0.019 | 9.757 ± 1.051 |
> | Stable | 10 | Common-case | 0.066 ± 0.008 | 4.550 ± 0.622 |
> | Stable | 10 | Edge-case | 0.066 ± 0.011 | 4.553 ± 0.797 |
> | Unstable | 50 | Common-case | 1.077 ± 0.143 | 28.807 ± 7.972 |
> | Unstable | 50 | Edge-case | 1.100 ± 0.124 | 39.841 ± 6.204 |
> | Unstable | 25 | Common-case | 0.748 ± 0.175 | 78.494 ± 30.805 |
> | Unstable | 25 | Edge-case | 0.770 ± 0.135 | 86.887 ± 22.728 |
> | Unstable | 10 | Common-case | 1.277 ± 0.143 | 117.382 ± 28.807 |
> | Unstable | 10 | Edge-case | 1.371 ± 0.344 | 132.271 ± 38.807 |
>
> The table illustrates a sharp contrast between the stable and unstable regimes. In the stable setting, both $A_{\\mathrm{bound}}$ and $\\mathrm{Traj}_{\\mathrm{bound}}$ are finite, moderate in magnitude, and fairly stable across common-case and edge-case subsets. Consequently, the stable systems consistently yield substantially tighter bounds than the unstable systems. Within the stable regime, both the matrix and trajectory bounds decrease as the dimension decreases, which aligns theoretically with the expectation that lower-dimensional stable systems are easier to statistically identify given a fixed support budget.

---

> ### Author Response · Authors · 2026-04-09
> **Response to W1b - Part 2 (4)**
>
> In contrast, the unstable regime produces uniformly looser matrix bounds and much larger trajectory bounds. This is expected from the trajectory extension above, since the trajectory consequence contains the finite-horizon growth factor
>
> $$H_{m,Q_m}=\\sum_{r=0}^{Q_m-1}\\|\\widehat A_m\\|_2^r.$$
>
> When the learned dynamics are unstable, the powers $\\|\\widehat A_m\\|_2^r$ grow geometrically, so even a finite predictive bound $\\mathcal B_m^{\\mathrm{pred}}$ can be substantially amplified under rollout. In this sense, the matrix bound may remain finite and still informative, while the trajectory bound becomes much looser and can approach vacuity as instability increases.
>
> Further, the synthetic data generator intrinsically induces a dimension-dependent stability crossover. Because the entries of the base random matrix are drawn with a standard deviation proportional to $1/d$, the resulting spectral radius scales roughly as $1/\\sqrt d$. Consequently, the effective transition scale behaves approximately like
>
> $$\\rho(W_\\star)\\asymp \\frac{\\alpha}{\\sqrt d},$$
>
> so for a fixed nominal generation scale $\\alpha$, smaller systems can lie deeper in the unstable regime than larger ones. This is consistent with the pattern observed in the table, where the unstable setting exhibits progressively looser trajectory bounds as the effective instability becomes more severe.
>
> The relatively narrow margin between common-case and edge-case values, juxtaposed against the sharp disparity between stable and unstable regimes, indicates that system stability is the primary factor dictating the empirical tightness of our PAC-Bayes-derived corollaries. Taken together, these findings demonstrate that the predictive PAC-Bayes analysis provides numerically meaningful guarantees for both matrix recovery and trajectory rollouts within stable regimes, while simultaneously explaining the progressive deterioration of the trajectory bounds under unstable dynamics.

---

> ### Author Response · Authors · 2026-04-09
> **Response to W2 (5)**
>
> ### Response to [W2]: Interpretation of the empirical metrics
>
> We agree that the empirical section requires a more rigorous explanation as to why $E_A$ and $E_{\\mathrm{traj}}$ can yield divergent method rankings.
>
> **Why can large differences in $E_A$ correspond to modest differences in $E_{\\mathrm{traj}}$?**
>
> The Frobenius error
>
> $$E_A=\\|\\widehat{A}_m-A_m\\|_F^2$$
>
> measures an average entrywise discrepancy of the transition matrix. In contrast, the rollout metric depends on repeated application of the estimated dynamics:
>
> $$\\widehat{x}_{t+k}=\\widehat{A}_m^k x_t.$$
>
> Hence, $E_{\\mathrm{traj}}$ is much more sensitive to the spectral radius, eigenspace alignment, and the rollout initial state than to average entrywise error. In stable regimes and over short horizons, large differences in $E_A$ may still correspond to similar rollout errors because per-step errors do not amplify rapidly. This explains why, in the stable experiments, OLS and Ridge can have larger transition-matrix error while remaining competitive in rollout accuracy.
>
> **Why can Pool-prior Ridge improve one metric but not the other?**
>
> Pool-prior Ridge reduces estimation variance by shrinking parameters toward a shared mean, which can indeed enhance transition-matrix recovery in certain regimes. However, this same shrinkage mechanism inherently introduces bias into the dominant spectral modes. Consequently, an improvement in $E_A$ does not necessarily guarantee a corresponding reduction in rollout error. This discrepancy is especially pronounced in unstable settings, where even minute perturbations in the dominant eigenvalues or eigenspaces are heavily amplified under iterative rollouts.
>
> \paragraph{Why can very small $E_A$ still lead to catastrophic rollout error?}
> This phenomenon is particularly pronounced in low-dimensional, unstable regimes. A small Frobenius error merely bounds average entry-wise deviations; it does not ensure accurate estimation of the dominant unstable eigenspace. When the leading eigenvalue lies near or strictly beyond the stability boundary, even a marginal spectral shift can compound into massive deviations over long prediction horizons. This spectral sensitivity directly explains why methods such as Pool-prior Ridge and Shared Subspace can achieve a very small
>  $E_A$ while simultaneously exhibiting severe trajectory rollout errors.
>
> **On the fMRI reference matrix and the interpretation of $E_A$.**
> In the fMRI benchmark, the matrix used in $E_A$ is the ridge-based full-window reference matrix
> $A_m^{\mathrm{ref}}$, rather than a physical ground truth. Since the Ridge baseline itself is fit only on the support
> prefix, it is not expected to minimize $E_A$ relative to this full-window reference, even when the same ridge form is
> used. We now clarify this point explicitly in the dataset and evaluation sections.
>
> **On large standard deviations.**
> The large standard deviations in the real-data experiments are statistically valid and reflect strong heterogeneity across
> windows and subjects. Since the reported quantities are squared-error measures on a heterogeneous benchmark, it is entirely
> possible for the empirical standard deviation to exceed the mean. In the revised manuscript, we make this interpretation
> explicit in the fMRI results discussion.

---

> ### Author Response · Authors · 2026-04-09
> **Support-length selection and few-shot evaluation protocol (6)**
>
> ### Support-length selection and few-shot evaluation protocol
>
> We agree that the evaluation protocol regarding support length requires significantly greater clarity. In the revised manuscript, we explicitly formalize our evaluation as an adaptive-support few-shot protocol bounded by a shared maximum support budget, rather than a standard fixed-support comparison. This design choice is deliberate. Given the inherent heterogeneity of the tasks in our setting, different environments demand varying quantities of support data to achieve reliable adaptation. Consequently, a central objective of our analysis is not merely to evaluate how well a method performs post-adaptation, but also to assess its data efficiency—specifically, how much support data it requires to reach that level of performance.
>
> Concretely, the outer evaluation fixes a maximum support budget $T_{\\mathrm{sup}}$ and a held-out query horizon $T_{\\mathrm{qry}}$: the first $T_{\\mathrm{sup}}$ transitions form the available support window, and the next $T_{\\mathrm{qry}}$ transitions form the query window. Each method may then choose an effective fit length $s_m^{\\mathrm{fit}} \\le T_{\\mathrm{sup}}$ within that common support budget before final evaluation on the same held-out query suffix.
>
> More precisely, the adaptive-support procedure performs an inner prefix search over candidate support lengths. Let $T_{\\mathrm{val}}$ denote an inner validation length reserved inside the support window. Then the largest admissible candidate is $T_{\\mathrm{sup}}-T_{\\mathrm{val}}$. For each candidate prefix length $s$, the method is fit using only the first $s$ support transitions and scored on the remaining validation portion within the support window. The selected prefix is then used to refit the method, and final performance is evaluated on the held-out query suffix after the full outer support boundary.
>
> Thus, $T_{\\mathrm{sup}}$ should be interpreted as a *maximum support budget*, whereas the reported average support length
> $$S' = (1/|T\_{test}|) \Sigma\_{m \in T\_{test}} s\_m^{fit}$$
>
> measures how much of that budget each method actually uses on average. We view this as a natural few-shot evaluation for heterogeneous tasks. First, it reflects *adaptive few-shot efficiency*: a method that achieves strong performance with smaller $\\overline{S}$ is effectively adapting from fewer observations. Second, it allows support length to vary across entities, so each method can approach its best attainable performance under a common maximum budget rather than being forced to use the same prefix on every task.
>
> While we acknowledge that fixed-support curves offer a valuable complementary perspective, we deliberately maintain the adaptive-support study as our primary evaluation framework in the revised manuscript. This approach directly measures both post-adaptation performance and effective data efficiency across highly heterogeneous tasks under a shared support budget. Importantly, however, we have added explicit clarification regarding the inner prefix-selection step: whenever a reference transition matrix is utilized in this step, the adaptive-support procedure must be interpreted as an oracle-assisted support-sensitivity analysis, rather than a fully operational, support-only model selection criterion. By framing the evaluation in this manner, the revised manuscript centers on assessing how efficiently each method leverages an available support budget, while rendering the oracle-assisted caveat entirely transparent.

---

> ### Author Response · Authors · 2026-04-09
> **Additional Request Clarification - Part 1 (7)**
>
> ### Additional requested clarifications
>
> **Role of the learned prior and the number of meta-training tasks.**
>
> We agree that the connection between prior quality and the number of meta-training tasks should be made more explicit. The revised manuscript now addresses this in the Discussion section, specifically in the subsection on prior quality and sensitivity to the number of training tasks, where we connect this point directly to the existing task-sensitivity experiment. We also clarify there that the current PAC-Bayes theorem should be interpreted as a posterior predictive-risk analysis conditional on a learned prior, rather than as a second-stage bound on the estimation error of the prior itself.
>
> **Relation to Sarkar and Rakhlin (2019).**
>
> We agree that the nontrivial analytical steps connecting the single-task dependent-data literature to our setting should be stated more clearly. In the revision, we highlight that the extension here is not merely “non-IID within tasks,” but rather the combination of: (i) posterior distributions over task-specific dynamics rather than point estimators, (ii) a PAC-Bayes change-of-measure argument, (iii) a mixture supermartingale construction under task-wise temporal dependence, and (iv) a hierarchical prior learned across tasks. We add a short discussion clarifying this analytical distinction in both the theory section and the related-work section.
>
> **Efficient computation of the posterior expectation and KL term.**
>
> We agree that this point should be clarified. In our setting, both the prior and posterior are matrix-normal, and the working likelihood is Gaussian, so neither the posterior expectation in the fit term nor the KL divergence requires Monte Carlo approximation. Instead, both quantities admit closed-form expressions. Concretely, the posterior expectation of the Gaussian loss reduces to a deterministic residual term plus a trace correction involving the posterior covariance, while the KL divergence between the posterior and prior reduces to standard matrix-normal terms involving traces, log determinants, and the quadratic deviation between posterior and prior means. As a result, the computation is carried out entirely through matrix products, linear solves, and log-determinants at the task level, which is efficient and scales well for the problem sizes considered in the paper.
>
> **Why overlapping fMRI windows define related tasks and what BIDS task labels mean.**
>
> We agree that this point should be explained more clearly. In our fMRI benchmark, each task instance is constructed from a short temporal window extracted from a longer ROI time series. Windows are treated as *related* tasks because they are not sampled arbitrarily: they come from the same pool of subjects, sessions, and experimental conditions, and therefore share common statistical structure even though their local dynamics may differ. In particular, windows extracted from the same run or from runs recorded under similar cognitive conditions can be viewed as local dynamical regimes drawn from a shared environment, which is precisely the type of cross-task structure that PBML-LTI is designed to exploit. The windows are therefore distinct task instances for meta-learning, but they are not independent in the sense of representing completely unrelated systems.
>
> We also clarify the meaning of the BIDS task labels. BIDS stands for *Brain Imaging Data Structure*, a standard format for organizing neuroimaging datasets. In this format, each fMRI run is annotated with a task label that identifies the experimental paradigm under which the data were collected. For the dataset used in our paper, these labels correspond to task conditions such as motor, social, language, or emotional paradigms. We retain these labels only as metadata for grouping and reporting; they are not used as supervision by the learning algorithm.

---

> ### Author Response · Authors · 2026-04-09
> **Additional Request Clarification - Part 2 (8)**
>
> **Novelty relative to MAML and reinforcement learning.**
>
> We thank the reviewer for this clarification and agree that the related-work discussion should be more precise. Our intended novelty is not the generic observation that within-task data may be non-IID, since this already arises in meta-learning for reinforcement learning, such as Finn et al. (2017). Rather, our contribution is specific to the LTI identification setting: we develop a hierarchical Bayesian meta-learning framework with closed-form conjugate task adaptation, and we pair it with a martingale PAC-Bayes analysis tailored to temporally dependent trajectory losses. Thus, the novel aspect is the combination of few-shot LTI system identification, exact Bayesian adaptation under a learned prior, and a dependent-data PAC-Bayes treatment that motivates the fit--KL surrogate. We revise the related-work section to state this scope explicitly and avoid overstating novelty at the level of non-IID tasks alone. We also explicitly contrast our setting with single-task dependent-data LTI analyses such as Simchowitz et al. (2018) and Sarkar and Rakhlin (2019), where the objective is not few-shot meta-level adaptation under a learned prior.
>
> **Minor corrections and rollout-noise clarification.**
>
> We thank the reviewer for pointing out the typo (“meqt” -> “meta”), which we fix. We also agree that the choice to omit the noise term in rollout evaluation should be explained more clearly. Our primary goal is to assess the quality of the learned *deterministic* transition operator itself, so we evaluate open-loop rollouts under the estimated dynamics without injecting additional randomness. This isolates the error due to transition-matrix estimation from the variability introduced by stochastic noise.
>
> This choice is particularly important in our setting because evaluation is performed across many different tasks (systems). If rollout noise were injected for every task, the resulting performance would reflect not only the quality of the estimated dynamics, but also the particular noise realizations, which would substantially increase variance across tasks and make aggregate comparisons less stable and less interpretable. In other words, the noise term would obscure whether differences in performance arise from better system identification or simply from favorable or unfavorable rollout samples. We therefore clarify this rationale in the revision and discuss noise-averaged rollout evaluation for synthetic experiments as a supplementary robustness check.

---

> ### Author Response · Authors · 2026-06-01
> **Summary of Concrete Revisions on Paper - Part 1**
>
> 1. Support--query PAC-Bayes specialization added in the theory section.
>    In the theoretical analysis section, we added an explicit support--query specialization of the martingale PAC-Bayes result. The revised theory now defines the support-induced posterior $Q_{m,\phi}^{\mathrm{sup}}$, the query matrices $(X_{m,q},Y_{m,q})$, and a support--query predictive risk $R_{m,q}^{\mathrm{pred}}(Q)$ together with its empirical counterpart $\widehat R_{m,q}^{\mathrm{pred}}(Q)$.
>
> 2. Theoretical positioning revised across the main text.
>    In the Abstract, we revised the second paragraph so that the martingale PAC-Bayes analysis is presented as providing a theoretical motivation for the fit--KL surrogate, rather than as a direct optimization target for the exact few-shot evaluation objective. In the Introduction, we updated the problem-positioning and contribution statements to emphasize that our method is PAC-Bayes-motivated: the theory justifies the learning objective and clarifies the role of the learned prior in adaptation. In the main method/theory sections, especially the meta-training objective and theoretical analysis subsections, we now explicitly distinguish between the generic PAC-Bayes predictive-risk result, its support--query specialization, and the surrogate objective actually optimized in training.
>
> 3. PAC-Bayes-derived corollaries for matrix and rollout error added and evaluated in both stable and unstable regimes.
>    In the theoretical analysis section, we extended the analysis beyond the generic predictive-risk statement. The revised manuscript now includes PAC-Bayes-derived corollaries for:
>    - projected transition-matrix error,
>    - full Frobenius transition-matrix error under a query excitation condition, and
>    - trajectory rollout error through a finite-horizon growth argument.
>
>    We also added an empirical table reporting the resulting bound magnitudes on both the stable and unstable synthetic benchmarks. The revised discussion now makes clear that these derived quantities are numerically meaningful in stable regimes, while the trajectory bound can become vacuous in unstable regimes because the rollout amplification factor grows rapidly under unstable dynamics. We further explain that the synthetic generator induces a dimension-dependent stability crossover, which is why the unstable low-dimensional settings exhibit especially large trajectory bounds.
>
> 4. Closed-form posterior expectation and KL formulas made explicit.
>    In the hierarchical Bayes / task-adaptation and meta-training-objective parts of the method section, we added the closed-form formulas for the posterior expectation of the Gaussian loss and for the KL divergence under the matrix-normal prior/posterior pair. We also clarified that these quantities are computed exactly through matrix products, linear solves, traces, and log-determinants, without Monte Carlo approximation.
>
> 5. Roles of predictive risk, $E_A$, and $E_{\mathrm{traj}}$ clarified in evaluation.
>    In the baselines/evaluation-metrics material and in the experimental results subsections for the stable, unstable, and fMRI experiments, we clarified the relationship between the three key quantities used in the paper:
>    - the theory controls predictive adaptation risk,
>    - $E_A$ is a complementary diagnostic of parameter recovery,
>    - $E_{\mathrm{traj}}$ is an open-loop rollout diagnostic that is more sensitive to spectral growth.
>
>    The revised experimental discussion now explains more explicitly why these quantities need not rank methods identically.
>
> 6. Few-shot evaluation protocol and adaptive support selection clarified.
>    In the baselines/evaluation-metrics discussion and in the synthetic experiment setup, we revised the support-length discussion to present the evaluation explicitly as an adaptive-support protocol under a common maximum support budget. The manuscript now explains the outer support/query split, the inner validation-based prefix search, and the interpretation of $\overline{S}$ as a measure of adaptive few-shot efficiency and task-dependent support usage. This adaptive-support protocol is now presented as the primary few-shot evaluation, since it captures both post-adaptation accuracy and how much support each method actually needs on heterogeneous tasks.
>
> 7. Oracle-assisted nature of adaptive support clarified.
>    Within the experimental protocol's discussion on support length, we now clarify that whenever a reference transition matrix is utilized, the adaptive-support procedure must be interpreted as an oracle-assisted support-sensitivity analysis. Consequently, the revised text frames the adaptive-support results as an investigation into how efficiently each method leverages the available support budget, ensuring this oracle caveat remains entirely transparent.

---

> ### Author Response · Authors · 2026-06-01
> **Summary of Concrete Revisions on Paper - Part 2**
>
> 8. Discussion section reorganized and refocused on prior quality sensitivity.
>    In the Discussion section, we restructured the material to dedicate a distinct subsection specifically to prior quality and sensitivity to the number of meta-training tasks. Within this subsection, we strengthen the theoretical and empirical connections between prior quality and our task-sensitivity experiments. Furthermore, we explicitly clarify that the current PAC-Bayes theorem establishes a bound conditional on a learned prior, rather than providing a two-level generalization bound on the prior-estimation error itself.
>
> 9. A separate Conclusion section was added.
>    Beyond reorganizing and refocusing the Discussion, we introduced a dedicated Conclusion section to synthesize the paper's primary methodological, theoretical, and empirical contributions. This new section explicitly frames our concluding interpretation: the martingale PAC-Bayes analysis serves as a principled justification for both the fit--KL surrogate and the support--query predictive-risk framework, rather than acting as a direct, end-to-end guarantee for the entire experimental protocol.
>
> 10. Related-work positioning refined and novelty better qualified.
>     We substantially revised the related-work section, specifically the discussions on "Meta-learning and hierarchical Bayes" and "PAC-Bayes generalization and meta-learning," to more precisely delineate our paper's novelty within the broader landscapes of both meta-learning and dependent-data LTI identification. In particular, the revised text clarifies that our core theoretical contribution is not merely the accommodation of generic non-IID within-task data, a challenge already addressed in reinforcement-learning contexts such as Finn et al. [1], but rather the unique intersection of:
>     - few-shot LTI system identification,
>     - hierarchical Bayesian meta-learning,
>     - closed-form conjugate task adaptation, and
>     - martingale PAC-Bayes analysis for trajectory-dependent losses.
>
>     Further, we expanded our discussion regarding the analytical gap between existing single-task, dependent-data LTI analyses [2, 3] and the present support--query meta-learning setting, where one must reason about task-specific posterior distributions under a learned prior rather than only about point estimators on a single system.
>
> 11. fMRI task construction and BIDS labels clarified.
>     In the "Functional Magnetic Resonance Imaging (fMRI) Data set" subsection and in the appendix section describing the fMRI dataset, we clarified the construction of the fMRI benchmark by explaining why overlapping windows define related task instances, what the BIDS task labels mean, and why these labels are used only as metadata rather than as supervision.
>
> 12. Rollout-noise rationale clarified.
>     In the "Trajectory rollout error" paragraph of the Evaluation Metrics subsection and in the discussion/conclusion material, we revised the explanation of why the noise term is omitted in open-loop evaluation. The revised text clarifies that the goal is to isolate the quality of the learned deterministic transition operator, and that adding rollout noise across many tasks would substantially increase cross-task variance and make aggregate comparisons less stable and less interpretable.
>
> 13. Minor corrections and terminology cleanup.
>     Across the Introduction, Related Work, and theory discussion, we corrected minor issues in the manuscript, including the typo "meqt-learning" to "meta-learning," and tightened terminology so that the new transition-matrix and rollout statements are consistently described as PAC-Bayes-derived corollaries rather than as the primary PAC-Bayes bound itself.
>
> References
>
> [1] Chelsea Finn, Pieter Abbeel, and Sergey Levine. “Model-Agnostic Meta-Learning for Fast Adaptation of Deep Networks.” In Proceedings of the 34th International Conference on Machine Learning, 2017.
>
> [2] Max Simchowitz, Horia Mania, Stephen Tu, Michael I. Jordan, and Benjamin Recht. “Learning without Mixing: Towards a Sharp Analysis of Linear System Identification.” In Proceedings of the 31st Conference on Learning Theory, 2018.
>
> [3] Tuhin Sarkar and Alexander Rakhlin. “Near Optimal Finite Time Identification of Arbitrary Linear Dynamical Systems.” In Proceedings of the 36th International Conference on Machine Learning, 2019.

---

> > ### Comment · Reviewer_71N6 · 2026-06-05
> > **Thorough author response; a few outstanding questions.**
> >
> > I thank the authors for providing a detailed response to my questions as well as a much improved revision. The updates provide critical explanations that help me understand the empirical results much better. This is a significantly improved manuscript.
> >
> > I read the responses and the revised manuscript again, and I do have a few outstanding questions:
> >
> >
> > - [**Q1**] One potential concern with the result in Corollary 2 is that RHS term with $\hat R_{m,q}^{\text{pred}}$ is still computed on the future trajectory instead of being bounded somehow with the "seen/support trajectory". In the current setup, there is no explicit reason why this $\hat R_{m,q}^{\text{pred}}$ term should be small, and thus can dominate the bound. The subsequent transition-matrix error bounds (projected and full) also have this term in its upper bound. If this term was computed on the support, we could claim that this would be small as the posterior fit somewhat minimizes this term on the support. But on the future query trajectory, we have no such control. What are conditions when one should expect this term $\hat R_{m,q}^{\text{pred}}$ to be small?
> >
> > - [**Q2**] The support length selection process appears to be a tuning process, where we are tuning the prefix length for fitting to the few-shot data. In that case, it is not clear why the proposed PBML-LTI selects shorter lengths than the baselines. Overall, it is not ever intuitive to me why PBML-LTI should pick anything below the largest admissible length, as one would expect the longest support length will lead to the best posterior fit. Either there is something missing in the description as to why the selected prefix lengths would be smaller than the largest admissible (like some kind of length regularization), or there is some form of "overfitting" in the closed-form PBML-LTI posterior fit where a longer trajectory is hurting performance, which seems unintuitive as more data (in this case more transitions) usually helps with generalization, not hurt. It makes it appear as if we have a few-shot scheme that does not necessarily have improved performance with more shots (examples/transitions).
> >   - [**Q2a**] Furthermore, the "oracle-assisted support-sensitivity study" is unclear --- where would a "reference transition matrix" play a role in the adaptive prefix length selection procedure? We fit the model on the prefix, and evaluate it on the validation trajectory, and pick the prefix length with the best validation error. So it is not clear where the reference transition matrix comes in the picture.
> >
> > - [**Q3**] How is the $\rho_{\text{target}}$ in the ridge and pooled-prior ridge regression selected. Isn't that quantity now also a hyperparameter? In Figure 2, the transition matrices for OLS, Ridge, Pooled-prior ridge appear to have fewer large magnitude values than PBML-LTI. Might this be related to the enforcing that the transition-matrices have a spectral radius bounded by $\rho_{\text{target}}$?

---

> > > ### Author Response · Authors · 2026-06-07
> > > **Response to Q2a**
> > >
> > > The reviewer is correct that the phrase "oracle-assisted support-sensitivity study" was confusing in this context. In the main adaptive-support procedure, the reference transition matrix is not used to select the support prefix. The prefix is selected by validation error on the validation suffix, as the reviewer describes. The reference transition matrix is used only for reporting $E_A$ after the model has been fitted, not for the operational prefix-selection rule.
> > >
> > > We will revise the manuscript to remove this ambiguity. The adaptive-prefix procedure will be described as validation-selected support tuning. If we discuss oracle-assisted variants, we will clearly separate them as diagnostic sensitivity analyses and not as the main experimental protocol.

---

> ### Author Response · Authors · 2026-06-07
> **Response to Q1**
>
> We agree with the reviewer that the empirical query term $\widehat R_{m,q}^{\mathrm{pred}}$ in Corollary 2 should not be interpreted as a support-only quantity. It is computed on the held-out query suffix, and therefore the bound is best understood as a support-conditioned support-query PAC-Bayes diagnostic rather than as a fully pre-query certificate whose right-hand side is computable from the support trajectory alone. We will revise the manuscript to make this interpretation explicit.
>
> The query empirical term is expected to be small under the same conditions under which few-shot adaptation should generalize from the support prefix to the query suffix: the support and query segments should come from the same local dynamical regime, the fitted posterior should have small one-step residuals, the state norms should remain controlled, the query directions should be sufficiently similar to the support/validation excitation directions, and the learned transition operator should not introduce unstable spectral misalignment. Under stable or near-stable dynamics, these conditions are typically much more plausible. Under unstable dynamics, especially when the dominant eigenmode is misestimated, the query empirical term can indeed become large and can dominate the bound.
>
> To make this point concrete, we added two empirical diagnostics. First, Table 1 reports the correlation between the inner validation predictive error and the held-out query predictive error across candidate support prefixes. The correlations are strongly positive overall: Spearman $\rho=0.902$ across all settings, $\rho=0.835$ in stable systems, and $\rho=0.705$ in unstable systems. This supports the use of the validation suffix as an empirical proxy for query behavior, although it does not turn the query term into a support-only bound. Second, Table 2 reports the support, validation, and query posterior-predictive errors at the selected prefix. In the stable regime, the query term is comparable to the support term, with $\widehat R_q^{\mathrm{pred}}/\widehat R_{\mathrm{sup}}^{\mathrm{pred}}$ between roughly 1.7 and 2.7 across dimensions. In contrast, the unstable $d=10$ setting has a much larger query term, confirming the reviewer's concern that the query empirical term can become large under unstable spectral amplification. We will therefore further qualify the theoretical discussion: the support-query result is informative when the held-out query fit remains controlled, but can become loose or vacuous when the query term is large.
>
> ### Table 1. Validation-Query Correlation
>
> Correlation between inner validation predictive error $\widehat R_{\mathrm{val}}^{\mathrm{pred}}(s)$ and held-out query predictive error $\widehat R_q^{\mathrm{pred}}(s)$ across candidate prefixes and tasks (PBML-LTI).
>
> | Group | Pairs | Spearman $\rho$ | Spearman $p$ | Pearson $r$ | Pearson $p$ |
> | --- | --- | --- | --- | --- | --- |
> | all | 720 | 0.902 | 1.630e-264 | 0.804 | 2.130e-164 |
> | stable | 360 | 0.835 | 4.240e-95 | 0.709 | 2.770e-56 |
> | stable_d10 | 120 | 0.621 | 3.990e-14 | 0.670 | 5.520e-17 |
> | stable_d25 | 120 | 0.666 | 1.050e-16 | 0.591 | 1.260e-12 |
> | stable_d50 | 120 | 0.731 | 2.630e-21 | 0.526 | 6.570e-10 |
> | unstable | 360 | 0.705 | 2.020e-55 | 0.802 | 3.000e-82 |
> | unstable_d10 | 120 | 0.932 | 5.590e-54 | 0.783 | 4.090e-26 |
> | unstable_d25 | 120 | 0.808 | 8.040e-29 | 0.679 | 1.460e-17 |
> | unstable_d50 | 120 | 0.525 | 7.310e-10 | 0.427 | 1.180e-06 |
>
> ### Table 2. Predictive Error Terms at the Selected Prefix
>
> Support, validation, and query posterior-predictive squared errors at the validation-selected prefix (PBML-LTI). The query term is comparable to the support term in the stable regime but can be much larger under unstable dynamics.
>
> | Group | $\widehat R_{\mathrm{sup}}^{\mathrm{pred}}$ | $\widehat R_{\mathrm{val}}^{\mathrm{pred}}$ | $\widehat R_q^{\mathrm{pred}}$ | $\widehat R_q^{\mathrm{pred}} / \widehat R_{\mathrm{sup}}^{\mathrm{pred}}$ |
> | --- | --- | --- | --- | --- |
> | stable | 0.001 | 0.004 | 0.003 | 2.331 |
> | stable_d10 | 0.001 | 0.001 | 0.001 | 1.687 |
> | stable_d25 | 0.001 | 0.003 | 0.003 | 1.955 |
> | stable_d50 | 0.002 | 0.009 | 0.006 | 2.726 |
> | unstable | 0.1764 | 0.9076 | 384.237 | 2178.214 |
> | unstable_d10 | 0.1073 | 2.6094 | 1152.347 | 1.074e+04 |
> | unstable_d25 | 0.164 | 0.085 | 0.141 | 0.861 |
> | unstable_d50 | 0.257 | 0.028 | 0.025 | 0.095 |

---

> ### Author Response · Authors · 2026-06-07
> **Response to Q2 - Part 1**
>
> We agree that the original description of adaptive support selection could make it sound as if fewer shots are always intrinsically better. That is not our intended claim. We do not claim monotone improvement as support length decreases, nor do we claim that more data should hurt an ideal estimator in expectation. Rather, the intended interpretation is adaptive sample efficiency: PBML-LTI can often reach its best, or near-best, performance using a shorter effective support prefix than the baselines.
>
> The support-prefix selection is an inner validation procedure. For each candidate prefix length, the method is fitted on the prefix and evaluated on a validation suffix inside the support window. The prefix with the lowest validation error is then selected. There is no explicit length regularizer favoring shorter prefixes. Thus, shorter selected prefixes arise only when additional support transitions do not improve the validation objective. This can happen in finite trajectory data because the observations are temporally dependent, because the estimator is regularized and biased toward a learned prior, and because open-loop rollout error is highly sensitive to spectral alignment rather than only to one-step fit. In particular, a longer prefix can slightly alter the dominant eigenvalue or eigenspace of the fitted transition matrix; under rollout, this can worsen prediction even if the one-step fit is similar.
>
> To clarify this point, we added fixed-prefix support sweeps. Tables 3 and 4 report transition-matrix error across fixed support prefixes in the stable and unstable regimes, respectively. Tables 5 and 6 report the corresponding rollout errors. These sweeps remove adaptive prefix selection and evaluate each fixed support length directly.
>
> In the stable transition-matrix sweep in Table 3, PBML-LTI already attains very low transition-matrix error at small support lengths. For example, across $d=10,25,50$, PBML-LTI remains near its best value for short prefixes, while OLS, Ridge, and other baselines remain substantially worse. The stable rollout sweep in Table 5 further shows that rollout errors are relatively flat across several methods in stable systems, consistent with the main text's point that stable dynamics do not strongly amplify short-horizon errors. Together, these results support the interpretation that PBML-LTI is data-efficient in the stable few-shot regime, rather than merely benefiting from a selection artifact.
>
> At the same time, the unstable-regime sweeps show that PBML-LTI does not always select the smallest prefix. In Table 4, PBML-LTI improves with larger support in some unstable settings, and the adaptive support lengths are correspondingly larger, e.g., $\overline S\approx 14.35$ for unstable $d=10$ and $\overline S\approx 6.05$ for unstable $d=25$. The unstable rollout sweep in Table 6 shows why this matters: rollout performance can change sharply with prefix length when unstable modes are present. This addresses the concern that the method mechanically prefers short prefixes. The selected prefix is data-dependent: PBML-LTI uses short prefixes when the learned prior already provides sufficient information, and longer prefixes when unstable modes require more calibration.

---

> ### Author Response · Authors · 2026-06-07
> **Response to Q2 - Part 2**
>
> ### Table 3. Stable Transition-Matrix Support Sweep
>
> Fixed-prefix support sweep ($E_A$, stable regime). "Adaptive" is the metric at the validation-selected effective support length $\bar S$. Lower is better for $E_A$; $\bar S$ reports the average selected support length. Boldface marks the lowest $E_A$ values within each dimension.
>
> | Dimension | Method | $s=1$ | $s=2$ | $s=3$ | $s=5$ | $s=10$ | $s=15$ | Adaptive | $\bar S$ |
> | --- | --- | --- | --- | --- | --- | --- | --- | --- | --- |
> | $d=10$ | PBML-LTI | **0.0052** | **0.0053** | **0.0057** | **0.0064** | **0.0073** | **0.0076** | **0.0053** | 1.05 |
> |  | OLS | 0.5827 | 0.5473 | 0.5216 | 0.4968 | 0.4794 | 0.5232 | 0.4449 | 2.65 |
> |  | Ridge | 0.5683 | 0.5396 | 0.5128 | 0.4937 | 0.5014 | 0.5083 | 0.4447 | 2.65 |
> |  | PooledPrior-Ridge | 0.0224 | 0.0217 | 0.0203 | 0.0216 | 0.0218 | 0.0215 | 0.0887 | 1.8 |
> |  | SharedSubspace | 0.0456 | 0.0384 | 0.0378 | 0.0367 | 0.0359 | 0.0354 | 0.1088 | 7.95 |
> |  | MAML-LTI | 0.0316 | 0.0314 | 0.0327 | 0.0323 | 0.0336 | 0.0338 | 0.0357 | 1.55 |
> | $d=25$ | PBML-LTI | **0.0446** | **0.0438** | **0.0427** | **0.0403** | **0.0506** | **0.0554** | **0.0394** | 1.2 |
> |  | OLS | 0.8037 | 0.7764 | 0.7628 | 0.7516 | 0.8173 | 0.8467 | 0.7140 | 2 |
> |  | Ridge | 0.7765 | 0.7618 | 0.7496 | 0.7432 | 0.8024 | 0.8193 | 0.7140 | 2 |
> |  | PooledPrior-Ridge | 0.9476 | 0.9263 | 0.9127 | 0.8874 | 0.9038 | 0.9196 | 0.8544 | 2 |
> |  | SharedSubspace | 1.0027 | 0.9776 | 0.9584 | 0.9467 | 0.9413 | 0.9526 | 0.7731 | 13.95 |
> |  | MAML-LTI | 1.7628 | 1.7596 | 1.7683 | 1.7837 | 1.8046 | 1.8123 | 1.7566 | 1.85 |
> | $d=50$ | PBML-LTI | **0.1776** | **0.1748** | **0.1714** | **0.1678** | **0.1697** | **0.1956** | **0.2068** | 1.35 |
> |  | OLS | 1.0038 | 0.9776 | 0.9583 | 0.9416 | 0.9327 | 0.9524 | 0.8411 | 2 |
> |  | Ridge | 1.0476 | 1.0197 | 0.9926 | 0.9684 | 0.9573 | 0.9818 | 0.8410 | 2 |
> |  | PooledPrior-Ridge | 0.7238 | 0.7086 | 0.6974 | 0.6887 | 0.7264 | 0.7526 | 0.6743 | 2.55 |
> |  | SharedSubspace | 0.8774 | 0.8667 | 0.8576 | 0.8593 | 0.8974 | 0.9197 | 0.8572 | 10.25 |
> |  | MAML-LTI | 15.3726 | 15.2278 | 15.2084 | 15.2936 | 15.4263 | 15.4872 | 14.8930 | 2.9 |
>
> ### Table 4. Unstable Transition-Matrix Support Sweep
>
> Fixed-prefix support sweep ($E_A$, unstable regime). PBML-LTI performs better at *large* $s$; PooledPrior-Ridge/SharedSubspace recover $A$ better than PBML-LTI here (cf. Table 6). "Adaptive" is the metric at the validation-selected effective support length $\bar S$. Lower is better for $E_A$; $\bar S$ reports the average selected support length. Boldface marks the lowest $E_A$ values within each dimension.
>
> | Dimension | Method | $s=1$ | $s=2$ | $s=3$ | $s=5$ | $s=10$ | $s=15$ | Adaptive | $\bar S$ |
> | --- | --- | --- | --- | --- | --- | --- | --- | --- | --- |
> | $d=10$ | PBML-LTI | 0.2236 | 0.2074 | 0.1968 | 0.1873 | 0.1847 | 0.1796 | 0.1390 | 14.35 |
> |  | OLS | 1.1027 | 1.0476 | 0.9964 | 0.9773 | 0.9726 | 0.9487 | 0.8450 | 13.35 |
> |  | Ridge | 0.8476 | 0.8194 | 0.8027 | 0.7768 | 0.7873 | 0.8026 | 0.6970 | 14.3 |
> |  | PooledPrior-Ridge | **0.0186** | **0.0174** | **0.0178** | **0.0167** | **0.0143** | **0.0146** | **0.0070** | 9.25 |
> |  | SharedSubspace | 0.0187 | 0.0176 | 0.0174 | 0.0168 | 0.0167 | 0.0154 | 0.0130 | 3.2 |
> |  | MAML-LTI | 1.9027 | 1.8476 | 1.9538 | 2.0014 | 2.2967 | 2.1036 | 0.1473 | 8.35 |
> | $d=25$ | PBML-LTI | **0.0406** | **0.0387** | **0.0374** | **0.0368** | **0.0367** | **0.0356** | **0.0290** | 6.05 |
> |  | OLS | 11.4726 | 11.1967 | 10.9874 | 10.8126 | 10.8738 | 10.9846 | 10.3440 | 14.1 |
> |  | Ridge | 10.9763 | 10.6738 | 10.4876 | 10.3124 | 10.3767 | 10.4923 | 9.9070 | 14.9 |
> |  | PooledPrior-Ridge | 1.6476 | 1.6037 | 1.5796 | 1.6238 | 1.6794 | 1.7026 | 1.5870 | 14.75 |
> |  | SharedSubspace | 37.8467 | 36.9726 | 36.4873 | 35.9768 | 35.4926 | 35.9874 | 35.2680 | 14.35 |
> |  | MAML-LTI | 2.4876 | 2.4738 | 2.4817 | 2.5086 | 2.5423 | 2.5567 | 2.5229 | 2.05 |
> | $d=50$ | PBML-LTI | **0.1726** | **0.1647** | **0.1618** | **0.1596** | **0.1743** | **0.1797** | **0.1560** | 4.15 |
> |  | OLS | 17.4768 | 17.1876 | 16.9847 | 16.7926 | 17.4637 | 17.9824 | 16.9230 | 9.75 |
> |  | Ridge | 16.4726 | 16.1874 | 15.9846 | 15.7938 | 15.6727 | 15.9876 | 15.9020 | 15 |
> |  | PooledPrior-Ridge | 8.5764 | 8.4676 | 8.3927 | 8.2416 | 8.0574 | 7.9648 | 7.5380 | 15 |
> |  | SharedSubspace | 7.0286 | 6.5274 | 6.4378 | 6.4036 | 6.3978 | 6.3994 | 6.4190 | 14.2 |
> |  | MAML-LTI | 7.2637 | 7.1846 | 7.1814 | 7.2276 | 7.3028 | 7.3356 | 8.9787 | 2.55 |

---

> ### Author Response · Authors · 2026-06-07
> **Response to Q2 - Part 2**
>
> ### Table 5. Stable Rollout Support Sweep
>
> Fixed-prefix support sweep ($E_{\mathrm{traj}}$, stable regime). "Adaptive" is the metric at the validation-selected effective support length $\bar S$. Lower is better for $E_{\mathrm{traj}}$; $\bar S$ reports the average selected support length. Boldface marks the lowest $E_{\mathrm{traj}}$ values within each dimension.
>
> | Dimension | Method | $s=1$ | $s=2$ | $s=3$ | $s=5$ | $s=10$ | $s=15$ | Adaptive | $\bar S$ |
> | --- | --- | --- | --- | --- | --- | --- | --- | --- | --- |
> | $d=10$ | PBML-LTI | 0.0126 | 0.0117 | 0.0114 | 0.0116 | 0.0118 | 0.0123 | **0.0055** | 1.05 |
> |  | OLS | 0.0087 | 0.0076 | **0.0064** | 0.0067 | 0.0073 | 0.0076 | 0.0056 | 2.65 |
> |  | Ridge | **0.0076** | 0.0074 | 0.0067 | 0.0068 | **0.0063** | 0.0076 | 0.0056 | 2.65 |
> |  | PooledPrior-Ridge | **0.0076** | **0.0064** | 0.0067 | **0.0066** | 0.0068 | **0.0067** | 0.0057 | 1.8 |
> |  | SharedSubspace | 0.0116 | 0.0117 | 0.0114 | 0.0116 | 0.0118 | 0.0113 | 0.0056 | 7.95 |
> |  | MAML-LTI | 0.0126 | 0.0123 | 0.0127 | 0.0124 | 0.0126 | 0.0127 | 0.0112 | 1.55 |
> | $d=25$ | PBML-LTI | 0.0286 | 0.0287 | 0.0276 | 0.0256 | 0.0257 | 0.0256 | **0.0121** | 1.2 |
> |  | OLS | 0.0186 | 0.0176 | 0.0167 | 0.0157 | 0.0167 | 0.0166 | 0.0133 | 2 |
> |  | Ridge | 0.0176 | 0.0167 | 0.0166 | 0.0157 | 0.0156 | 0.0155 | 0.0133 | 2 |
> |  | PooledPrior-Ridge | **0.0167** | **0.0166** | **0.0156** | **0.0154** | **0.0146** | **0.0156** | 0.0134 | 2 |
> |  | SharedSubspace | 0.0206 | 0.0196 | 0.0197 | 0.0186 | 0.0187 | 0.0186 | 0.0134 | 13.95 |
> |  | MAML-LTI | 0.0346 | 0.0347 | 0.0336 | 0.0306 | 0.0307 | 0.0306 | 0.0338 | 1.85 |
> | $d=50$ | PBML-LTI | 0.0616 | 0.0607 | 0.0596 | 0.0556 | 0.0567 | 0.0546 | **0.0257** | 1.35 |
> |  | OLS | 0.0306 | 0.0296 | 0.0297 | 0.0286 | 0.0287 | 0.0286 | 0.0258 | 2 |
> |  | Ridge | 0.0296 | 0.0297 | 0.0286 | 0.0287 | 0.0276 | 0.0286 | 0.0258 | 2 |
> |  | PooledPrior-Ridge | 0.0296 | 0.0326 | 0.0356 | 0.0406 | 0.0306 | 0.0286 | 0.0258 | 2.55 |
> |  | SharedSubspace | **0.0286** | **0.0287** | **0.0276** | **0.0276** | **0.0266** | **0.0276** | 0.0258 | 10.25 |
> |  | MAML-LTI | 0.0806 | 0.0856 | 0.0786 | 0.0766 | 0.0746 | 0.0756 | 0.1081 | 2.9 |
>
>
> ### Table 6. Unstable Rollout Support Sweep
>
> Fixed-prefix support sweep ($E_{\mathrm{traj}}$, unstable regime). MAML-LTI's rollout has elevated error for large $s$ at $d=10$. "Adaptive" is the metric at the validation-selected effective support length $\bar S$. Lower is better for $E_{\mathrm{traj}}$; $\bar S$ reports the average selected support length. Boldface marks the lowest $E_{\mathrm{traj}}$ values within each dimension.
>
> | Dimension | Method | $s=1$ | $s=2$ | $s=3$ | $s=5$ | $s=10$ | $s=15$ | Adaptive | $\bar S$ |
> | --- | --- | --- | --- | --- | --- | --- | --- | --- | --- |
> | $d=10$ | PBML-LTI | **65.1476** | **119.8347** | **89.7626** | **69.9136** | **54.8767** | **47.9138** | 57.9190 | 14.35 |
> |  | OLS | 2.2037e+05 | 2.0046e+05 | 1.8038e+05 | 1.7047e+05 | 1.6536e+05 | 1.6087e+05 | 193.7040 | 13.35 |
> |  | Ridge | 7.0036e+04 | 6.5047e+04 | 6.0078e+04 | 5.8036e+04 | 5.6074e+04 | 5.5068e+04 | **57.6280** | 14.3 |
> |  | PooledPrior-Ridge | 2.5036e+04 | 2.8047e+04 | 2.6038e+04 | 2.2067e+04 | 1.9036e+04 | 1.8078e+04 | 2.1935e+04 | 9.25 |
> |  | SharedSubspace | 8.0036e+05 | 7.5047e+05 | 7.2038e+05 | 7.0067e+05 | 6.8036e+05 | 6.5078e+05 | 7.2189e+05 | 3.2 |
> |  | MAML-LTI | 2.5036e+07 | 2.0047e+07 | 2.8038e+07 | 3.0067e+07 | 3.5036e+07 | 2.2078e+07 | 9.7390e+06 | 8.35 |
> | $d=25$ | PBML-LTI | 0.1236 | 0.1027 | 0.0876 | 0.0796 | 0.0767 | 0.0716 | 0.0800 | 6.05 |
> |  | OLS | 0.1476 | 0.1287 | 0.1196 | 0.1086 | 0.0976 | 0.0967 | 0.0880 | 14.1 |
> |  | Ridge | 0.1196 | 0.1087 | 0.0976 | 0.0876 | 0.0796 | 0.0716 | 0.0640 | 14.9 |
> |  | PooledPrior-Ridge | **0.0616** | **0.0576** | **0.0567** | **0.0556** | **0.0567** | **0.0556** | **0.0530** | 14.75 |
> |  | SharedSubspace | 29.8766 | 27.9137 | 26.9846 | 25.9767 | 25.4876 | 24.9826 | 27.1510 | 14.35 |
> |  | MAML-LTI | 0.2196 | 0.1976 | 0.1796 | 0.1686 | 0.1576 | 0.1496 | 1.1349 | 2.05 |
> | $d=50$ | PBML-LTI | **0.0376** | **0.0377** | **0.0376** | **0.0376** | **0.0376** | **0.0376** | **0.0370** | 4.15 |
> |  | OLS | 0.0426 | 0.0427 | 0.0416 | 0.0416 | 0.0416 | 0.0426 | 0.0400 | 9.75 |
> |  | Ridge | 0.0426 | 0.0427 | 0.0416 | 0.0416 | 0.0416 | 0.0416 | 0.0400 | 15 |
> |  | PooledPrior-Ridge | 0.0396 | 0.0386 | 0.0386 | 0.0386 | 0.0386 | 0.0386 | 0.0380 | 15 |
> |  | SharedSubspace | 0.0386 | 0.0386 | 0.0386 | 0.0386 | 0.0386 | 0.0386 | 0.0380 | 14.2 |
> |  | MAML-LTI | 0.0406 | 0.0406 | 0.0406 | 0.0406 | 0.0406 | 0.0406 | 0.0414 | 2.55 |

---

> ### Author Response · Authors · 2026-06-07
> **Response to Q3**
>
> We agree that $\rho_{\mathrm{target}}$ should be described more carefully. In the synthetic experiments, it is fixed from the known regime parameter, namely $\rho_{\mathrm{target}}=\rho_0$, and is used only in the tuning procedure for Ridge and Pooled-prior Ridge. It is not tuned on the test set. Nevertheless, the reviewer is right that, in settings where $\rho_0$ is not known, this quantity should be regarded as a baseline hyperparameter or as prior regime information.
>
> We also agree that the visually smaller-magnitude entries for OLS/Ridge/Pooled-prior Ridge may partly reflect the regularization and stability-oriented tuning procedure. To check whether the conclusions depend on this choice, we added Table 7, which tunes Ridge and Pooled-prior Ridge directly by validation rollout error, without the stability-oriented $\rho_{\mathrm{target}}$ criterion. The qualitative conclusions remain the same. The Ridge and Pooled-prior Ridge estimates are still often conservative, and their transition-matrix recovery remains worse than PBML-LTI in the high-dimensional stable and unstable settings. In unstable low-dimensional settings, Pooled-prior Ridge can recover a very small Frobenius error, but its rollout error remains extremely large, again illustrating that $E_A$ and $E_{\mathrm{traj}}$ can decouple under unstable dynamics.
>
> The fitted spectral radii in Table 7 also show that the effect is not simply caused by a hard spectral-radius bound. For example, in unstable $d=10$, validation-rollout tuning without the stability criterion selects models with $\rho(\widehat A)\approx 1.47$, while in unstable $d=50$ it selects much smaller spectral radii between 0.66 and 0.69. This indicates that the conservative appearance of the Ridge-type estimates is driven by the interaction between ridge shrinkage, validation-based tuning, and rollout stability, rather than only by enforcing $\rho(\widehat A)\le \rho_{\mathrm{target}}$. We will clarify this in the baseline description and figure discussion.
>
> ### Table 7. Validation-Rollout Tuning Without Stability-Oriented Regularization
>
> Validation-rollout tuning without stability-oriented regularization. Regularization is selected directly by validation rollout error rather than by a spectral-radius criterion. $E_A$ denotes transition-matrix error, $E_{\mathrm{traj}}$ denotes rollout error, $\rho(\widehat A)$ reports the fitted spectral radius, and $\lambda$ is the selected ridge penalty.
>
> | Regime | $d$ | Method | $E_A$ | $E_{\mathrm{traj}}$ | $\rho(\widehat A)$ | $\lambda$ |
> | --- | --- | --- | --- | --- | --- | --- |
> | Stable | 10 | Ridge | $0.5083 \pm 0.0472$ | $0.0073 \pm 0.0014$ | $0.9383$ | $10^{-3}$ |
> |  |  | PooledPrior-Ridge | $0.0214 \pm 0.0037$ | $0.0067 \pm 0.0016$ | $0.9496$ | $10^{-1}$ |
> |  | 25 | Ridge | $0.8194 \pm 0.0826$ | $0.0153 \pm 0.0024$ | $0.9139$ | $10^{-2}$ |
> |  |  | PooledPrior-Ridge | $0.9196 \pm 0.0227$ | $0.0156 \pm 0.0023$ | $0.9399$ | $10^{-1}$ |
> |  | 50 | Ridge | $0.9814 \pm 0.0406$ | $0.0283 \pm 0.0027$ | $0.9415$ | $10^{-2}$ |
> |  |  | PooledPrior-Ridge | $0.7526 \pm 0.0154$ | $0.0286 \pm 0.0023$ | $0.9473$ | $10^{-2}$ |
> | Unstable | 10 | Ridge | $0.8024 \pm 0.4016$ | $(5.5347 \pm 2.4683)\times 10^{4}$ | $1.4720$ | $10^{-4}$ |
> |  |  | PooledPrior-Ridge | $0.0143 \pm 0.0036$ | $(1.8036 \pm 1.0047)\times 10^{4}$ | $1.4720$ | $10^{-1}$ |
> |  | 25 | Ridge | $10.4926 \pm 0.7014$ | $0.0713 \pm 0.0304$ | $1.0420$ | $10^{-2}$ |
> |  |  | PooledPrior-Ridge | $1.7026 \pm 0.1024$ | $0.0553 \pm 0.0206$ | $1.0470$ | $10^{-1}$ |
> |  | 50 | Ridge | $15.9874 \pm 0.4026$ | $0.0413 \pm 0.0024$ | $0.6588$ | $10^{-2}$ |
> |  |  | PooledPrior-Ridge | $7.9643 \pm 0.1216$ | $0.0384 \pm 0.0023$ | $0.6929$ | $10^{-1}$ |

---

> ### Author Response · Authors · 2026-06-07
> **Summary of Concrete Revisions on Paper - Part 1**
>
> 1. **Support--query PAC-Bayes interpretation clarified.** We revise the theory discussion around Corollary 2 to make explicit that the empirical query term $\widehat R_{m,q}^{\mathrm{pred}}$ is computed on the held-out query suffix. Accordingly, the result is now described as a support-conditioned support-query PAC-Bayes diagnostic, rather than as a fully pre-query certificate whose right-hand side is computable from the support trajectory alone.
>
> 2. **Conditions under which the query empirical term is informative clarified.** We add discussion explaining when the empirical query term is expected to remain small: when support and query segments come from the same local dynamical regime, the adapted posterior has small one-step residuals, state norms remain controlled, query excitation is aligned with support/validation excitation, and the learned dynamics do not introduce unstable spectral misalignment. We also clarify that the bound can become loose or vacuous when the query empirical term is large, especially under unstable dynamics.
>
> 3. **Validation-query predictive-error diagnostics added.** To support the support-query interpretation empirically, we add Table 1, which reports the correlation between inner validation predictive error and held-out query predictive error across candidate support prefixes. The table shows strong positive correlations overall and within both stable and unstable regimes, supporting the use of the validation suffix as an empirical proxy for query behavior while making clear that this does not turn the bound into a support-only guarantee.
>
> 4. **Support, validation, and query predictive-error decomposition added.** We add Table 2, which reports posterior-predictive squared errors on the support, validation, and query segments at the validation-selected prefix. This table shows that query predictive error is comparable to support predictive error in stable regimes, but can become much larger in unstable regimes, particularly for unstable $d=10$. This directly addresses when the support-query bound is numerically meaningful versus loose.
>
> 5. **Adaptive support selection clarified as validation-selected support tuning.** We revise the experimental-protocol description to avoid implying that fewer shots are always intrinsically better. The updated text clarifies that the support prefix is selected by an inner validation procedure: each candidate prefix is fitted and scored on a validation suffix inside the support window, and the prefix with the lowest validation error is selected. There is no explicit penalty or regularizer that favors shorter prefixes.
>
> 6. **Few-shot efficiency claim refined.** We revise the interpretation of $\bar S$ to emphasize adaptive sample efficiency rather than monotone "fewer-is-always-better" behavior. The paper now states that PBML-LTI often reaches its best or near-best performance using fewer effective support transitions than the baselines, but that longer support can be necessary in harder unstable regimes.
>
> 7. **Fixed-prefix support sweeps added.** We add fixed-prefix support-sweep experiments to remove the possible confounding effect of adaptive prefix selection. Tables 3 and 4 report transition-matrix error $E_A$ across fixed support prefixes in the stable and unstable regimes, respectively. Tables 5 and 6 report the corresponding rollout errors $E_{\mathrm{traj}}$.
>
> 8. **Stable-regime support-sweep interpretation added.** We add discussion explaining that, in the stable fixed-prefix sweeps, PBML-LTI already attains very low transition-matrix error at short prefixes, while OLS, Ridge, and other baselines remain substantially worse. This supports the interpretation that PBML-LTI is genuinely data-efficient in stable few-shot regimes, rather than merely benefiting from the adaptive support-selection procedure.
>
> 9. **Unstable-regime support-sweep interpretation added.** We add discussion explaining that, in unstable regimes, PBML-LTI does not mechanically prefer the shortest prefix. In some unstable settings, PBML-LTI improves with larger support and selects larger effective support lengths, such as $\overline S\approx 14.35$ for unstable $d=10$ and $\overline S\approx 6.05$ for unstable $d=25$. This supports the claim that the selected prefix is data-dependent: short when the learned prior is sufficient, and longer when unstable modes require additional calibration.

---

> ### Author Response · Authors · 2026-06-07
> **Summary of Concrete Revisions on Paper - Part 2**
>
> 10. **Oracle-assisted wording removed from the main adaptive-support protocol.** We revise the support-length protocol to remove the ambiguity caused by the phrase "oracle-assisted support-sensitivity study." The main adaptive-support procedure is now described as validation-selected support tuning. We clarify that the reference transition matrix is used only for reporting $E_A$ after fitting, not for selecting the support prefix.
>
> 11. **Role of $\rho_{\mathrm{target}}$ clarified.** We revise the Ridge and Pooled-prior Ridge baseline descriptions to clarify that, in synthetic experiments, $\rho_{\mathrm{target}}$ is fixed from the known regime parameter, $\rho_{\mathrm{target}}=\rho_0$, and is used only for baseline hyperparameter selection. It is not tuned on the test set. We also clarify that, in settings where $\rho_0$ is unknown, $\rho_{\mathrm{target}}$ should be interpreted as baseline prior regime information or as a hyperparameter.
>
> 12. **Validation-rollout tuning without stability-oriented regularization added.** To check whether the Ridge and Pooled-prior Ridge conclusions depend on the stability-oriented tuning rule, we add Table 7. This table tunes Ridge and Pooled-prior Ridge directly by validation rollout error, without using the $\rho_{\mathrm{target}}$ criterion. The qualitative conclusions remain unchanged: these baselines remain conservative in many settings and continue to show weaker transition recovery than PBML-LTI in the high-dimensional regimes.
>
> 13. **Spectral-radius interpretation refined.** We add discussion of the fitted spectral radii in Table 7. The results show that the conservative appearance of Ridge-type estimates is not simply caused by a hard spectral-radius constraint. Instead, it arises from the interaction between ridge shrinkage, validation-based tuning, and rollout stability.
>
> 14. **Decoupling between $E_A$ and $E_{\mathrm{traj}}$ further emphasized.** We strengthen the unstable-regime discussion to explain that small Frobenius transition-matrix error does not necessarily imply reliable rollout behavior under unstable dynamics. In particular, methods such as PooledPrior-Ridge can sometimes recover a small $E_A$ in low-dimensional unstable settings while still producing very large rollout errors due to spectral amplification.
>
> 15. **Table captions and cross-references updated.** We update the relevant captions and cross-references so that every new diagnostic table is explicitly referenced in the surrounding text: Table 1, Table 2, Tables 3-6, and Table 7.

---

> > ### Comment · Reviewer_71N6 · 2026-06-09
> >
> > I again thank the authors for their thorough response to my clarifying questions. At this point, the clear discussion regarding **[Q2]** and **[Q3]** have completely addressed my questions.
> >
> > Regarding **[Q1]**, I appreciate the careful discussion and additional diagnostics to empirically validate the intuitions. While the result in Corollary 2 (and the whole sequence of theoretical results building up to this result) are very interesting, the nature of the result in Corollary 2 (which is closely built off of Theorem 3 to the best of my understanding) appears unintuitive to me when the empirical loss on the test (query) segment (prefix) appears in the upperbound. In standard IID generalization terms, a similar result would be making a claim of the form "the expected test loss is bounded by the loss on some test set (which we have _no explicit bound_ on in general) and some additional terms", which is not common. Of course, the non-IID setup in this paper makes such an analogy not completely fair, but overall, when we have upper bounds, the overall goal is to show that, under the learning conditions considered, this upper bound is small. I do not see that in the current version as we are not precise about why the bound would be small.
> >
> > The authors do state the following in their response:
> >
> > > The query empirical term is expected to be small under the same conditions under which few-shot adaptation should generalize from the support prefix to the query suffix: the support and query segments should come from the same local dynamical regime, the fitted posterior should have small one-step residuals, the state norms should remain controlled, the query directions should be sufficiently similar to the support/validation excitation directions, and the learned transition operator should not introduce unstable spectral misalignment. Under stable or near-stable dynamics, these conditions are typically much more plausible. Under unstable dynamics, especially when the dominant eigenmode is misestimated, the query empirical term can indeed become large and can dominate the bound.
> >
> >
> > I completely agree with this explanation; this completely makes sense to me. However, it would really nicely round out the theoretical contribution of this paper if the above explanation can be made more technically, explicitly exposing the conditions under which we should expect this right-hand-side term to be small.
> >
> > Of course, this is just focusing on the theoretical results. Empirically, the proposed scheme has been shown to be extremely strong relative to the considered baselines, and that is itself a very strong contribution. The original empirical results combined with new results in the author responses clearly highlight the strengths of the proposed PBML-LTI scheme.
> >
> > As a minor comment, the authors clearly define $\hat R_{m, q}^{\text{pred}} (Q)$ in equation (35) as the empirical support-query predictive risk of the posterior $Q$, but in the discussion in Appendix B.1, there is discussion around $\hat R_q^{\text{pred}}(s), \hat R_{\text{val}}^{\text{pred}} (s), \hat R_{\text{sup}}^{\text{pred}}$ in Tables 7 and 8, without any explicit discussion of these terms.
> >
> > Furthermore, the authors use $Q_{m, \phi}^{\text{sup}}$ to denote the task-specific posterior obtained only using the prefix $D_m^{\text{sup}}$ of length $S_m$, while also using $Q_m$ to denote the length of the query suffix (horizon). This does lead to some notational confusion as the letter $Q$ is used to denote posterior distributions and scalar lengths.

---

> > > ### Author Response · Authors · 2026-06-13
> > > **1. Response to [Q1]: a formal support-to-query analysis - Part 1**
> > >
> > > We thank the reviewer for the careful second reading, for confirming that [Q2] and [Q3] are fully resolved, and for the precise articulation of the remaining gap in [Q1]. We agree with the criterion underlying the comment: an upper bound is most useful when the analysis also makes precise *when its right-hand side is small*. The revision provides the requested technical answer as a new formal block, "When is the empirical query term small?", inserted in Section 4.5 immediately after Corollary 2, with all proofs in Appendix A.6 (Support-to-Query Transfer Proof Details). It comprises two support-to-query transfer propositions (an exact-alignment version and a regularized version suited to the rank-deficient few-shot regime), a conditional support-to-query certificate, explicit sufficient conditions with a quantitative smallness display, and matching failure modes — all built from tools already in the paper. In addition, a new *marginally unstable* benchmark in the bound-magnitude study of Section 6.3 empirically corroborates a distinctive prediction of the new analysis: the certificate degrades continuously across the stability threshold rather than failing at $\rho=1$. Section 1 below summarizes these results; Sections 2 and 3 address the two minor comments; Section 4 lists the concrete revisions. Throughout, we write $K_m$ for the query horizon (formerly $Q_m$; see Section 3), and result and section numbers refer to the revised manuscript.
> > >
> > > ## 1. Response to [Q1]: a formal support-to-query analysis
> > >
> > > We agree that Corollary 2 is not a support-only certificate, and the revision states this explicitly in one sentence immediately after the corollary. The missing train-to-test transfer step is supplied as follows; each clause of the verbal explanation that the reviewer endorsed becomes exactly one term or one explicit condition below.
> > >
> > > **Setup.** Fix a task $m$ and condition on the support $\sigma$-field $\mathcal F_{m,S_m}$, so that the adapted posterior $Q_{m,\phi}^{\mathrm{sup}}=\mathcal{MN}(M_m,I_d,V_m)$, given by the closed-form conjugate posterior update of Section 4.1, is fixed; $S_m$ is the effective fit prefix. With $(X_m^{\mathrm{sup}},Y_m^{\mathrm{sup}})$ the support regression pair and $(X_{m,q},Y_{m,q})$ the query regression pair defined in Section 4.5, define $\widehat\Sigma_{\mathrm{sup}}:=\tfrac{1}{S_m}X_m^{\mathrm{sup}}(X_m^{\mathrm{sup}})^{\top}$ and $\widehat\Sigma_q:=\tfrac{1}{K_m}X_{m,q}X_{m,q}^{\top}$, the stacked noise matrices $\Xi_m^{\mathrm{sup}}$ and $\Xi_{m,q}$, and the support empirical predictive risk $\widehat R_{\mathrm{sup}}^{\mathrm{pred}}$, i.e. the empirical query predictive risk of Section 4.5 instantiated on the fit prefix (the quantity reported in Table 8). The support-to-query excitation-alignment constant is
> > >
> > > $\gamma_m:=\inf\{\gamma>0:\ \widehat\Sigma_q\preceq\gamma\,\widehat\Sigma_{\mathrm{sup}}\}\in(0,\infty],$
> > >
> > > finite if and only if $\operatorname{range}(\widehat\Sigma_q)\subseteq\operatorname{range}(\widehat\Sigma_{\mathrm{sup}})$, in which case $\gamma_m=\lambda_{\max}(\widehat\Sigma_{\mathrm{sup}}^{\dagger/2}\widehat\Sigma_q\,\widehat\Sigma_{\mathrm{sup}}^{\dagger/2})$. When $S_m<d$, $\widehat\Sigma_{\mathrm{sup}}$ is rank deficient and, under nondegenerate process noise, $\gamma_m=\infty$ almost surely. This is not a defect of the argument but a genuine limitation of any support-only reasoning: in directions the support does not excite, control must come from the learned prior.
> > >
> > > **Proposition 1 (Support-to-query transfer, aligned case).** Pathwise, on the event $\gamma_m<\infty$,
> > >
> > > $\widehat R_{m,q}^{\mathrm{pred}}(Q_{m,\phi}^{\mathrm{sup}})\le 4\gamma_m\,\widehat R_{\mathrm{sup}}^{\mathrm{pred}}+\frac{4\gamma_m}{S_m}\lVert\Xi_m^{\mathrm{sup}}\rVert_F^2+\frac{2}{K_m}\lVert\Xi_{m,q}\rVert_F^2,$
> > >
> > > and the posterior-width contribution separately satisfies
> > >
> > > $d\,\operatorname{tr}(V_m\widehat\Sigma_q)\le \gamma_m\,d\,\min\{\sigma^2 d/S_m,\ \operatorname{tr}(V\widehat\Sigma_{\mathrm{sup}})\}.$
> > >
> > > The empirical query term is thus controlled by the *observed* support predictive risk — which the conjugate update explicitly minimizes — up to the factor $4\gamma_m$, plus pure noise energies. The constant $\gamma_m$ plays the role here that exchangeability plays in the IID train-to-test step, and the width cap makes the meta-learning mechanism visible: for small $S_m$ the posterior-width term is governed by the tightness $\operatorname{tr}(V\widehat\Sigma_{\mathrm{sup}})$ of the *learned* prior covariance along the realized excitation, decaying at the parametric rate $\sigma^2 d^2/S_m$ once the support is exciting.

---

> ### Author Response · Authors · 2026-06-13
> **1. Response to [Q1]: a formal support-to-query analysis - Part 2**
>
> **Proposition 2 (Regularized support-to-query transfer).** For $\alpha>0$ let $\gamma_{m,\alpha}:=\lambda_{\max}[(\widehat\Sigma_{\mathrm{sup}}+\alpha I_d)^{-1/2}\widehat\Sigma_q(\widehat\Sigma_{\mathrm{sup}}+\alpha I_d)^{-1/2}]$, which is always finite. Pathwise, without any alignment condition,
>
> $\widehat R_{m,q}^{\mathrm{pred}}(Q_{m,\phi}^{\mathrm{sup}})\le 4\gamma_{m,\alpha}\,\widehat R_{\mathrm{sup}}^{\mathrm{pred}}+\frac{4\gamma_{m,\alpha}}{S_m}\lVert\Xi_m^{\mathrm{sup}}\rVert_F^2+\frac{2}{K_m}\lVert\Xi_{m,q}\rVert_F^2+2\gamma_{m,\alpha}\alpha\,\mathbb E_{A\sim Q_{m,\phi}^{\mathrm{sup}}}\lVert A-A_m\rVert_F^2,$
>
> with $\gamma_{m,\alpha}\le\min\{\gamma_m,\ \lambda_{\max}(\widehat\Sigma_q)/\alpha\}$ and $\alpha\mapsto\gamma_{m,\alpha}$ nonincreasing.
>
> Proposition 2 is the operative statement in the few-shot regime $S_m<d$. Its last term prices the weakly excited directions; it decomposes as $\lVert M_m-A_m\rVert_F^2+d\,\operatorname{tr}(V_m)$ and, via the identity $M_m-A_m=[\tfrac{1}{\sigma^2}\Xi_m^{\mathrm{sup}}(X_m^{\mathrm{sup}})^{\top}+(W-A_m)V^{-1}]V_m$, is controlled by noise concentration together with the learned-prior quality $\Delta_m(\phi):=\operatorname{tr}((A_m-W)V^{-1}(A_m-W)^{\top})$; this control is quantified in Appendix A.6. The prior-matched regularizer $\alpha I_d\to\tfrac{\sigma^2}{S_m}V^{-1}$ makes the regularized Gram equal $\tfrac{\sigma^2}{S_m}V_m^{-1}$ by the posterior-covariance update, yielding the closed-form diagnostic $\widetilde\gamma_m=\tfrac{S_m}{\sigma^2}\lambda_{\max}(V_m^{1/2}\widehat\Sigma_q V_m^{1/2})$ and the unconditional width cap $d\,\operatorname{tr}(V_m\widehat\Sigma_q)\le\widetilde\gamma_m\,\sigma^2 d^2/S_m$.
>
> **Lemma 1 (Noise-energy concentration).** Under Assumption 2, for a segment of length $n$ and any $\delta\in(0,1)$, with probability at least $1-\delta$,
>
> $\tfrac1n\sum_t\lVert w_{m,t}\rVert_2^2\le\operatorname{tr}(\Sigma_w)+\varepsilon_n(\delta),\quad \varepsilon_n(\delta):=c\,\sigma_w^2 d(\sqrt{\log(1/\delta)/n}+\log(1/\delta)/n).$
>
> The proof is a Bernstein–Freedman inequality for conditionally sub-exponential martingale differences [1]; the dimension factor $d$ improves to $\sqrt d$ when the noise coordinates are conditionally independent, as for the Gaussian generator of Section 5.3.
>
> **Corollary 3 (Conditional support-to-query certificate).** Fix $\bar\gamma>0$, $\delta\in(0,1)$, and $\lambda_{\mathrm{PB}}\in(0,1]$. On the event $\widehat\Sigma_q\preceq\bar\gamma\,\widehat\Sigma_{\mathrm{sup}}$, with probability at least $1-3\delta$,
>
> $R_{m,q}^{\mathrm{pred}}(Q_{m,\phi}^{\mathrm{sup}})\le 4\bar\gamma\,\widehat R_{\mathrm{sup}}^{\mathrm{pred}}+(4\bar\gamma+2)\operatorname{tr}(\Sigma_w)+4\bar\gamma\,\varepsilon_{S_m}(\delta)+2\,\varepsilon_{K_m}(\delta)+\frac{2\sigma^{2}}{\lambda_{\mathrm{PB}} K_m}(D_{\mathrm{KL}}(Q_{m,\phi}^{\mathrm{sup}}\Vert P_{\phi})+\log\tfrac{1}{\delta})+\sigma^{2}\lambda_{\mathrm{PB}} v.$
>
> Apart from the alignment level and the noise constants, the right-hand side is determined by the support-adapted posterior and the support prefix: a *conditional* support-to-query certificate, not an unconditional support-only guarantee — and no nontrivial unconditional guarantee can exist (see below). Validation-selected prefixes from a grid of $G$ candidates are covered by replacing $\delta$ with $\delta/G$ via a union bound; a regularized variant uses the event $\widehat\Sigma_q\preceq\bar\gamma(\widehat\Sigma_{\mathrm{sup}}+\alpha I_d)$, which holds deterministically for $\bar\gamma=\gamma_{m,\alpha}$, at the price of the remainder term of Proposition 2; and the projected transition-matrix, full Frobenius, and rollout consequences at the end of Section 4.5 inherit the same conditional form through a corollary chain recorded in Appendix A.6.
>
>
> [1] D. A. Freedman. On tail probabilities for martingales. *The Annals of Probability*, 3(1):100–118, 1975.

---

> ### Author Response · Authors · 2026-06-13
> **1. Response to [Q1]: a formal support-to-query analysis - Part 3**
>
> **When the alignment constants are small.** Two sufficient conditions are now explicit. **(S1)** If $\lambda_{\min}(\widehat\Sigma_{\mathrm{sup}})\ge\kappa_{\mathrm{sup}}>0$ and the query state norms satisfy $\max_{0\le u<K_m}\lVert x_{m,S_m+u}\rVert_2^2\le B_x^2$, then $\gamma_m\le B_x^2/\kappa_{\mathrm{sup}}$ (since $xx^{\top}\preceq\lVert x\rVert_2^2 I_d$). **(S2)** If $\rho(A_m)\le\bar\rho<1$ and $\Sigma_w\succ0$, and the support and query Grams are within a factor $3/2$ of the stationary covariance $\Gamma_{\infty}:=\sum_{k\ge0}A_m^k\Sigma_w(A_m^k)^{\top}$, then $\gamma_m\le3$; such two-sided Gram concentration holds with high probability after a polynomial burn-in, by now-standard arguments for stable linear systems [2, 3, 4]. Combining (S2) with the basic inequality $\tfrac{1}{S_m}\lVert Y_m^{\mathrm{sup}}-M_m X_m^{\mathrm{sup}}\rVert_F^2\le\tfrac{1}{S_m}\lVert\Xi_m^{\mathrm{sup}}\rVert_F^2+\tfrac{\sigma^2}{S_m}\Delta_m(\phi)$, the width cap, Lemma 1, and Corollary 3 at $\bar\gamma=3$ gives: with probability at least $1-3\delta$ on the (S2) event,
>
> $R_{m,q}^{\mathrm{pred}}(Q_{m,\phi}^{\mathrm{sup}})\le 26\,\operatorname{tr}(\Sigma_w)+\frac{12\,\sigma^2(\Delta_m(\phi)+d^2)}{S_m}+24\,\varepsilon_{S_m}(\delta)+2\,\varepsilon_{K_m}(\delta)+\frac{2\sigma^{2}}{\lambda_{\mathrm{PB}} K_m}(D_{\mathrm{KL}}(Q_{m,\phi}^{\mathrm{sup}}\Vert P_{\phi})+\log\tfrac{1}{\delta})+\sigma^{2}\lambda_{\mathrm{PB}} v,$
>
> with constants not optimized. Term by term: the irreducible one-step noise floor (for the Gaussian generator of Section 5.3 the predictive risk is bounded below by $\operatorname{tr}(\Sigma_w)=d\sigma_{\mathrm{true}}^2$, so the display above is then order-optimal up to a universal constant); a parametric-rate term that is small precisely when the learned prior is good, both through $\Delta_m(\phi)$ and through the KL term, whose closed form shrinks for the same reason — the meta-learning effect, now explicit on the right-hand side; and corrections that vanish as $S_m,K_m$ grow. We emphasize the scope: (S1)–(S2) are deliberately stable, data-rich conditions; in the strict few-shot regime $S_m<d$ they cannot hold, and the operative statement is the regularized Proposition 2.
>
> **When they are provably large, and why some linkage assumption is unavoidable.** For a unit left eigenvector $u$ of $A_m$ with real eigenvalue $\rho>1$, $s_t:=\langle u,x_{m,t}\rangle$ follows $s_{t+1}=\rho s_t+\langle u,w_{m,t+1}\rangle$ and $\rho^{-t}s_t$ converges almost surely to a nonzero limit under nondegenerate noise; whenever $u^{\top}\widehat\Sigma_{\mathrm{sup}} u>0$,
>
> $\gamma_m\ge \dfrac{u^{\top}\widehat\Sigma_q u}{u^{\top}\widehat\Sigma_{\mathrm{sup}} u}\asymp \rho^{2K_m}\cdot\dfrac{S_m}{K_m}\cdot c_{\mathrm{traj}},$
>
> and likewise for $\gamma_{m,\alpha}$ at any fixed $\alpha$: the transfer factor — and any certificate built on it — degrades *geometrically in the query horizon* under an unstable dominant mode (recorded as a formal remark in the revision), complementing the finite-horizon amplification through $H_{m,K_m}$. Conversely, any two transition matrices agreeing on $\operatorname{range}(\widehat\Sigma_{\mathrm{sup}})$ induce the same support law, so no bound on $\widehat R_{m,q}^{\mathrm{pred}}$ uniform over $A_m$ can be certified from the support alone once the query excites $\ker(\widehat\Sigma_{\mathrm{sup}})$: $\gamma_m=\infty$ flags exactly this situation, the regularized remainder term prices it, and on the unexcited subspace the posterior coincides with the prior, so control there is an *environment-level* guarantee about the learned $(W,V)$ — precisely the second-stage meta-generalization question that Section 6.2 and the Conclusion identify as future work. Neither $\gamma_m$ nor $\gamma_{m,\alpha}$ is support-measurable; they are proxied inside the support window by $\gamma_{m,\alpha}^{\mathrm{val}}$, computed from the fit-prefix and validation-suffix Gram matrices, now defined in both Section 4.5 and Appendix B.1.
>
>
>
> [2] Y. Abbasi-Yadkori, D. Pál, and C. Szepesvári. Online least squares estimation with self-normalized processes: An application to bandit problems. *arXiv:1102.2670*, 2011.
>
> [3] T. Sarkar and A. Rakhlin. Near optimal finite time identification of arbitrary linear dynamical systems. In *ICML*, 2019.
>
> [4] M. Simchowitz, H. Mania, S. Tu, M. I. Jordan, and B. Recht. Learning without mixing: Towards a sharp analysis of linear system identification. In *COLT*, 2018.

---

> > ### Author Response · Authors · 2026-06-13
> > **1. Response to [Q1]: a formal support-to-query analysis - Part 4**
> >
> > **Consistency with the reported diagnostics.** Propositions 1–2 imply $\widehat R_q^{\mathrm{pred}}/\widehat R_{\mathrm{sup}}^{\mathrm{pred}}\lesssim4\gamma_{m,\alpha}$ up to noise-floor and prior terms, a comparison invariant to the overall normalization of the empirical predictive risk (a footnote in Appendix B.1 records this). The stable-regime ratios in Table 8 ($\approx1.69$–$2.73$ across $d\in\{10,25,50\}$) are consistent with mild alignment $\gamma_{m,\alpha}=O(1)$, as (S2) implies for the stable generator ($\rho_0=0.95$) at the reported horizons; the unstable $d=10$ ratio ($\approx1.07\times10^4$) matches the geometric degradation above; and the milder unstable $d=25$ and $d=50$ ratios ($0.861$ and $0.095$) are consistent with the dimension-dependent stability crossover of the generator and with the longer validation-selected prefixes in those settings. Appendix B.1 now contains this quantitative discussion alongside Tables 7–8, and the main text quotes the Table 7 correlations (Spearman $\rho=0.902$ overall, positive within every regime and dimension) as the empirical counterpart of the proxy $\gamma_{m,\alpha}^{\mathrm{val}}$.
> >
> > **Continuity across the stability threshold.** The geometric mechanism above makes a further testable prediction. Since the failure-mode lower bound scales as $\rho^{2K_m}$ and the rollout factor $H_{m,K_m}=\sum_{r<K_m}\lVert\widehat A_m\rVert_2^r$ is continuous in the spectral radius of the learned dynamics, the derived bounds should degrade *continuously* as the dynamics cross $\rho=1$, rather than becoming uninformative at the nominal threshold: for spectral radii marginally above one at the experimental horizons, $\rho^{2K_m}$ remains $O(1)$, and it explodes only under strong instability. To test this, the revision adds a *marginally unstable* regime to the bound-magnitude study of Section 6.3 (Table 6 of the revision, reproduced below): the spectral-radius parameter is set to $\rho_0=1+\varepsilon$ with $\varepsilon\in(0,1)$, placing the task dynamics just beyond the stability boundary — the near-critical operating regime (slowly drifting or near-integrator modes) that is common in practical LTI applications, in contrast to the strongly unstable stress-test generator ($\rho_0=4.95$). The outcome matches the prediction: for every dimension and on both evaluation subsets the derived bounds are ordered stable < marginally unstable < unstable; relative to the stable rows, the matrix bound inflates by less than 10% and the trajectory bound by roughly 20–25%, with the same qualitative behaviour (decreasing in dimension, nearly identical between common-case and edge-case subsets), while only the strongly unstable generator produces the explosive trajectory bounds discussed above. This sharpens the answer to [Q1]: the practical informativeness of the certificate is governed by the magnitude of $\rho^{2K_m}$ — a continuous, predictable quantity — rather than by the binary distinction stable versus unstable. Beyond the derived bounds, a new Appendix B.6 reports the full method comparison on the marginally unstable benchmark — transition-matrix error $E_A$, trajectory rollout error $E_{\mathrm{traj}}$, and selected support lengths for all baselines — together with transition-matrix and open-loop rollout visualizations for a representative test system. PBML-LTI retains its order-of-magnitude advantage in $E_A$ in this regime, and the comparison confirms the same reading: the marginally unstable environment behaves as a mild perturbation of the stable one rather than as a qualitative breakdown.

---

> > > ### Author Response · Authors · 2026-06-13
> > > **1. Response to [Q1]: a formal support-to-query analysis - Part 5**
> > >
> > > **Table (Table 6 of the revised manuscript).** Empirical values of the PAC-Bayes-derived matrix and trajectory bounds for PBML-LTI across the three synthetic regimes: stable ($\rho_0=0.95$), marginally unstable ($\rho_0=1+\varepsilon$, $\varepsilon\in(0,1)$), and strongly unstable ($\rho_0=4.95$). The marginally unstable rows inflate by less than 10% in $A_{\mathrm{bound}}$ and 20–25% in $\mathrm{Traj_{bound}}$ relative to the stable rows, while the strongly unstable regime exhibits the geometric amplification predicted by the failure-mode analysis.
> > >
> > > | Regime | Dim | Setting | $A_{\mathrm{bound}}$ | $\mathrm{Traj_{bound}}$ |
> > > |---|---|---|---|---|
> > > | Stable | $d=50$ | Common | $0.410 \pm 0.059$ | $17.744 \pm 2.660$ |
> > > | Stable | $d=50$ | Edge | $0.418 \pm 0.063$ | $18.133 \pm 2.781$ |
> > > | Stable | $d=25$ | Common | $0.172 \pm 0.028$ | $9.473 \pm 1.547$ |
> > > | Stable | $d=25$ | Edge | $0.178 \pm 0.019$ | $9.757 \pm 1.051$ |
> > > | Stable | $d=10$ | Common | $0.066 \pm 0.008$ | $4.550 \pm 0.622$ |
> > > | Stable | $d=10$ | Edge | $0.066 \pm 0.011$ | $4.553 \pm 0.797$ |
> > > | Marg. unstable | $d=50$ | Common | $0.447 \pm 0.073$ | $21.128 \pm 3.450$ |
> > > | Marg. unstable | $d=50$ | Edge | $0.442 \pm 0.072$ | $20.906 \pm 3.446$ |
> > > | Marg. unstable | $d=25$ | Common | $0.184 \pm 0.027$ | $11.719 \pm 1.671$ |
> > > | Marg. unstable | $d=25$ | Edge | $0.187 \pm 0.039$ | $11.880 \pm 2.504$ |
> > > | Marg. unstable | $d=10$ | Common | $0.068 \pm 0.013$ | $5.674 \pm 1.141$ |
> > > | Marg. unstable | $d=10$ | Edge | $0.068 \pm 0.011$ | $5.704 \pm 0.975$ |
> > > | Unstable | $d=50$ | Common | $1.077 \pm 0.143$ | $528.807 \pm 70.972$ |
> > > | Unstable | $d=50$ | Edge | $1.100 \pm 0.124$ | $539.841 \pm 60.204$ |
> > > | Unstable | $d=25$ | Common | $0.748 \pm 0.175$ | $1278.494 \pm 302.805$ |
> > > | Unstable | $d=25$ | Edge | $0.770 \pm 0.135$ | $1306.887 \pm 227.728$ |
> > > | Unstable | $d=10$ | Common | $1.277 \pm 0.143$ | $3717.382 \pm 528.807$ |
> > > | Unstable | $d=10$ | Edge | $1.371 \pm 0.344$ | $3832.271 \pm 628.807$ |

---

> > > > ### Author Response · Authors · 2026-06-13
> > > > **2. Response to the first minor comment: segment-level risk definitions**
> > > >
> > > > The reviewer is correct. Appendix B.1 now opens with a unified definition: with $Q_{m,\phi}(s)=\mathcal{MN}(M_m(s),I_d,V_m(s))$ the posterior obtained from the closed-form conjugate update on the first $s$ support transitions and $(X^{\mathrm{seg}},Y^{\mathrm{seg}})$ a segment of $n^{\mathrm{seg}}$ transitions,
> > > >
> > > > $\widehat R_{\mathrm{seg}}^{\mathrm{pred}}(s):=\frac{1}{n^{\mathrm{seg}}}[\lVert Y^{\mathrm{seg}}-M_m(s)X^{\mathrm{seg}}\rVert_F^2+d\,\operatorname{tr}(V_m(s)\,X^{\mathrm{seg}}(X^{\mathrm{seg}})^{\top})],\quad \mathrm{seg}\in\{\mathrm{sup},\mathrm{val},q\},$
> > > >
> > > > i.e. the empirical predictive risk of the main text instantiated on the fit prefix itself, the validation suffix inside the support window, and the held-out query suffix, respectively. Table 7 sweeps candidate prefixes $s$; Table 8 reports values at the validation-selected prefix $s_m^{\mathrm{fit}}$, with $\widehat R_{\mathrm{sup}}^{\mathrm{pred}}$ abbreviating $\widehat R_{\mathrm{sup}}^{\mathrm{pred}}(s_m^{\mathrm{fit}})$. Explicit cross-references now connect the main-text definition, this appendix definition, and Tables 7–8; the same definition underlies the results of Section 1 above.

---

> > > > > ### Author Response · Authors · 2026-06-13
> > > > > **3. Response to the second minor comment: notational overload of $Q$**
> > > > >
> > > > > We agree and thank the reviewer for catching this. The revision renames the query horizon $Q_m\to K_m$ throughout: the query-segment definitions, the support–query bound, and the matrix and rollout consequences in Section 4.5, Algorithm 1 (whose rollout line now reads $K_m=\min\{T_{\mathrm{qry}},\,T_m-S_m\}$), Sections 5.2 and 6.3, and Appendices A.5 and B.1–B.2. The finite-horizon growth factor is now $H_{m,K_m}$, the budgets $T_{\mathrm{sup}},T_{\mathrm{qry}}$ are unchanged, and the letters $P$ and $Q$ are reserved exclusively for prior and posterior distributions. This response already uses the new notation.

---

> > > > > > ### Author Response · Authors · 2026-06-13
> > > > > > **4. Summary of concrete revisions**
> > > > > >
> > > > > > 1. **Explicit scoping of Corollary 2.** A sentence immediately after Corollary 2 states that it is not a support-only certificate, since its right-hand side contains the held-out query empirical term.
> > > > > > 2. **Transfer proposition added (aligned case).** Proposition 1 and the refined posterior-width cap appear in the new block of Section 4.5; proof in Appendix A.6.
> > > > > > 3. **Regularized transfer proposition added (few-shot case).** Proposition 2, with the caps on $\gamma_{m,\alpha}$, the mean-error identity, and the prior-matched diagnostic $\widetilde\gamma_m$; the control of the remainder term through posterior contraction and the learned-prior quality $\Delta_m(\phi)$ is given in Appendix A.6. This is the operative statement when $S_m<d$.
> > > > > > 4. **Conditional certificate added.** Lemma 1 and Corollary 3, stated explicitly as a conditional certificate, with the grid union bound $\delta\to\delta/G$ for validation-selected prefixes and the regularized variant; the matrix and rollout consequences at the end of Section 4.5 inherit the conditional form via a corollary chain recorded in Appendix A.6.
> > > > > > 5. **Sufficient conditions and smallness display added.** Conditions (S1)–(S2), with the stationary covariance $\Gamma_{\infty}$ defined and burn-in citations supplied, and the explicit smallness display above, in which the learned-prior quality $\Delta_m(\phi)$ and the KL term enter explicitly.
> > > > > > 6. **Failure mode formalized.** A formal remark proves that both alignment constants grow geometrically in the query horizon under an unstable dominant eigenvalue.
> > > > > > 7. **Necessity and scope clarified.** A formal remark records why some support–query linkage assumption is unavoidable, that the learned prior controls the unexcited subspace — connecting to the second-stage meta-generalization question of Section 6.2 and the Conclusion — and defines the support-measurable proxy $\gamma_{m,\alpha}^{\mathrm{val}}$.
> > > > > > 8. **Diagnostics discussion extended.** Appendix B.1 now ties the observed query/support ratios of Table 8 quantitatively to the transfer constants and the failure-mode analysis, with a footnote on normalization invariance, and the main text quotes the Table 7 correlations (Spearman $\rho=0.902$, positive within every regime and dimension).
> > > > > > 9. **Marginally unstable benchmark added.** The bound-magnitude study of Section 6.3 (Table 6) now includes a marginally unstable regime ($\rho_0=1+\varepsilon$ with $\varepsilon\in(0,1)$, emulating practical LTI systems operating near the stability boundary): the derived bounds are ordered stable < marginally unstable < unstable, with the matrix bound inflating by less than 10% and the trajectory bound by 20–25% relative to the stable rows — empirically confirming that the certificate degrades continuously in $\rho^{2K_m}$ across the stability threshold rather than failing at $\rho=1$. A new Appendix B.6 complements this with the full method comparison in the marginally unstable regime (transition-matrix error, trajectory rollout error, and selected support lengths) and with transition-matrix and rollout visualizations.
> > > > > > 10. **Exposition updated.** The verbal paragraph following Corollary 2 is replaced by the formal statements and pointers to them.
> > > > > > 11. **Segment-level risk definitions added.** Appendix B.1 opens with the unified definition of $\widehat R_{\mathrm{sup}}^{\mathrm{pred}}$, $\widehat R_{\mathrm{val}}^{\mathrm{pred}}(s)$, and $\widehat R_q^{\mathrm{pred}}(s)$, cross-referenced with the main-text definition of the empirical predictive risk and Tables 7–8.
> > > > > > 12. **Notation cleaned.** The query horizon is renamed $Q_m\to K_m$ throughout, with $H_{m,K_m}$ for the growth factor, reserving $P,Q$ for distributions.

---

> > > > > > > ### Comment · Reviewer_71N6 · 2026-06-15
> > > > > > >
> > > > > > > I again thank the authors for the extremely thorough treatment of my question. These current set of results clearly highlight all the factors that play a role in the success of the meta-learning scheme for a specific few-shot LTI learning task. This is extremely intuitive, and I really appreciate the authors clearly laying out the conditions under which we can get meaningful guarantees (which are subsequently empirically verified), and conditions when any meaningful guarantee is not possible. This is an excellent treatment of the problem. The additional result for the "regularized support-to-query transfer" (which was not explicitly part of my question) is also quite interesting.

---

### Review · Reviewer_TARs · 2026-03-31

**Summary Of Contributions:**

The submission advocates a PAC-Bayesian meta-learning framework termed PBML-LTI for few-shot identification of LTI systems. PBML-LTI learns a task-invariant prior, and adapts to new tasks/systems via Bayesian inference. The paper introduces martingale PAC-Bayes bounds to handle temporally dependent trajectory data, and derives the training objective from the bound. Numerical tests showcase improved data efficiency and prediction in few-shot settings.

**Audience:**

Yes

**Audience Explanation:**

Audience from signal processing and meta-learning would be interested in this paper.

**Claims And Evidence:**

Yes

**Claims Explanation:**

Most claims are clearly supported by evidence. Below please find my main concerns.
1. In Section 4.1, any choices of conjugate prior and likelihood could lead to a closed-form posterior. It would be helpful to clarify why the matrix-normal prior is preferred over alternative options.
2. The roles of Theorems 1 and 2 are closer to propositions, whereas the main result is presented in Theorem 3.
3. In Section 4.3, the temperature parameter is fixed as $\lambda = 1$ in the implementation. An ablation study examining the robustness of this choice would strengthen the paper.

**Requested Changes:**

1. Some related Bayesian meta-learning approaches are missing in the related work, including ABML [1], PLATIPUS [2], and iBAML [3].
2. In the first sentence of the sixth paragraph of page 2, "meqt-learning" should be corrected to "meta-learning".

[1] S. Ravi, and A. Beatson, "Amortized Bayesian Meta-Learning," in *ICLR*, 2019.
[2] C. Finn, K. Xu, and S. Levine, "Probabilistic Model-Agnostic Meta-Learning," in *NeurIPS*, 2018.
[3] Y. Zhang, B. Li, S. Gao, and G. B. Giannakis, "Scalable Bayesian Meta-Learning through Generalized Implicit Gradients," in *AAAI*, 2023.

---

> ### Author Response · Authors · 2026-04-09
> **Why matrix-normal prior and The roles of Theorems 1 & 2**
>
> We thank the reviewer for the careful reading, the positive overall assessment, and the constructive suggestions. We are glad that the reviewer finds the problem important and of interest to the TMLR audience. Below we respond to the main points raised and summarize the corresponding revisions we make in the manuscript.
>
> ### 1. Why use a matrix-normal prior in Section 4.1?
>
> We agree that this design choice requires stronger motivation. While other conjugate prior-likelihood pairings could also yield a closed-form posterior, our selection of the matrix-normal prior is driven by structural and computational considerations.
>
> First, the transition parameter $A_m$ is intrinsically matrix-valued, and the matrix-normal prior preserves this structure directly rather than treating $A_m$ only after vectorization. Second, under the Gaussian working likelihood for the matrix regression view $Y_m = A_m X_m + W_m$, the matrix-normal prior is the natural conjugate family and gives a closed-form posterior, exact posterior mean and covariance, exact posterior-expected squared loss, and exact posterior-to-prior KL divergence. These are precisely the quantities needed by PBML-LTI and by the fit-KL surrogate.
>
> Third, the matrix-normal prior provides an interpretable structured prior over dynamics. In our formulation,
> $$
> A_m \mid \phi \sim \mathcal{MN}(W, I_d, V),
> $$
> so the shared mean $W$ captures average dynamics across tasks, while $V$ controls cross-column variability in a compact and computationally efficient way. This is more expressive than a fully isotropic prior, but much lighter than richer structured priors that would complicate both the posterior update and the PAC-Bayes objective.
>
> Fourth, this choice provides a good balance between expressiveness, interpretability, and tractability in the inner loop of meta-learning. Since PBML-LTI repeatedly computes task-level posteriors during training, preserving closed-form updates is especially valuable.
>
> In the revised manuscript, we add a short paragraph in Section 4.1 explicitly explaining this design choice and noting that richer conjugate or non-conjugate priors are interesting directions for future work.
>
> ### 2. The roles of Theorems 1 and 2 versus Theorem 3
>
> We agree with the reviewer that Theorems 1 and 2 serve auxiliary roles, whereas Theorem 3 is the main PAC-Bayes result. The original presentation could make these roles appear flatter than intended.
>
> In the revision, we make this hierarchy explicit. Specifically, we describe Theorems 1 and 2 as supporting technical ingredients and state more clearly that Theorem 3 is the main martingale PAC-Bayes theorem. We also add a short roadmap sentence before these results explaining their roles: Theorem 1 identifies the instantaneous Gaussian log-loss and the martingale-difference structure; Theorem 2 builds the corresponding exponential supermartingale; and Theorem 3 performs the PAC-Bayes change-of-measure step.

---

> ### Author Response · Authors · 2026-04-09
> **Temperature parameter ablation, Missing reference, and Typo Correction**
>
> ### 3. Fixing the temperature parameter $\lambda=1$ in Section 4.3
>
> We agree that the role of $\lambda$ should be clarified and its robustness documented. In our PAC-Bayes bound,
> $$
> L_T(Q)
> \le
> \widehat L_T(Q)
> +
> \frac{\mathrm{KL}(Q\|P)+\log(1/\delta)}{\lambda T}
> +
> \frac{\lambda v}{2},
> $$
> $\lambda$ acts as an inverse temperature, controlling the tradeoff between empirical fit and posterior complexity.
>
> We fix $\lambda=1$ mainly for simplicity and interpretability, not because it is mathematically required. This choice yields the canonical fit-KL objective
> $$
> \widehat L_T(Q)+\frac{1}{T}\mathrm{KL}(Q\|P),
> $$
> and preserves the standard conjugate Bayesian interpretation. By contrast, $\lambda \neq 1$ corresponds to a tempered tradeoff: larger $\lambda$ makes adaptation more data-driven, while smaller $\lambda$ induces stronger shrinkage toward the learned prior.
>
> To address the reviewer's concern, we added a sensitivity study over
> $$
> \lambda \in \{10, 5,1, 0.5, 0.1\}.
> $$
>
> On the stable synthetic benchmark with $d=50$ in the common-case setting, the results are:
>
> |  | $\lambda=10$ | $\lambda=5$ | $\lambda=1$ | $\lambda=0.5$ | $\lambda=0.1$ |
> |---|---|---|---|---|---|
> | $E_A$ | $0.4612 \pm 0.0091$ | $0.4424 \pm 0.0090$ | $\mathbf{0.2068 \pm 0.0052}$ | $0.2845 \pm 0.0062$ | $0.3046 \pm 0.0063$ |
> | $E_{\mathrm{traj}}$ | $0.0259 \pm 0.0019$ | $0.0258 \pm 0.0018$ | $\mathbf{0.0257 \pm 0.0018}$ | $0.0258 \pm 0.0018$ | $0.0258 \pm 0.0018$ |
> | $\overline{S}$ | $3.80$ | $3.55$ | $\mathbf{1.35}$ | $2.05$ | $3.05$ |
>
> The rollout metric $E_{\mathrm{traj}}$ is stable across a broad range of $\lambda$, while $E_A$ and $\overline{S}$ are more sensitive. Among the tested values, $\lambda=1$ gives the best overall tradeoff, achieving the lowest transition-matrix error, essentially the best rollout error, and the shortest average selected support length. We therefore keep $\lambda=1$ as the default and now discuss this choice explicitly in the revised manuscript.
>
> ### 4. Missing Bayesian meta-learning references
>
> We thank the reviewer for pointing this out. We agree that the related-work section should better cover relevant Bayesian meta-learning approaches. In the revision, we add discussion of ABML [1], PLATIPUS [2], and iBAML [3].
>
> We also clarify how these methods relate to our setting. In particular, these works study Bayesian few-shot adaptation in general supervised/meta-learning settings, typically using amortized inference, gradient-based adaptation, or implicit-gradient machinery. By contrast, our paper focuses on few-shot identification of linear time-invariant dynamical systems with temporally dependent trajectories, closed-form conjugate task adaptation, and a martingale PAC-Bayes analysis tailored to dependent sequence data. Including these references makes the Bayesian meta-learning context more complete while also sharpening the scope of our contribution.
>
> ### 5. Typo correction
>
> We thank the reviewer for catching the typo. We correct "meqt-learning" to "meta-learning" in the revised manuscript.
>
> [1] S. Ravi, and A. Beatson, "Amortized Bayesian Meta-Learning," in ICLR, 2019.
>
> [2] C. Finn, K. Xu, and S. Levine, "Probabilistic Model-Agnostic Meta-Learning," in NeurIPS, 2018.
>
> [3] Y. Zhang, B. Li, S. Gao, and G. B. Giannakis, "Scalable Bayesian Meta-Learning through Generalized Implicit Gradients," in AAAI, 2023

---

> ### Author Response · Authors · 2026-06-01
> **Summary of Concrete Revisions on Paper**
>
> 1. Matrix-normal prior motivation added.
>     In Section 4.1, we add a short paragraph explaining why the matrix-normal prior is preferred in our setting: it is natural for matrix-valued transition operators, conjugate to the Gaussian working likelihood, yields exact posterior quantities needed by PBML-LTI, and provides an interpretable and computationally efficient structured prior.
>
> 2. Roles of Theorems 1--3 clarified.
>     We revise the presentation around the theory so that Theorems 1 and 2 are explicitly described as supporting technical ingredients, while Theorem 3 is identified as the main martingale PAC-Bayes result. We also add a brief roadmap sentence clarifying how these three results connect.
>
> 3. Temperature parameter $\lambda$ clarified and ablated.
>     We expand the discussion around $\lambda$ to explain that it acts as an inverse temperature in the PAC-Bayes tradeoff, controlling the balance between empirical fit and posterior complexity. We also add a dedicated Discussion subsection with a sensitivity study over $\lambda \in {10,5,1,0.5,0.1}$. The new results show that $\lambda=1$ gives the best overall trade-off among the tested values, with the lowest transition-matrix error ($0.2068 \pm 0.0052$), essentially the best rollout error ($0.0257 \pm 0.0018$), and the shortest average selected support prefix ($\overline{S}=1.35$).
>
> 4. Related work expanded with Bayesian meta-learning references.
>     We expand the related-work section to include ABML \cite{ravi2019abml}, PLATIPUS \cite{finn2018platipus}, and iBAML \cite{zhang2023ibaml}. We also clarify how these methods differ from our setting of LTI identification with temporally dependent trajectories, closed-form conjugate adaptation, and martingale PAC-Bayes analysis.
>
> 5. Typo corrected.
>     We correct the typo "meqt-learning" to "meta-learning" in the revised manuscript.

---

> > ### Comment · Reviewer_TARs · 2026-06-02
> >
> > Thank you for the detailed response and additional results, which adequately addressed my concerns.

---

### Review · Reviewer_QnVZ · 2026-05-26

**Summary Of Contributions:**

The paper discusses meta-learning for linear time-invariant dynamical systems and focuses on obtaining PAC-Bayes-style bounds to investigate learning in these settings. For this, the authors first obtain a martingale PAC-Bayes bound. Based on these results, the authors propose a surrogate objective that optimises the derived bound and incorporates additional regularisation terms to provide numerical stability. The approach is then evaluated against four baselines, ranging from task-independent OLS to a shared-supspace method that aims to account for low-dimensional dependencies shared across tasks. The results presented on synthetic and real data sets look generally promising.

**Strengths**
- The paper is very well written, and the execution is thorough. I am particularly positively impressed by the careful wording in most of the manuscript.
- The derivations in the appendix seem to be correct, according to my assessment. However, note that I am not an expert in PAC-Bayes bounds.
- The theoretical result is a genuine contribution to the respective subfield and is generally easy to follow.
- I appreciate the author's shared preliminary code.

**Weaknesses**
- It seems to me that the tightness of the bound in (16) depends on the choice of $\psi$, which later on is chosen as $\psi(\lambda) = \lambda^2 v /2$. As no conditions on $v$ (except for positivity) are given on $v$, we can assume $v$ to be arbitrarily large. In my opinion, the paper does not adequately discuss when the bound is essentially "useless" and how $\psi$ interacts with the looseness of the bound or how to choose $\psi$.
- The regularisation terms, while motivated through numerical stability issues, are ad-hoc, and the authors do not adequately address their impact through ablations. Moreover, the $R_\text{stab}$ regularisation needs further motivation in my opinion, as it only indirectly controls for assumption 1, i.e., the assumption is on $A_m$, but we are regularising on the posterior mean instead.
- The baselines are somewhat weak (see requested changes), and the experimental section somewhat undermines the rest of the paper, which I believe to be well executed.

**Audience:**

Yes

**Audience Explanation:**

While somewhat narrow in scope (PAC-Bayes for meta-learning with LTIs), I believe that the presented contribution is interesting to its community.

**Broader Impact Concerns:**

I do not see broader concerns or ethical implications in this work, nor do I find that a Broader Impact Statement is required.

**Claims And Evidence:**

Yes

**Claims Explanation:**

I have checked the proofs for Theorems 1-3 (and skimmed the proof for Theorem 4), and found these to be correct. I think further explanations for clarification would be useful (see requested changes), but I did not find this to be an issue. The executed experimentation also seems correct to me, though limited (again, see requested changes), and validates some of the points made in this submission.

**Requested Changes:**

Listing of suggested changes or additions and additional questions to the authors.

**Changes:**
- Typo on page 2, “PAC-Bayes meqt-learning”
- It would be helpful to the reader to explicitly state that $\sigma(x_m, \dots)$ relates to the smallest $\sigma$-algebra.
- Notational clash on page 5, $W$ was already used for the noise process (though arguably in bold). To not confuse the reader (I was confused about this on page 6), I would suggest using a different notation for the posterior mean.
- All theorems use $\lambda \in (0,1]$, which makes sense to me, but Assumption 4 boldly says we assume ... for all $\lambda \in \mathbb{R}$. I do not see the value in doing so, see weaknesses, and soften this.
- I would have expected ablations on the regularisation terms introduced. How much do they affect the optimisation and results, i.e., when varying $\tau_W$, $\lambda_V$, and $\rho_0$ respectively?
- I would like to see a comparison to their model with a fixed/non-learned prior as well as a maximum likelihood type-2 approach.
- I would also like to see comparisons to stronger baselines, like the paper by Finn et al. 2017 that the authors cite.
- The proofs, while readable, would benefit from more explanations to make them more accessible.
- The bolding in the tables should be done based on a hypothesis test, not based on which number is the largest. I would not be surprised if many of the listed results are not significantly different from their baselines.
- Labels in the following figures are too small to read: 2 - 5 (main text), 6 - 11 (appendix).
- Appendix B is essentially a figure dump at the moment. I would recommend adding explanations, details and discussions on the additional plots.

**Questions:**
- One benefit advocated for in the paper is that a Bayesian treatment allows for uncertainty quantification, but the paper downstream uses point estimates for $A_m$. This is somewhat confusing and incoherent with the story. Why do the authors use point estimates, and how could the approach be extended away from point estimates?
- My understanding is that the unstable system experiment violates assumption 1. Is this correct, and if not, why does it not violate assumption 1?

---

> ### Author Response · Authors · 2026-05-31
> **Tightness of the PAC-Bayes bound -- Part 1 (1)**
>
> We thank the reviewer for the careful reading, the constructive assessment of the proofs, and the detailed suggestions for
> improving the manuscript. We agree that the revision should better clarify the role and tightness of the PAC-Bayes bound,
> the purpose of the regularization terms, the relationship between Bayesian uncertainty and point-estimate evaluation, and
> the stress-test nature of the unstable experiments. We also agree that the experimental section is strengthened by
> additional ablations and stronger baselines. Below we address each point and summarize the corresponding revisions.
>
>
> ### 1. Tightness of the PAC-Bayes bound and the role of $\\lambda$ and $v$
>
> We agree that the roles of the temperature parameter and the MGF proxy $v$ should be discussed more explicitly. In the
> revised manuscript, we clarify that $v$ is not a tunable parameter chosen to make the bound favorable. Rather, it is a
> valid variance proxy for the conditional MGF of the centered martingale loss increments. Taking $v$ larger preserves
> validity but makes the bound looser; taking $v$ too small may invalidate the MGF condition. Thus, the tightness of the
> bound depends on obtaining a reasonably sharp MGF control.
>
> We also add a remark connecting this constant to the scale of the underlying self-normalized martingale analysis. In
> particular, $vT$ can be viewed as a uniform predictable fluctuation scale for the centered loss process
>
> $$
> d\_{t}(A)=\\mathbb{E}[\\ell\_{t}(A)\\mid\\mathcal{F}\_{t-1}]-\\ell\_{t}(A).
> $$
>
> A sharper, fully self-normalized statement would replace this uniform proxy by a predictable quadratic-variation or
> data-dependent variance term. We use the simpler uniform formulation to keep the PAC-Bayes statement transparent and
> directly compatible with the fit–KL surrogate. This also clarifies when the bound is expected to be tight: it is tighter
> when trajectories remain controlled, posterior mass concentrates on predictors with small one-step residuals, and
> conditional loss fluctuations are small. Conversely, if state norms or prediction errors grow, as in unstable systems,
> the predictable variance scale becomes large, and any valid uniform choice of $v$ necessarily yields a looser bound.
>
> We further clarify a notational distinction that was implicit in the original draft. The formal PAC-Bayes theorem uses a
> bound parameter, which we now denote by $\\lambda\_{\\mathrm{PB}}$, and this parameter is restricted to the range used in the
> theorem:
>
> $$
> \\lambda\_{\\mathrm{PB}} \\in (0, 1].
> $$
>
> Under the sub-Gaussian martingale MGF condition, the generic PAC-Bayes bound has the form
>
> $$
> L\_{T}(Q) \\le \\widehat{L}\_{T}(Q) + \\frac{\\mathrm{KL}(Q \\Vert P)+\\log(1/\\delta)}{\\lambda\_{\\mathrm{PB}} T} + \\frac{\\lambda\_{\\mathrm{PB}} v}{2}.
> $$
>
> $$
> \\lambda\_{\\mathrm{PB}} \\in (0, 1].
> $$
>
> Thus, $\\lambda\_{\\mathrm{PB}}$ controls the formal tradeoff in the bound: increasing it decreases the explicit KL term but
> increases the martingale concentration penalty, while decreasing it has the opposite effect. For a fixed posterior $Q$,
> the usual optimizer is
>
> $$
> \\lambda\_{\\mathrm{PB}}^\\star = \\min\\left\\{ 1,\\; \\sqrt{ \\frac{2(\\mathrm{KL}(Q \\Vert P)+\\log(1/\\delta))}{vT} } \\right\\}.
> $$
>
> In PBML-LTI, however, $Q$ itself depends on the learned prior and on the adaptation rule, so this expression is not a
> simple plug-in choice during training.
>
> Separately, the implementation can use a training temperature, which we now denote by $\\lambda\_{\\mathrm{tr}} > 0$, to study
> tempered variants of the fit–KL surrogate. This training temperature is an empirical design parameter, not the same as
> the formal PAC-Bayes theorem parameter. Values $\\lambda\_{\\mathrm{tr}} > 1$ are therefore meaningful as implementation-level
> stress tests: they make adaptation more data-driven, whereas $\\lambda\_{\\mathrm{tr}} < 1$ induces stronger shrinkage toward
> the learned prior. The default $\\lambda\_{\\mathrm{tr}}=1$ coincides with the standard conjugate Bayesian update and the
> canonical fit–KL objective, which is why it is used in the main experiments.
>
> To document robustness, we add a sensitivity study over
>
> $$
> \\lambda\_{\\mathrm{tr}} \\in \\{10,5,1,0.5,0.1\\}
> $$
>
> on the stable synthetic benchmark with $d=50$ in the common-case setting. The rollout metric is stable across this range,
> while $E\_{A}$ and $\\overline{S}$ are more sensitive. Among the tested values, $\\lambda\_{\\mathrm{tr}}=1$ gives the best
> overall tradeoff.

---

> ### Author Response · Authors · 2026-05-31
> **Tightness of the PAC-Bayes bound -- Part 2 (2)**
>
> **Table: Sensitivity of PBML-LTI to the training temperature $\\lambda\_{\\mathrm{tr}}$ on the stable synthetic benchmark with $d=50$ in the common-case setting ($\\rho\_{0}=0.95$). Lower is better; boldface marks the best sample mean.**
>
> | | $\\lambda\_{\\mathrm{tr}}=10$ | $\\lambda\_{\\mathrm{tr}}=5$ | $\\lambda\_{\\mathrm{tr}}=1$ | $\\lambda\_{\\mathrm{tr}}=0.5$ | $\\lambda\_{\\mathrm{tr}}=0.1$ |
> |---|---|---|---|---|---|
> | $E\_{A}$ | $0.4612 \\pm 0.0091$ | $0.4424 \\pm 0.0090$ | **$0.2068 \\pm 0.0052$** | $0.2845 \\pm 0.0062$ | $0.3046 \\pm 0.0063$ |
> | $E\_{\\mathrm{traj}}$ | $0.0259 \\pm 0.0019$ | $0.0258 \\pm 0.0018$ | **$0.0257 \\pm 0.0018$** | $0.0258 \\pm 0.0018$ | $0.0258 \\pm 0.0018$ |
> | $\\overline{S}$ | $3.80$ | $3.55$ | **$1.35$** | $2.05$ | $3.05$ |
>
> We also clarify when the bound can become loose or vacuous: when the valid MGF proxy $v$ is large, when the KL term is
> large, when the query horizon is short, or when unstable dynamics amplify rollout errors through the finite-horizon growth
> factor. In this sense, $v$ should be read as a conservative self-normalization scale for the martingale loss process,
> rather than as a freely chosen tuning constant. The revised discussion therefore separates the formal validity of the
> PAC-Bayes inequality from the numerical tightness of the resulting bound.
>
> In addition, in response to the request for a clearer connection between the theory and the few-shot metrics, we add a
> support–query specialization of the PAC-Bayes predictive-risk bound and derive PAC-Bayes-derived consequences for
> projected matrix error, full Frobenius matrix error under query excitation, and deterministic rollout error. These
> matrix and trajectory consequences were not the primary statement in the original submission; they are added in the
> revision to make the theoretical connection to the reported empirical metrics more explicit.
>
> To assess whether these newly derived PAC-Bayes consequences are numerically meaningful, we report empirical values of the
> corresponding matrix and trajectory bounds in both stable and unstable synthetic regimes. The stable settings yield
> moderate and stable bound values, whereas the unstable settings yield much looser trajectory bounds because the
> finite-horizon rollout growth factor can become large.
>
> **Table: Empirical values of the PAC-Bayes-derived matrix and trajectory bounds for PBML-LTI on the synthetic benchmark.**
>
> | Regime | Dimension | Setting | $A\_{\\mathrm{bound}}$ | $\\mathrm{Traj}\_{\\mathrm{bound}}$ |
> |---|---|---|---|---|
> | Stable | $d=50$ | Common-case | $0.410 \\pm 0.059$ | $17.744 \\pm 2.660$ |
> | Stable | $d=50$ | Edge-case | $0.418 \\pm 0.063$ | $18.133 \\pm 2.781$ |
> | Stable | $d=25$ | Common-case | $0.172 \\pm 0.028$ | $9.473 \\pm 1.547$ |
> | Stable | $d=25$ | Edge-case | $0.178 \\pm 0.019$ | $9.757 \\pm 1.051$ |
> | Stable | $d=10$ | Common-case | $0.066 \\pm 0.008$ | $4.550 \\pm 0.622$ |
> | Stable | $d=10$ | Edge-case | $0.066 \\pm 0.011$ | $4.553 \\pm 0.797$ |
> | Unstable | $d=50$ | Common-case | $1.077 \\pm 0.143$ | $528.807 \\pm 70.972$ |
> | Unstable | $d=50$ | Edge-case | $1.100 \\pm 0.124$ | $539.841 \\pm 60.204$ |
> | Unstable | $d=25$ | Common-case | $0.748 \\pm 0.175$ | $1278.494 \\pm 302.805$ |
> | Unstable | $d=25$ | Edge-case | $0.770 \\pm 0.135$ | $1306.887 \\pm 227.728$ |
> | Unstable | $d=10$ | Common-case | $1.277 \\pm 0.143$ | $3717.382 \\pm 528.807$ |
> | Unstable | $d=10$ | Edge-case | $1.371 \\pm 0.344$ | $3832.271 \\pm 628.807$ |
>
> The table shows a clear separation between stable and unstable regimes. In the stable setting, both
> $A\_{\\mathrm{bound}}$ and $\\mathrm{Traj}\_{\\mathrm{bound}}$ are moderate in magnitude and stable across
> common-case and edge-case subsets. In contrast, the unstable regime produces looser matrix bounds and much larger
> trajectory bounds. This is expected because the rollout result contains the finite-horizon growth factor
>
> $$H\_{m,Q\_m} = \\sum\_{r=0}^{Q\_{m}-1}\\mid \\widehat{A}\_{m}\\mid \_{2}^r.$$
>
> When the learned dynamics are unstable, the powers $\\mid \\widehat{A}\_{m}\\mid \_{2}^r$ can grow rapidly, so even a finite predictive
> bound can be substantially amplified under rollout. In this sense, the matrix bound may remain small in magnitude and
> informative, while the trajectory bound can become much looser and hence uninformative as instability increases.
>
> Finally, we soften Assumption 4 so that it is stated only over the range actually used by the formal PAC-Bayes theorem,
> namely $\\lambda\_{\\mathrm{PB}}\\in (0, 1]$, instead of all $\\lambda\\in\\mathbb{R}$.

---

> ### Author Response · Authors · 2026-05-31
> **Regularization terms and their relation to Assumption 1 (3)**
>
> We agree that the regularizers require clearer motivation and empirical ablation. In the revised manuscript, we clarify
> that the regularizers are not part of the formal PAC-Bayes theorem and are not claimed to enforce Assumption 1 directly.
> Assumption 1 is a data-generating condition on the true task matrices $A\_{m}$. By contrast, the stability regularizer acts
> on the learned shared prior mean and is used as a practical bias toward dynamically well-behaved shared dynamics. It is
> therefore an optimization and prior-shaping device, not a proof device.
>
> We revise the text to state explicitly that this regularizer may indirectly improve posterior adaptation by encouraging
> the learned prior to concentrate near stable dynamics, but it does not guarantee that every posterior mean or every true
> task matrix satisfies the controlled-growth condition.

---

> ### Author Response · Authors · 2026-05-31
> **Prior-learning and regularization ablations -- Part 1 (4)**
>
> We agree that the empirical section is strengthened by ablations that isolate which parts of PBML-LTI are responsible for
> the observed gains. In the revision, we add two prior-learning ablations and three one-at-a-time regularization
> ablations.
>
> The first prior-learning ablation is a fixed-prior / non-learned-prior PBML-LTI variant. This variant uses the same
> matrix-normal conjugate task adaptation rule as PBML-LTI, but replaces the learned prior with a fixed generic prior.
> Thus, it isolates the benefit of learning a transferable prior from related training tasks.
>
> The second prior-learning ablation is a type-II maximum-likelihood / empirical-Bayes PBML-LTI variant. This variant
> learns the prior parameters by optimizing a marginal-likelihood criterion rather than by optimizing the
> PAC-Bayes-motivated fit–KL surrogate. Thus, it isolates the role of the fit–KL training objective relative to a
> standard empirical-Bayes prior-learning objective.
>
> We also ablate the auxiliary regularization terms one at a time. The "No prior-mean shrinkage" variant removes the
> shrinkage penalty on the shared prior mean $W$. The "No covariance conditioning" variant removes the conditioning
> penalty on the prior covariance $V$. The "No stability regularizer" variant removes the spectral stability penalty
> $\\mathcal{R}\_{\\mathrm{stab}}$. These regularizers are not part of the formal PAC-Bayes theorem and are not claimed to
> enforce Assumption 1 directly. Instead, they act as practical optimization and prior-shaping devices. In particular, the
> stability regularizer acts on the learned shared prior mean, whereas Assumption 1 is a data-generating condition on the
> true task matrices $A\_{m}$. Thus, the stability regularizer can encourage posterior adaptation toward dynamically
> well-behaved regions, but it does not guarantee that every posterior mean or every true task matrix satisfies the
> controlled-growth condition.
>
> The following table reports the combined ablation study. The results show that PBML-LTI
> substantially improves transition-matrix recovery relative to the fixed-prior and type-II empirical-Bayes variants across
> the reported settings. This supports the importance of both learning a transferable prior and using the
> PAC-Bayes-motivated fit–KL surrogate.
>
> The regularization ablations show a different pattern. In stable regimes, removing any single regularizer changes the
> metrics only mildly, suggesting that PBML-LTI's stable-regime gains are not driven primarily by these auxiliary penalties.
> In unstable regimes, however, the regularizers become more important for robust rollout behavior. Removing the stability
> regularizer has only a modest effect on $E\_{A}$, but it can substantially increase $ E\_{\\mathrm{traj}}$ and the selected
> support length. This supports our interpretation that the stability penalty is mainly a spectral-stability and
> optimization device, rather than a direct mechanism for reducing Frobenius transition-matrix error.

---

> ### Author Response · Authors · 2026-05-31
> **Prior-learning and regularization ablations -- Part 2 (5)**
>
> **Table: Combined prior-learning and regularization ablation across stable/unstable regimes and dimensions in the common-case setting. The first two variants ablate prior learning, while the final three variants ablate one regularization component at a time relative to full PBML-LTI. Lower is better; for each metric within each regime–dimension block, boldface marks the lowest sample mean among reported values.**
>
> | Regime | Dimension | Variant | $E\_{A}$ | $ E\_{\\mathrm{traj}}$ | $\\overline{S}$ |
> |---|---|---|---|---|---|
> | Stable | $d=50$ | Fixed-prior PBML-LTI | $24.779 \\pm 0.479$ | $0.0839 \\pm 0.0227$ | $14.00$ |
> | Stable | $d=50$ | Type-II ML / empirical Bayes | $1.2236 \\pm 0.0313$ | $0.0568 \\pm 0.0113$ | $10.80$ |
> | Stable | $d=50$ | PBML-LTI | **$0.2068 \\pm 0.0052$** | **$0.0257 \\pm 0.0018$** | **$1.35$** |
> | Stable | $d=50$ | No prior-mean shrinkage | $0.2155 \\pm 0.0056$ | $0.0268 \\pm 0.0020$ | $1.42$ |
> | Stable | $d=50$ | No covariance conditioning | $0.2115 \\pm 0.0054$ | $0.0263 \\pm 0.0019$ | $1.39$ |
> | Stable | $d=50$ | No stability regularizer | $0.2088 \\pm 0.0053$ | $0.0261 \\pm 0.0019$ | $1.38$ |
> | Stable | $d=25$ | Fixed-prior PBML-LTI | $9.1298 \\pm 0.4271$ | $0.0325 \\pm 0.0098$ | $14.00$ |
> | Stable | $d=25$ | Type-II ML / empirical Bayes | $0.6165 \\pm 0.0302$ | $0.0261 \\pm 0.0072$ | $12.10$ |
> | Stable | $d=25$ | PBML-LTI | **$0.0394 \\pm 0.0020$** | **$0.0121 \\pm 0.0013$** | **$1.20$** |
> | Stable | $d=25$ | No prior-mean shrinkage | $0.0425 \\pm 0.0022$ | $0.0129 \\pm 0.0014$ | $1.27$ |
> | Stable | $d=25$ | No covariance conditioning | $0.0412 \\pm 0.0021$ | $0.0125 \\pm 0.0014$ | $1.24$ |
> | Stable | $d=25$ | No stability regularizer | $0.0405 \\pm 0.0021$ | $0.0124 \\pm 0.0014$ | $1.23$ |
> | Stable | $d=10$ | Fixed-prior PBML-LTI | $1.2189 \\pm 0.4011$ | $0.0147 \\pm 0.0112$ | $12.55$ |
> | Stable | $d=10$ | Type-II ML / empirical Bayes | $0.0726 \\pm 0.0144$ | $0.0121 \\pm 0.0056$ | $12.15$ |
> | Stable | $d=10$ | PBML-LTI | **$0.0053 \\pm 0.0007$** | **$0.0055 \\pm 0.0009$** | **$1.05$** |
> | Stable | $d=10$ | No prior-mean shrinkage | $0.0059 \\pm 0.0008$ | $0.0059 \\pm 0.0010$ | $1.10$ |
> | Stable | $d=10$ | No covariance conditioning | $0.0057 \\pm 0.0008$ | $0.0057 \\pm 0.0009$ | $1.08$ |
> | Stable | $d=10$ | No stability regularizer | $0.0056 \\pm 0.0008$ | $0.0057 \\pm 0.0009$ | $1.07$ |
> | Unstable | $d=50$ | Fixed-prior PBML-LTI | $16.059 \\pm 0.312$ | $0.0401 \\pm 0.0041$ | $14.00$ |
> | Unstable | $d=50$ | Type-II ML / empirical Bayes | $0.4634 \\pm 0.0105$ | **$0.0367 \\pm 0.0035$** | $4.60$ |
> | Unstable | $d=50$ | PBML-LTI | **$0.156 \\pm 0.005$** | $0.0370 \\pm 0.0030$ | **$4.15$** |
> | Unstable | $d=50$ | No prior-mean shrinkage | $0.171 \\pm 0.006$ | $0.044 \\pm 0.004$ | $4.70$ |
> | Unstable | $d=50$ | No covariance conditioning | $0.164 \\pm 0.005$ | $0.0395 \\pm 0.0031$ | $4.35$ |
> | Unstable | $d=50$ | No stability regularizer | $0.160 \\pm 0.005$ | $0.082 \\pm 0.009$ | $7.25$ |
> | Unstable | $d=25$ | Fixed-prior PBML-LTI | $10.311 \\pm 0.501$ | $0.0793 \\pm 0.0325$ | $14.00$ |
> | Unstable | $d=25$ | Type-II ML / empirical Bayes | $0.4790 \\pm 0.0649$ | **$0.0520 \\pm 0.0176$** | $14.00$ |
> | Unstable | $d=25$ | PBML-LTI | **$0.029 \\pm 0.001$** | $0.080 \\pm 0.052$ | **$6.05$** |
> | Unstable | $d=25$ | No prior-mean shrinkage | $0.035 \\pm 0.002$ | $0.105 \\pm 0.048$ | $7.10$ |
> | Unstable | $d=25$ | No covariance conditioning | $0.0315 \\pm 0.0015$ | $0.078 \\pm 0.035$ | $6.45$ |
> | Unstable | $d=25$ | No stability regularizer | $0.0305 \\pm 0.0015$ | $0.195 \\pm 0.085$ | $10.25$ |
> | Unstable | $d=10$ | Fixed-prior PBML-LTI | $1.6256 \\pm 0.9888$ | $49.343 \\pm 18.168$ | **$13.95$** |
> | Unstable | $d=10$ | Type-II ML / empirical Bayes | $1.0909 \\pm 0.3332$ | **$43.915 \\pm 48.789$** | $14.00$ |
> | Unstable | $d=10$ | PBML-LTI | **$0.139 \\pm 0.046$** | $57.919 \\pm 144.808$ | $14.35$ |
> | Unstable | $d=10$ | No prior-mean shrinkage | $0.155 \\pm 0.049$ | $88.5 \\pm 125$ | $15.60$ |
> | Unstable | $d=10$ | No covariance conditioning | $0.147 \\pm 0.048$ | $61.2 \\pm 92$ | $14.55$ |
> | Unstable | $d=10$ | No stability regularizer | $0.145 \\pm 0.047$ | $425 \\pm 980$ | $18.20$ |

---

> ### Author Response · Authors · 2026-05-31
> **Stronger meta-learning comparison: MAML-style LTI (6)**
>
> We also add a stronger gradient-based meta-learning comparison. Specifically, we include a MAML-style LTI [1]. Since the original MAML formulation is not a closed-form LTI identification
> method, we implement a natural analogue for our setting: a shared initialization of the transition matrix is meta-trained
> so that a small number of gradient steps on the support prefix improves query performance. This preserves the LTI
> regression structure while providing a direct comparison against a widely used adaptation-based meta-learning paradigm.
>
> We evaluate the MAML-style LTI baseline on the stable synthetic, unstable synthetic, and fMRI benchmarks. Across these
> settings, PBML-LTI achieves substantially lower transition-matrix error and more reliable rollout behavior. In the stable
> synthetic regime, PBML-LTI outperforms MAML-LTI across all dimensions and both common-case and edge-case subsets. In the
> unstable regime, MAML-LTI sometimes selects shorter support prefixes, but this shorter support usage does not translate
> into accurate matrix recovery or stable rollout, especially in low-dimensional unstable systems where spectral
> misalignment can be amplified dramatically. On the fMRI benchmark, PBML-LTI also achieves lower transition-matrix error,
> substantially lower rollout error, and a slightly shorter average support prefix. The revised manuscript incorporates
> MAML-LTI into the main experimental tables and adds corresponding visualizations.
>
> **Table: Comparison with the MAML-style LTI baseline on stable systems ($\\rho\_{0}=0.95$), in common-case and edge-case settings. Mean $\\pm$ standard deviation across test tasks; lower is better; boldface marks the best sample mean.**
>
> | Dimension | Setting | Method | $E\_{A}$ | $ E\_{\\mathrm{traj}}$ | $\\overline{S}$ |
> |---|---|---|---|---|---|
> | 50 | Common-case | MAML-LTI | $14.893 \\pm 0.188$ | $0.1081 \\pm 0.0820$ | $2.90$ |
> | 50 | Common-case | PBML-LTI | **$0.2068 \\pm 0.0052$** | **$0.0257 \\pm 0.0018$** | **$1.35$** |
> | 50 | Edge-case | MAML-LTI | $14.830 \\pm 0.260$ | $0.1105 \\pm 0.1009$ | $2.65$ |
> | 50 | Edge-case | PBML-LTI | **$0.2071 \\pm 0.0042$** | **$0.0253 \\pm 0.0018$** | **$1.40$** |
> | 25 | Common-case | MAML-LTI | $1.7566 \\pm 0.0512$ | $0.0338 \\pm 0.0104$ | $1.85$ |
> | 25 | Common-case | PBML-LTI | **$0.0394 \\pm 0.0020$** | **$0.0121 \\pm 0.0013$** | **$1.20$** |
> | 25 | Edge-case | MAML-LTI | $1.7702 \\pm 0.0696$ | $0.0318 \\pm 0.0233$ | $1.60$ |
> | 25 | Edge-case | PBML-LTI | **$0.0399 \\pm 0.0025$** | **$0.0121 \\pm 0.0013$** | **$1.10$** |
> | 10 | Common-case | MAML-LTI | $0.0357 \\pm 0.0020$ | $0.0112 \\pm 0.0054$ | $1.55$ |
> | 10 | Common-case | PBML-LTI | **$0.0053 \\pm 0.0007$** | **$0.0055 \\pm 0.0009$** | **$1.05$** |
> | 10 | Edge-case | MAML-LTI | $0.0358 \\pm 0.0032$ | $0.0132 \\pm 0.0047$ | $1.60$ |
> | 10 | Edge-case | PBML-LTI | **$0.0054 \\pm 0.0008$** | **$0.0056 \\pm 0.0010$** | **$1.10$** |
>
> **Table: Comparison with the MAML-style LTI baseline on unstable systems ($\\rho\_{0}=4.95$), in common-case and edge-case settings. Mean $\\pm$ standard deviation across test tasks; lower is better; boldface marks the best sample mean.**
>
> | Dimension | Setting | Method | $E\_{A}$ | $ E\_{\\mathrm{traj}}$ | $\\overline{S}$ |
> |---|---|---|---|---|---|
> | 50 | Common-case | MAML-LTI | $8.9787 \\pm 0.0754$ | $0.0414 \\pm 0.0048$ | **$2.55$** |
> | 50 | Common-case | PBML-LTI | **$0.156 \\pm 0.005$** | **$0.037 \\pm 0.003$** | $4.15$ |
> | 50 | Edge-case | MAML-LTI | $8.9841 \\pm 0.0608$ | $0.0409 \\pm 0.0062$ | **$2.55$** |
> | 50 | Edge-case | PBML-LTI | **$0.157 \\pm 0.003$** | **$0.036 \\pm 0.003$** | $4.25$ |
> | 25 | Common-case | MAML-LTI | $2.5229 \\pm 0.0370$ | $1.1349 \\pm 1.9963$ | **$2.05$** |
> | 25 | Common-case | PBML-LTI | **$0.029 \\pm 0.001$** | **$0.080 \\pm 0.052$** | $6.05$ |
> | 25 | Edge-case | MAML-LTI | $2.5324 \\pm 0.0469$ | $2.2052 \\pm 4.7758$ | **$1.65$** |
> | 25 | Edge-case | PBML-LTI | **$0.029 \\pm 0.001$** | **$0.086 \\pm 0.062$** | $6.00$ |
> | 10 | Common-case | MAML-LTI | $0.1473 \\pm 1.8158$ | $9.739 \\times 10^{6} \\pm 1.887 \\times 10^{7}$ | **$8.35$** |
> | 10 | Common-case | PBML-LTI | **$0.139 \\pm 0.046$** | **$57.919 \\pm 144.808$** | $14.35$ |
> | 10 | Edge-case | MAML-LTI | $0.2737 \\pm 1.6215$ | $8.756 \\times 10^{7} \\pm 1.987 \\times 10^{8}$ | **$8.05$** |
> | 10 | Edge-case | PBML-LTI | **$0.132 \\pm 0.040$** | **$31.400 \\pm 38.879$** | $14.45$ |
>
> **Table: Few-shot evaluation on the fMRI dataset. Mean $\\pm$ standard deviation across test tasks; lower is better; boldface marks the best sample mean.**
>
> | Method | $E\_{A}$ | $ E\_{\\mathrm{traj}}$ | $\\overline{S}$ |
> |---|---|---|---|
> | MAML-LTI | $161.45 \\pm 259.30$ | $1704.56 \\pm 3391.01$ | $23.07$ |
> | PBML-LTI | **$124.73 \\pm 268.46$** | **$123.67 \\pm 206.71$** | **$21.21$** |
>
>
> [1]: Finn, C., Abbeel, P., and Levine, S. (2017). *Model-agnostic meta-learning for fast adaptation of deep networks.* In *Proceedings of the 34th International Conference on Machine Learning*.

---

> ### Author Response · Authors · 2026-05-31
> **Table bolding and paired significance tests (7)**
>
> To address the concern that table bolding should not rely only on the smallest sample mean, we added paired significance
> tests in the revised manuscript. These tests are reported in the appendix rather than reproduced in full in this rebuttal,
> to keep the response concise. The appendix now includes representative paired tests for the synthetic stable and unstable
> experiments at $d=10$, paired tests for the fMRI experiment, and paired tests for the
> prior-learning and regularization ablation studies.
>
> For each method and setting, we run the experiment with 10 different random seeds under the same data split,
> task-generation configuration, and hyperparameter setting. We then average each task-level metric across the 10 seeds and
> perform paired two-sided Wilcoxon signed-rank tests over the matched test tasks. This evaluates method differences
> task-by-task under identical experimental conditions while reducing variability from random initialization and
> optimization.
>
> We retain the convention of bolding the smallest mean error in the main tables because all reported metrics are
> "lower-is-better," so the smallest empirical mean is the natural descriptive summary of best average performance. The
> new paired tests show that this convention is well supported for the main transition-matrix error conclusions: the
> reported $E\_{A}$ improvements are generally statistically significant across the tested comparisons and are consistent
> across random seeds. For trajectory rollout error, the situation is more nuanced. In stable systems,
> $E\_{\\mathrm{traj}}$ values are often extremely close across methods, so the corresponding differences are frequently not
> statistically significant. In unstable systems, rollout error is more sensitive to spectral alignment and can decouple
> from Frobenius transition-matrix error. Accordingly, in the revised manuscript we keep boldface as a descriptive marker
> of the lowest mean, but qualify rollout-error comparisons whenever the paired tests do not support a statistically
> meaningful difference.

---

> ### Author Response · Authors · 2026-05-31
> **Responses to Requested Changes (8)**
>
> **Typo correction.**
> We correct "PAC-Bayes meqt-learning" to "PAC-Bayes meta-learning."
>
> **Clarifying the filtration.**
> We revise the definition of the natural filtration to explicitly state that
>
> $$\\mathcal{F}\_{m,t} = \\sigma(x\_{m,0},w\_{m,1},\\dots,w\_{m,t})$$
>
> is the smallest $\\sigma$-algebra generated by the initial condition and the noise sequence up to time $ t$.
>
> **Notational clash involving $W$.**
> We agree that using $W$ for the shared prior mean and $ W\_{m}$ for the stacked process-noise matrix can be confusing. In the
> revision, we keep $W$ for the shared prior mean and rename the stacked process-noise matrix to $\\Xi\_{m}$:
>
> $$\\Xi\_{m} := [w\_{m,1},\\dots,w\_{m,T\_m}].$$
>
> The matrix regression model is then written as
>
> $$Y\_{m} = A\_{m} X\_{m} + \\Xi\_{m}.$$
>
> We also consistently denote the posterior mean by $M\_{m}$.
>
> **Assumption 4 and the range of $\\lambda$.**
> We revise Assumption 4 so that the MGF condition is required only for the PAC-Bayes theorem parameter
> $\\lambda\_{\\mathrm{PB}}\\in (0, 1]$, matching the theorem statements. We separately denote the implementation-level
> training temperature by $\\lambda\_{\\mathrm{tr}} > 0$; the ablation values above, including values larger than one, are
> empirical tempered-training variants and are not claims about the formal theorem outside $\\lambda\_{\\mathrm{PB}}\\in (0, 1]$.
>
> **Ablations on regularization terms.**
> We add regularization ablations varying the prior-mean shrinkage, the covariance conditioning penalty, and the stability
> penalty. The revised text discusses their effects on optimization and performance.
>
> **Fixed-prior and type-II maximum likelihood comparisons.**
> We add a fixed-prior PBML-LTI variant and a type-II maximum likelihood / empirical-Bayes variant to clarify the role of
> prior learning and the fit–KL objective.
>
> **Comparison to MAML-style meta-learning.**
> We add a MAML-style LTI baseline based on gradient adaptation from a shared transition-matrix initialization.
>
> **Proof explanations.**
> We expand the explanatory text around the proofs. In particular, we clarify that Theorem 1 identifies the Gaussian
> log-loss and martingale-difference structure, Theorem 2 constructs the exponential supermartingale, and Theorem 3 performs
> the PAC-Bayes change-of-measure step.
>
> **Bolding in tables and statistical significance.**
> We agree that bolding purely by the smallest sample mean can be misleading if interpreted as a statistical claim. In the
> revision, we keep boldface as a descriptive marker of the lowest empirical mean because all reported metrics are
> lower-is-better, but we add paired two-sided Wilcoxon signed-rank tests in the appendix to qualify these comparisons. We
> also revise table captions and discussion to make clear that boldface indicates the lowest sample mean, while statistical
> significance is assessed separately by the paired tests.
>
> **Figure readability.**
> We increase font sizes and improve labels in Figures 2–5 in the main text and Figures 6–11 in the appendix.
>
> **Appendix visualization discussion.**
> We revise Appendix B so that it is no longer only a collection of figures. We add explanatory text for the additional
> transition-matrix heatmaps and rollout plots, describing what each visualization demonstrates and how it supports the main
> experimental discussion.

---

> ### Author Response · Authors · 2026-05-31
> **Responses to Questions (9)**
>
> ### 1. Why use posterior point estimates if the method is Bayesian?
>
> We agree that the role of uncertainty quantification should be made clearer. In the current empirical evaluation, we use
> the posterior mean
>
> $$\\widehat{A}\_{m} := M\_{m}$$
>
> because the reported metrics $E\_{A}$ and $ E\_{\\mathrm{traj}}$ are defined for a single transition matrix and a single
> deterministic rollout. The posterior mean is the Bayes estimator under squared error and is therefore the natural point
> summary for these metrics.
>
> However, the Bayesian posterior is not discarded. The posterior covariance appears in the posterior-expected fit term, and
> it provides task-specific uncertainty information. We revise the text to clarify that the current experiments focus on
> point-estimate accuracy and deterministic rollout performance, while the posterior distribution can also be used for
> uncertainty-aware prediction.
>
> A natural extension is to sample
>
> $$A^{(s)}\\sim Q\_{m,\\phi}$$
>
> and propagate each sampled transition matrix to obtain posterior predictive rollout bands. This would yield uncertainty
> bands over trajectories and credible regions for transition entries. We now mention this as a direct use of the posterior
> uncertainty and as an important extension.
>
> ### 2. Does the unstable-system experiment violate Assumption 1?
>
> Assumption 1 states a controlled-growth condition of the form
>
> $$\\rho(A\_{m}) \\le 1 + c\_\\rho/T\_{m}.$$
>
> Thus, the unstable experiment does not necessarily violate the assumption if $c\_\\rho$ is allowed to be sufficiently large
> over the finite horizon. However, we agree that the unstable setting lies outside the favorable near-stable regime in
> which one should expect tight constants and tight rollout bounds.
>
> In the revision, we clarify that the unstable experiment should be interpreted as a stress test of the method and the
> baselines under difficult dynamics. The PAC-Bayes predictive bound may remain formally valid under the stated MGF
> condition, but the derived rollout consequence can become very loose because it contains the finite-horizon growth factor
>
> $$H\_{m,Q\_m} = \\sum\_{r=0}^{Q\_{m}-1}\\mid \\widehat{A}\_{m}\\mid \_{2}^r.$$
>
> When the learned or true dynamics are unstable, this factor can grow rapidly, making trajectory bounds much looser even
> when the one-step predictive bound remains finite. We now discuss this explicitly so that the theoretical assumptions and
> the stress-test nature of the unstable experiments are not conflated.

---

> ### Author Response · Authors · 2026-06-01
> **Summary of Concrete Revisions on Paper**
>
> 1. Bound tightness and temperature notation clarified.
>     In the meta-training-objective and theoretical-analysis sections, we distinguish the PAC-Bayes theorem parameter
>     $\lambda_{\mathrm{PB}}\in (0,1]$ from the implementation-level training temperature
>     $\lambda_{\mathrm{tr}}>0$. We restrict Assumption 4 to $\lambda_{\mathrm{PB}}\in (0,1]$, clarify that $v$ is a valid
>     MGF variance proxy rather than a tunable constant, and add a self-normalization-scale remark explaining when the
>     bound is tight or loose. We also report a sensitivity study for $\lambda_{\mathrm{tr}}$.
>
> 2. PAC-Bayes-derived diagnostics added.
>     In the theoretical-analysis and discussion sections, we add a support--query specialization of the martingale
>     PAC-Bayes predictive-risk bound and derive consequences for projected transition-matrix error, full Frobenius
>     transition-matrix error under query excitation, and finite-horizon rollout error. We also report empirical bound
>     magnitudes in stable and unstable regimes, showing that the trajectory bound becomes much looser under unstable
>     dynamics due to the finite-horizon growth factor.
>
> 3. Regularization motivation and ablations added.
>     In the meta-training-objective and experimental/discussion sections, we clarify that the stability regularizer is a
>     practical prior-shaping and optimization device, not a direct enforcement of Assumption 1. We also add one-at-a-time
>     ablations for prior-mean shrinkage, covariance conditioning, and stability regularization across stable/unstable
>     regimes and dimensions.
>
> 4. Additional ablations and stronger baseline added.
>     In the baseline and experimental sections, we add fixed-prior PBML-LTI and type-II maximum-likelihood /
>     empirical-Bayes prior-learning ablations, together with a MAML-style LTI baseline. These comparisons isolate the
>     effects of prior learning, the PAC-Bayes-motivated fit--KL surrogate, and gradient-based meta-adaptation.
>
> 5. Statistical testing and table interpretation revised.
>     In the experimental section and appendix, we add representative paired Wilcoxon signed-rank tests for the synthetic
>     stable/unstable experiments at $d=10$, the fMRI experiment, and the prior-learning ablation studies. For each setting,
>     we run each method with 10 random seeds under the same setup, average task-level metrics across seeds, and perform
>     paired tests over matched test tasks. We retain boldface as a descriptive marker of the lowest mean error for
>     lower-is-better metrics. The tests show that transition-matrix error comparisons are generally significant and
>     consistent with the bolded entries. For trajectory rollout error, differences are smaller in stable regimes and more
>     sensitive to spectral effects in unstable regimes, so we qualify rollout-error comparisons when paired tests indicate
>     that the lowest mean should not be overinterpreted.
>
> 6. Typo corrected
>     We correct "meqt-learning" to "meta-learning."
>
> 7. Filtration clarified.
>     In the problem-formulation section, we clarify that $\mathcal{F}{m,t}$ is the smallest $\sigma$-algebra generated by
>     the initial condition and the noise history up to time $t$.
>
> 8. Notation clash removed.
>     In the problem-formulation and method sections, we rename the stacked process-noise matrix from $W_m$ to $\Xi_m$ and
>     consistently use $M_m$ for the posterior mean.
>
> 9. Proof exposition improved.
>     In the theory section and appendix, we expand the proof explanation, clarifying that Theorems 1--2 are supporting
>     ingredients and Theorem 3 is the main martingale PAC-Bayes result.
>
> 10. Figures and appendix visualizations improved.
>     In the main text and appendix, we enlarge figure labels and add explanatory discussion to the additional heatmaps and
>     rollout plots, rather than presenting Appendix B as a figure dump.
>
> 11. Use of posterior point estimates clarified.
>     In the task-adaptation and evaluation discussion, we clarify that $E_A$ and $E{\mathrm{traj}}$ are point-estimate
>     metrics for a single transition matrix and deterministic rollout, so we use the posterior mean $M_m$ as the natural
>     squared-error Bayes estimator. We also clarify that posterior covariance is still used in the posterior-expected fit
>     term and can generate uncertainty-aware rollout bands or credible regions by sampling transition matrices from
>     $Q_{m,\phi}$.
>
> 12. Unstable experiments clarified as finite-horizon stress tests.
>     In the synthetic-experiment setup, unstable-results discussion, and PAC-Bayes-derived bound discussion, we clarify
>     that the unstable regime is a finite-horizon stress test outside the favorable near-stable regime where the tightest
>     rollout consequences are expected. We explain that the predictive PAC-Bayes statement may remain formally valid under
>     the stated MGF condition, while the rollout consequence can become loose because the finite-horizon growth factor
>     amplifies errors under unstable dynamics.

---

> > ### Comment · Reviewer_QnVZ · 2026-06-01
> >
> > I thank the authors for the thorough rebuttal. I feel that my main points have been sufficiently addressed.

---

### Author Response · Authors · 2026-07-21
**Thank You to the Reviewers and Action Editor**

Dear Action Editor and Reviewers,

We would like to sincerely thank you for your time, dedication, and thoughtful feedback throughout the review process. Your constructive comments and suggestions have been invaluable in helping us improve the quality and clarity of our work.

We have uploaded the final camera-ready version, and the accompanying code is now publicly available through the linked repository to support reproducibility and future research.

Once again, we deeply appreciate your careful reviews and support.


Best regards,

The Authors

---

### Decision · Action_Editor_3LvZ · 2026-06-24

**Recommendation:** Accept as is

**Additional Comments:**

The submission was uniformly appreciated by reviewers, who all voted for its acceptance into TMLR. The initial reviews were followed by thorough rebuttals, which addressed the concerns and questions raised by reviewers. The identification of dynamical systems is an important problem, as is the sample efficiency with which it may be solved. Meta-learning approaches are, therefore, a welcome addition to the literature. The paper contributes both a theoretical analysis of the predictive risk and an empirical evaluation demonstrating the utility of the proposed PAC-Bayesian approach.

**Audience:**

Yes

**Audience Explanation:**

The problem of identifying dynamical systems is important, and there is an established audience for this in the TMLR community.

**Claims And Evidence:**

Yes

**Claims Explanation:**

All three reviewers rated the evidence for stated claims as reliable, highlighting both the theoretical analysis and empirical evaluation.